# Frequency and mechanisms of LINE-1 retrotransposon insertions at CRISPR/Cas9 sites

Jianli Tao [1,4✉], Qi Wang [1,4], Carlos Mendez-Dorantes [2], Kathleen H. Burns [2] & Roberto Chiarle [1,3✉]

CRISPR/Cas9-based genome editing has revolutionized experimental molecular biology and entered the clinical world for targeted gene therapy. Identifying DNA modifications occurring at CRISPR/Cas9 target sites is critical to determine efficiency and safety of editing tools. Here we show that insertions of LINE-1 (L1) retrotransposons can occur frequently at CRISPR/Cas9 editing sites. Together with PolyA-seq and an improved amplicon sequencing, we characterize more than 2500 de novo L1 insertions at multiple CRISPR/Cas9 editing sites in HEK293T, HeLa and U2OS cells. These L1 retrotransposition events exploit CRISPR/Cas9-induced DSB formation and require L1 RT activity. Importantly, de novo L1 insertions are rare during genome editing by prime editors (PE), cytidine or adenine base editors (CBE or ABE), consistent with their reduced DSB formation. These data demonstrate that insertions of retrotransposons might be a potential outcome of CRISPR/Cas9 genome editing and provide further evidence on the safety of different CRISPR-based editing tools.

[1] Department of Pathology, Boston Children's Hospital and Harvard Medical School, Boston, MA 02115, USA. [2] Department of Oncologic Pathology, Dana-Farber Cancer Institute, Boston, MA 02115, USA. [3] Department of Molecular Biotechnology and Health Sciences, University of Torino, Torino 10126, Italy. [4]These authors contributed equally: Jianli Tao, Qi Wang. ✉email: jianli.tao@childrens.harvard.edu; roberto.chiarle@childrens.harvard.edu

Genome editing with CRISPR/Cas9 holds great promise for the treatment of human genetic diseases[1], resulting in a mix of intended and unintended genetic alterations. Canonical CRISPR/Cas9 DNA editing induces double-strand breaks (DSBs) that are resolved by the classical non-homologous end joining pathway (c-NHEJ) and typically result in insertions or deletions (indels) of relatively small DNA sequences at both on-target and off-target sites, leading to disruption of the target sequence[2–4]. While c-NHEJ is the main cellular DNA repair pathway of CRISPR/Cas9-induced DSBs, homology-directed repair (HDR) can compete by inserting homologous DNA sequences that result in precise gene editing[5]. More rarely, DSBs induced by CRISPR/Cas9 can lead to larger genomic rearrangements such as large chromosomal deletions, inversions, or translocations[6–10] or even more catastrophic events such as chromothripsis[11] and chromosome loss[12–14]. Additional outcomes of the canonical CRISPR/Cas9 editing include integrations of exogenous sequences including lentivirus[15,16], adeno-associated virus (AAV)[17], plasmids[9,18,19], and small DNA fragments[20,21]. These events exploit the availability of frequent DSBs generated by CRISPR/Cas9 to favor the insertion of exogenous DNA fragments using microhomology-mediated end joining (MMEJ)[17,22].

While it is known that mobile genetic elements can be captured at DSB sites[23–28], little is known on the frequency and mechanisms of retrotransposon integration into CRISPR/Cas9 editing sites because few such events have been described so far only in mouse zygotes[28]. Retrotransposons are self-propagating sequences that generate de novo insertions through reverse transcription of RNA intermediates. In humans, the only autonomously active family is the long interspersed element-1 (LINE-1 or L1)[29]. L1-derived sequences are prevalent in the genome (~17% of the human genome[30]), and ~100 loci are full-length, retrotransposition competent L1s[31–35]. These competent sequences code for two proteins: open reading frame 1 protein (ORF1p) and ORF2p. ORF1p associates with L1 RNA to serve as an RNA chaperone[36], while ORF2p contains endonuclease (EN)[37] and reverse transcriptase (RT)[29] activities that are critical for retrotransposition of L1 RNA[38]. Recently, the ORF0 was identified on the antisense strand of primate LINE-1 5′UTRs. ORF0 can induce the generation of fusion proteins of ORF0 with proximal exons and is translated as a short peptide that enhances LINE-1 mobility[39].

Canonical L1 retrotransposition takes place by target-primed reverse transcription (TPRT)[40,41]. The mechanism includes generation of a DNA nick by the EN activity of ORF2p at the consensus target sequence (5′-TT/AAAA-3′) to expose a 3′-OH followed by the formation of a primer-template structure by base pairing between thymines at the cleavage site and the PolyA stretch at the 3′ end of L1 mRNA[42–44]. The RT activity of ORF2p extends the DNA 3′-OH using the L1 RNA template to generate L1 cDNA, which is process into de novo L1 insertions through poorly understood mechanisms[38]. These L1 insertions are commonly incomplete because the majority of L1 insertions is 5′-truncated. In contrast, EN-independent L1 retrotransposition is hypothesized to occur at sites of damaged DNA and on the lagging strand of DNA replication forks, and does not require to start from intact L1 3′ ends with the constraint of a canonical consensus motif[24,26,43,45].

Two different types of reporters are commonly used to study L1 retrotransposition events in cellular models[46]. A native L1 reporter (L1RP) uses native L1RP sequences and the native L1 5′ UTR promoter to closely mimic endogenous L1 retrotransposition, while a synthetic L1 reporter (L1-ORFeus) contains a codon-optimized human L1 sequence that allows to distinguish de novo L1 insertions from insertions of inactive endogenous L1

fragments. In this work, we describe the occurrence of de novo L1RP and L1-ORFeus insertions into multiple canonical CRISPR/Cas9 editing sites in three human cell lines by PolyA-seq and an improved amplicon sequencing, and use structures of more than 2500 de novo L1 insertion events to infer underlying mechanisms. Moreover, we show that de novo L1-ORFeus insertions are rare in CRISPR-based genome editing tools (PE and BE) that do not require the formation of DSB intermediates.

## Results

**De novo L1-ORFeus insertions occur at DSBs induced by CRISPR/Cas9.** To study the occurrence of de novo L1 retrotransposition at DSB sites induced by canonical CRISPR/Cas9 activity, we examined repair outcomes of generated CRISPR/Cas9 DSBs in human cells expressing a L1 vector competent for retrotransposition. The L1-ORFeus construct contains a codon-optimized human L1 sequence[46] coding for ORF1p, ORF2p, and a GFP reporter for retrotransposition[43,47] (Fig. 1a, Supplementary Fig. 1a, Supplementary Sequence 1).

First, we determined the efficiency of L1 retrotransposition in HEK293T and HeLa cells based on the percentage of induced GFP-expressing cells, and found that the proportion of cells containing the retrotransposed reporter increased over time up to 14 days after transfection and was much higher in HEK293T cells than in HeLa cells, as previously shown[48] (Supplementary Fig. 1b). As previously reported[29,37,38,43], L1-ORFeus mutations that disrupt either the endonuclease (EN) activity (L1-ORFeus-ENm (H230A), hereafter termed as L1-ENm) or reverse transcriptase activity (L1-ORFeus-RTm (D702Y), hereafter termed as L1-RTm) of ORF2p severely limited L1 integrations as compared to wild type (Supplementary Figs. 1a, c–e, 2a, b), despite comparable expression of ORF1 and ORF2 mRNA (Supplementary Fig. 1f) and ORF1p protein (Supplementary Fig. 1g). Given the higher efficiency, for most of the experiments, we decided to use HEK293T cells which have been widely used to test on-target and off-target activities of CRISPR/Cas9[10,16,20,49,50].

We targeted one active gene (*MYC*) and one inactive gene (*RAG1*) for inducing DSBs by CRISPR/Cas9, two loci differing in their transcriptional activity and chromatin accessibility. Human RAG1 gene is also a proposed target for gene correction therapy[10,51,52]. We transduced HEK293T cells with the CRISPR/Cas9-containing lentivirus at day 5 of L1 expression and retrotransposition, then we analyzed repair outcomes of CRISPR/Cas9-mediated DSBs at day 11 on the bulk cell population (Fig. 1b). CRISPR/Cas9 editing efficiency on *MYC* and *RAG1* target sites was high and comparable in cells expressing L1-ORFeus, L1-ENm, L1-RTm, or GFP control (Supplementary Figs. 1h, 2c).

De novo insertions of L1-ORFeus were readily distinguishable from endogenous human L1 fragments (Supplementary Sequence 2). By improved amplicon sequencing of the fragments inserted into the CRISPR/Cas9-mediated DSBs (Supplementary Fig. 1i), we identified 546 de novo insertions of the ectopic L1-ORFeus at the *MYC* locus and 734 insertions at the *RAG1* locus (Fig. 1d), consistent with the observation that L1 insertions are independent of gene expression or chromatin accessibility[43,44]. The number of de novo L1-ORFeus insertions at the CRISPR/Cas9-mediated DSBs that were derived from the L1-EN mutant construct, were reduced relative to the L1-ORFeus reporter, and extremely low in the case of L1-RTm (Fig. 1c–e, Supplementary Fig. 2d, f), indicating that the EN activity of the ORF2p was dispensable while the RT activity was required for L1 integrations into CRISPR/Cas9-initiated DSBs. The rare L1-ORFeus insertions found in L1-RTm could be due to trans-complementation from

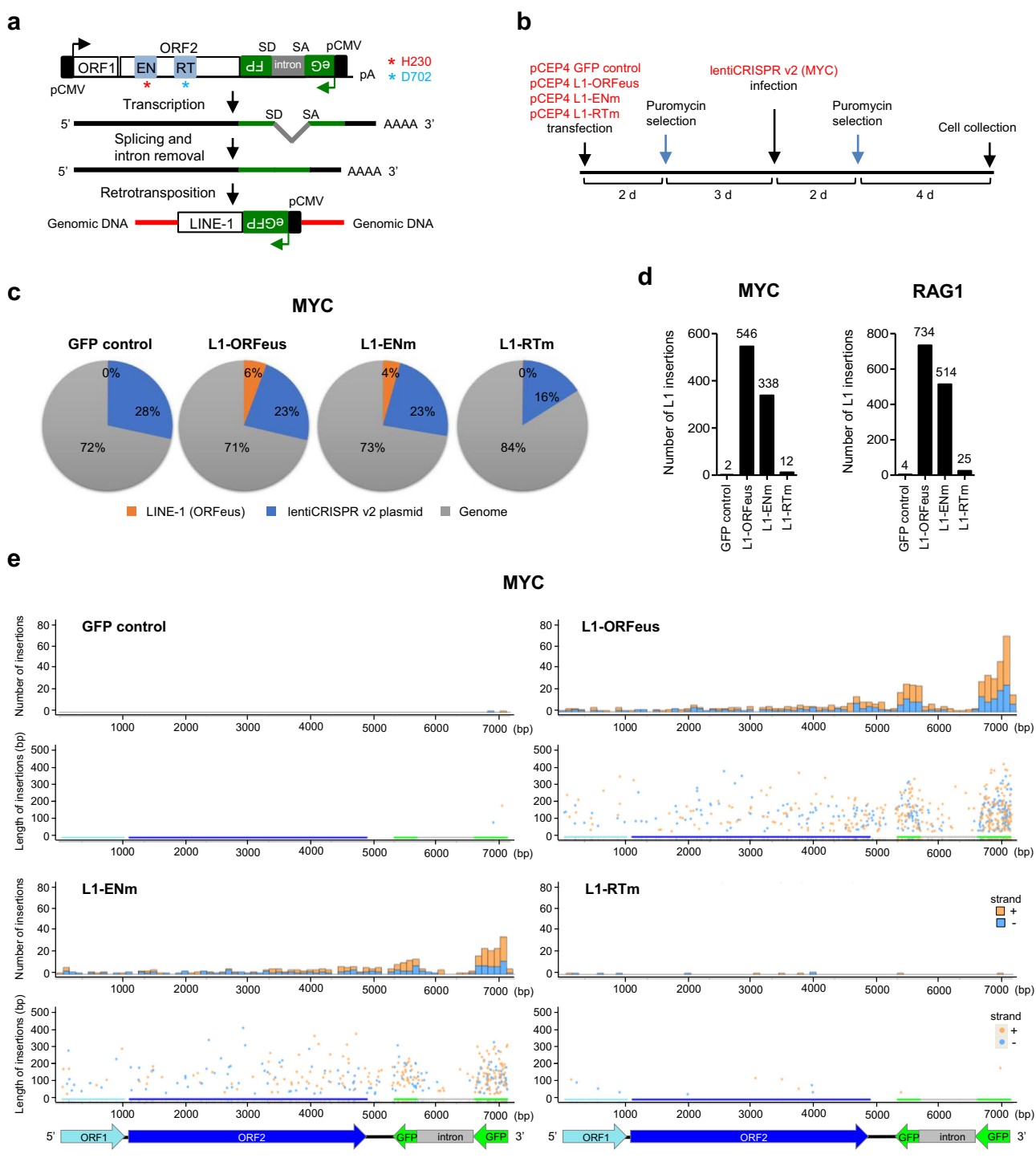

endogenous active L1s or more likely to the L1 plasmid integrations into CRISPR/Cas9-initiated DSB as in GFP control. De novo L1-ORFeus insertions showed a balanced 5′ to 3′ or 3′ to 5′ orientation, suggesting that L1 insertion can equally integrate at either side of the CRISPR/Cas9-mediated DSB (Fig. 2a–c, Supplementary Fig. 2i). L1-ORFeus insertions showed joining to the DSB either direct (no microhomology) or via short microhomologies, mostly 1 bp to 6 bp with a decreasing trend over length (Fig. 2d), consistent with the microhomologies reported for the 5′-junctions of human L1 insertions that occurred via TPRT[53].

More importantly, the distribution of de novo L1 insertions derived from L1-ORFeus or L1-ENm reporters showed a strong enrichment for sequences mapping to the 3′end of L1-GFP reporter and a sharp decrease in coverage spanning the intron (Fig. 1e, Supplementary Fig. 2f). These observations indicate that retrotransposition of the spliced L1-ORFeus at CRISPR/Cas9-mediated DSBs is RT-dependent and initiates at its 3′ end to generate 5′ truncated insertions, reminiscent of canonical L1 insertions mediated by the L1 protein machinery[54]. In contrast, insertions from the lentivirus encoding for Cas9 and the sgRNA were comparable in cells expressing L1-ORFeus, L1-ENm,

**Fig. 1 De novo L1-ORFeus insertions at CRISPR/Cas9 target sites in a reverse transcriptase (RT) dependent manner. a** De novo L1-ORFeus retrotransposition reporter. The L1-ORFeus expression plasmid contains an eGFP cassette in the L1 3′UTR which is in the opposite orientation of the L1 and is interrupted by an intron. Cells express eGFP only after the L1-ORFeus transcript undergoes splicing and intron removal, reverse transcription, and integration into chromosomal DNA. **b** Schematic of the experimental strategy employed to examine de novo L1-ORFeus retrotransposition events at the *MYC* CRISPR/Cas9 target site in HEK293T cells. **c** Pie charts show the relative abundance of three main types of obtained by amplicon sequencing at the *MYC* locus targeted by *MYC* CRISPR/Cas9 in HEK293T cells expressing L1-ORFeus, L1-ENm, L1-RTm, or GFP control. **d** Numbers of L1-ORFeus insertions obtained by amplicon sequencing at the *MYC* locus targeted by *MYC* CRISPR/Cas9 (**d**, left) or at the *RAG1* locus targeted by *RAG1* CRISPR/Cas9 (**d**, right) in HEK293T cells expressing L1-ORFeus, L1-ENm, L1-RTm, or GFP control. Number of L1-ORFeus insertions was pooled from 4 independent experiments with similar results. **e** Numbers and fragment lengths of L1-ORFeus insertions obtained by amplicon sequencing at the *MYC* locus targeted by *MYC* CRISPR/Cas9 in HEK293T cells expressing L1-ORFeus, L1-ENm, L1-RTm, or GFP control. The X-axis indicates the nucleotide position of the full-length L1 (ORFeus) reporter cassette. Top: each histogram represents the number of L1 insertions mapping to a 100 bp interval of the L1 (ORFeus) sequence represented on the X-axis. In orange are insertions oriented from 5′ to 3′ end of the L1 (ORFeus) sequence (+), in blue are insertions oriented from 3′ to 5′ (−); bottom: Length of the L1 insertions mapping to the L1 (ORFeus) sequence represented on the X-axis. Each dot indicates the center of the insertion fragment oriented either 5′ to 3′ (+) or 3′ to 5′ (−). Each bar represents the projection of the dot to indicate the nucleotide position in the full-length L1 (ORFeus) reporter cassette. Example L1-ORFeus insertions bridging the intron of GFP in cells expressing L1-ORFeus are shown in Supplementary Sequence 6.

L1-RTm, or GFP control, and evenly distributed with peaks at the flanking LTRs as previously described[17] (Supplementary Figs. 1j, k, 2e, g). Likewise, insertions of genomic fragments were similar in cells expressing each distinct L1 construct, and originated from regions (approx. ±5 kb) flanking the 5′ and 3′ ends of the CRISPR/Cas9-induced DSB, consistent with previous reports[9] (Supplementary Figs. 1j, l, 2e, h).

Furthermore, we found de novo L1-ORFeus insertions at induced DSBs at a third locus (i.e., *CCR5*) where CRISPR/Cas9 has been applied for therapeutic editing to prevent HIV entry into cells[55,56] (Supplementary Fig. 3a, b, d, e), though we detected a lower number of L1-ORFeus insertions compared to *MYC* and *RAG1* loci, likely due to the lower efficiency of the *CCR5* CRISPR/Cas9 (Supplementary Fig. 3c) as well as only one copy of the *CCR5* gene can be edited since HEK293T cells are heterozygous for CCR5 delta 32 mutation[57]. Again, retrotransposition of the spliced L1-ORFeus transcript were observed at *CCR5* CRIPSR/Cas9 editing site only in L1-ORFeus, but not in L1-RTm (Supplementary Fig. 3g), in contrast to insertions of lentiviral plasmid (Supplementary Fig. 3f, h) and genomic fragments (Supplementary Fig. 3f) that were comparable in cells expressing L1-ORFeus or L1-RTm. The relative frequency of L1-ORFeus, genomic fragments, and lentiviral insertions varied between *MYC*, *RAG1*, and *CCR5* editing sites, indicating a competition between these insertional events for the same DSB (Fig. 1c, Supplementary Figs. 2d, 3d).

**De novo L1-ORFeus insertions at DSBs induced by CRISPR/Cas9 are insertions independent of the EN consensus motif.** We next sought to evaluate features of de novo L1-ORFeus reporter insertions at CRISPR/Cas9-initiated DSBs versus de novo L1-ORFeus reporter insertions genome-wide. By adapting the high-throughput genome-wide translocation sequencing (HTGTS) technique[58,59], we implemented one-sided, nested amplification approach to enrich de novo L1 insertion sites for sequencing by capturing the 3′ ends of newly inserted L1 and their flanking genomic DNA. This technique, termed as PolyA-seq, originates from SV40 PolyA and imposes a requirement for L1 insertions to start from the intact 3′ ends and have a minimal 15 bp Poly(A) tail typical of canonical TPRT products, similar to a previously described approach[43] (Supplementary Fig. 4a, b). Using PolyA-seq, we detected >36,000 of de novo L1-ORFeus insertions from L1-ORFeus expressing cells in the entire genome (Supplementary Fig. 4c) that displayed enrichment of the 7mer consensus motif 5′-TT/AAAAA-3′ at L1 pre-integration sites (Supplementary Fig. 4d), consistent with the previously described L1 integration preference mediated by the ORF2p EN

activity[37,43,44]. As expected, these L1-ORFeus insertions ended with 3′ poly(A) tracts (range 15–75 bp) (Fig. 3a, Supplementary Fig. 4e). In contrast, L1-ORFeus insertions obtained by PolyA-seq were markedly diminished with L1-ENm, indicating that PolyA-seq preferentially captured endonuclease-dependent L1 retrotransposition events.

By PolyA-seq, we identified de novo L1-ORFeus insertions from L1-ORFeus expressing cells that peaked in occurrence at the CRISPR/Cas9 target sites and, strikingly, extended for about 5 kilobases 5′ or 3′ of the DSB (Fig. 3b, c, Supplementary Fig. 4g). These L1-ORFeus insertions were markedly reduced with L1-ENm and undetectable with L1-RTm (Fig. 3b). Interestingly, the consensus motif of L1 pre-integration site was lost within ±5 kb of the CRISPR/Cas9-induced DSB whereas it was readily identified in insertions distant from the DSB (Fig. 3d, Supplementary Fig. 4f, h), indicating that L1 retrotransposition in the region flanking the CRISPR/Cas9-induced DSB does not rely on the EN activity of ORF2p.

PolyA-seq also revealed that de novo L1-ORFeus integrations occurred at CRISPR/Cas9 off-target sites, although less frequently than at on-target sites (Fig. 4a, b). Amplicon sequencing at *MYC* CRISPR off-target #1 site (MYC OT1)[59] further validated these findings, in which RT-dependent retrotransposition of the spliced L1-ORFeus transcript were observed (Fig. 4c, d, Supplementary Fig. 5c), although about 50 times lesser frequency than *MYC* on-target. In contrast, insertions of lentiviral plasmid or genomic fragments were comparable in cells expressing L1-ORFeus or L1-RTm (Supplementary Fig. 5b, d). L1 insertions also competed with integrations of genomic DNA and lentiviral plasmid at the off-target sites (Fig. 4c, Supplementary Fig. 5a).

Key experiments were reproduced with CRISPR/Cas9 editing the *MYC* locus and expressing L1-ORFeus or L1-RTm in HeLa (Supplementary Fig. 6a–c) and U2OS cells (Supplementary Fig. 7a–c). Similar to HEK293T cells, amplicon sequencing detected RT-dependent de novo L1-ORFeus retrotransposition events mediated by short microhomologies (Fig. 5a, Supplementary Fig. 6d, e, h, i), in contrast to the comparable lentiviral plasmid and genomic integrations in cells expressing L1-ORFeus or L1-RTm (Supplementary Fig. 6d, f, g). PolyA-seq revealed 307 (HeLa) or 370 (U2OS) RT-dependent de novo L1-ORFeus insertions at the *MYC* locus, respectively (Fig. 5b, c, Supplementary Figs. 6j, k, 7d–g). Like in HEK293T cells, the consensus motif of L1 pre-integration site was lost within ±5 kb of the CRISPR/Cas9-induced DSB (Fig. 5d, Supplementary Fig. 7h).

**RT-dependent de novo L1RP insertions at CRISPR/Cas9 target site.** While the L1-ORFeus reporter is advantageous to distinguish

**a**

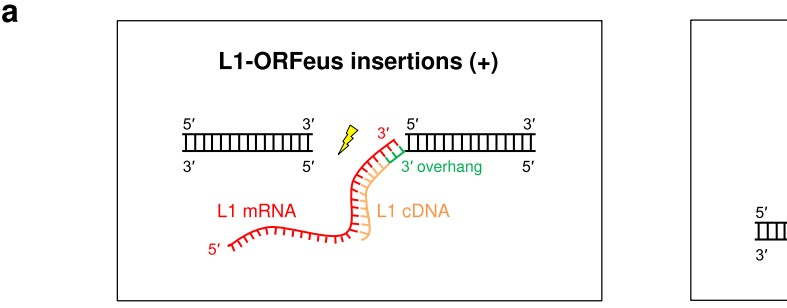
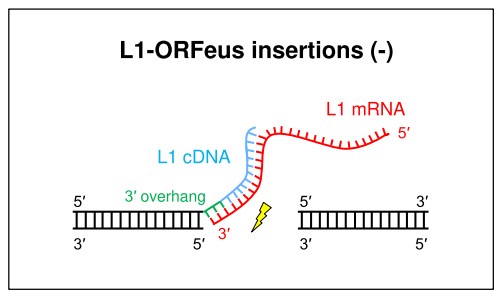

**b**

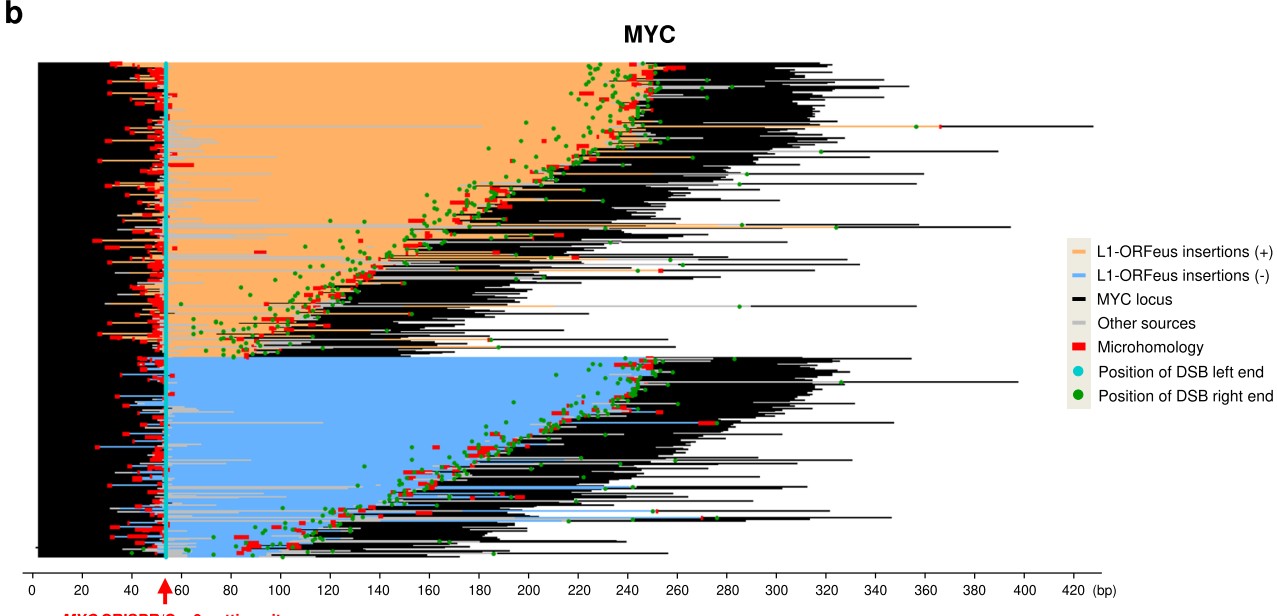

**c**                                                    **d**

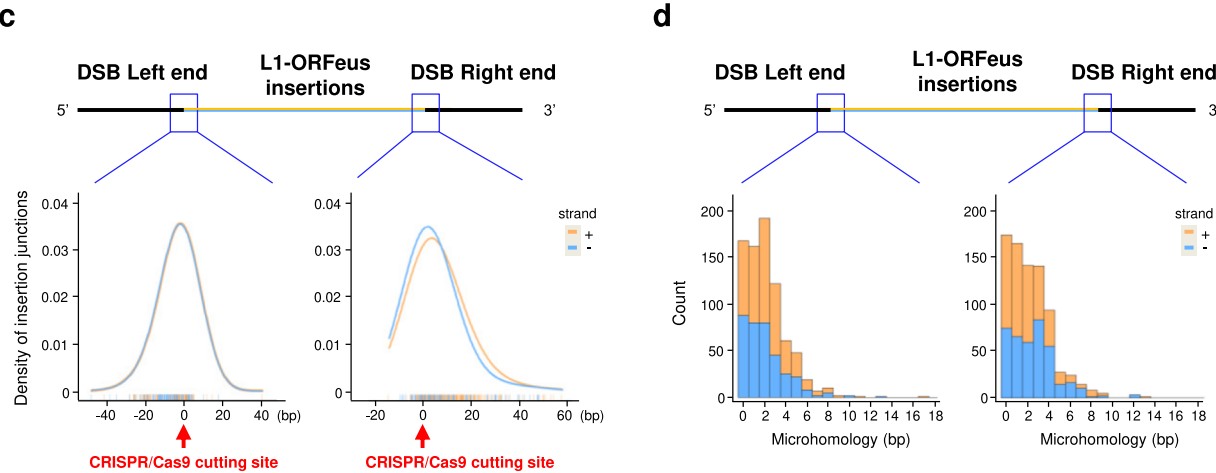

**Fig. 2 Detailed characterization of de novo L1-ORFeus insertions at CRISPR/Cas9 target sites. a** A schematic model of L1-ORFeus insertions at either DNA end of the CRISPR/Cas9-mediated DSB. CRISPR/Cas9 cutting and subsequent end-resection provide a 3'overhang (green) that can be primed by 3' end of L1-ORFeus mRNA (red) for retrotranscription to synthesize the L1 cDNA. The L1 insertions (+) in orange represents the orientation of sequenced L1-ORFeus sequence is from 5' to 3' (**a**, left), while (−) in blue represents the orientation of sequenced L1-ORFeus sequence is from 3' to 5' (**a**, right). **b** L1-ORFeus junction analysis obtained by amplicon sequencing at the *MYC* locus targeted by *MYC* CRISPR/Cas9 in HEK293T cells expressing L1-ORFeus. The sequence alignment was centered on the left end of CRISPR-Cas9-mediated DSBs. Other sources include genomic fragments or plasmid insertions. **c** Density and distribution of L1-ORFeus insertion junctions pooled from MYC (**b**) and RAG1 (Supplementary Fig. 2i). The bar in the bottom represents the position of the insertion junction that join to the left or right end of the CRISPR/Cas9-mediated DSB in HEK293T cells expressing L1-ORFeus. The orientation of L1-ORFeus insertion junction is shown in orange when the fragment is oriented 5' to 3' (+) or blue when 3' to 5' (−). **d** Histogram plots of microhomology lengths in junctions (from **c**) joining L1-ORFeus to the left or right end of the CRISPR/Cas9-mediated DSB in HEK293T cells expressing L1-ORFeus. The orientation of L1-ORFeus insertions is shown in orange when the fragment is oriented 5' to 3' (+) or blue when 3' to 5' (−).

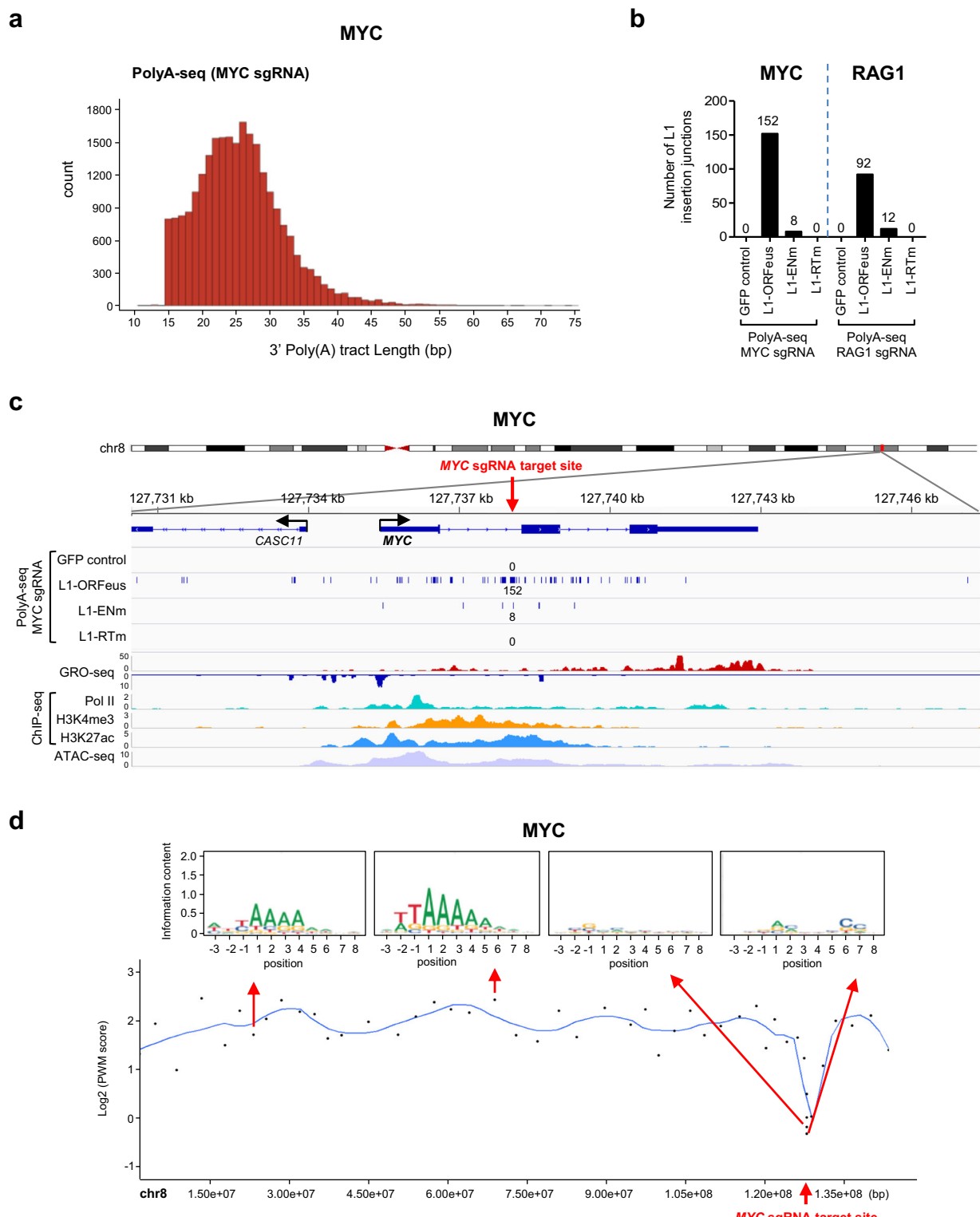

de novo L1 insertions from insertions of endogenous L1 genomic fragments, we sought to reproduce key findings with a more physiologic wild-type L1Hs retrotransposon reporter. For these assays, we used a construct containing a full-length endogenous L1Hs sequence competent of retrotransposition (L1RP)[60]. This L1RP construct uses native L1RP sequences[46], together with the native L1 5′UTR promoter to more closely mimic endogenous

transcription. It also contains a 3′ GFP-artificial intron cassette. The L1RP sequence was further mutated to generate the EN mut (H230A) (L1RP-ENm) or the RT mut (D702Y) (L1RP-RTm) reporters (Supplementary Fig. 8a, Supplementary Sequence 13).

Immunoblot analysis showed an expression of the ORF1p protein consistent across L1RP, L1RP-ENm, and L1RP-RTm reporters, and slightly lower than ORF1p protein expression observed with the

**Fig. 3 EN consensus motif-independent de novo L1-ORFeus insertions at CRISPR/Cas9 target sites. a** Distribution of 3' Poly(A) tract lengths of L1-ORFeus insertions obtained by PolyA-seq in HEK293T cells expressing L1-ORFeus and targeted by *MYC* CRISPR/Cas9. **b** Numbers of L1-ORFeus insertions obtained by PolyA-seq at the *MYC* locus targeted by *MYC* CRISPR/Cas9 (**b**, left) or at the *RAG1* locus targeted by *RAG1* CRISPR/Cas9 (**b**, right) in HEK293T cells expressing L1-ORFeus, L1-ENm, L1-RTm, or GFP control. **c** Detailed view of the distribution of L1-ORFeus insertions at the *MYC* locus obtained by PolyA-seq in HEK293T cells targeted by *MYC* CRISPR/Cas9 expressing L1-ORFeus, L1-ENm, L1-RTm, or GFP control. Each line corresponds to an independent L1-ORFeus insertion; number of L1-ORFeus insertions is pooled from two independent PolyA-seq. Corresponding GRO-seq, ChIP-seq (including Pol II, H3K4me3, and H3K27ac), and ATAC-seq profiles are shown (**c**, bottom). Red arrow shows the *MYC* CRISPR/Cas9 target site. **d** Top: Sequence logo representing the consensus motif detected at L1-ORFeus pre-integration sites in example regions proximal or distant to the *MYC* CRISPR/Cas9 DSB. Each dot represents a chromosomal region encompassing 30 independent L1-ORFeus insertions. Bottom: Spline interpolation curve of consensus motif position-weighted matrix (PWM) score in chromosome 8 obtained by PolyA-seq in HEK293T cells targeted by *MYC* CRISPR/Cas9 that express L1-ORFeus. The mean interval size covered by the chromosomal region encompassing 30 independent L1-ORFeus insertions was 1353.5 bp (range 198–8440 bp) within approx. ±5 kb of the *MYC* CRISPR/Cas9 target site, and 2,824,123 bp (range 801,848–6,764,412 bp) outside of the CRISPR/Cas9 target site.

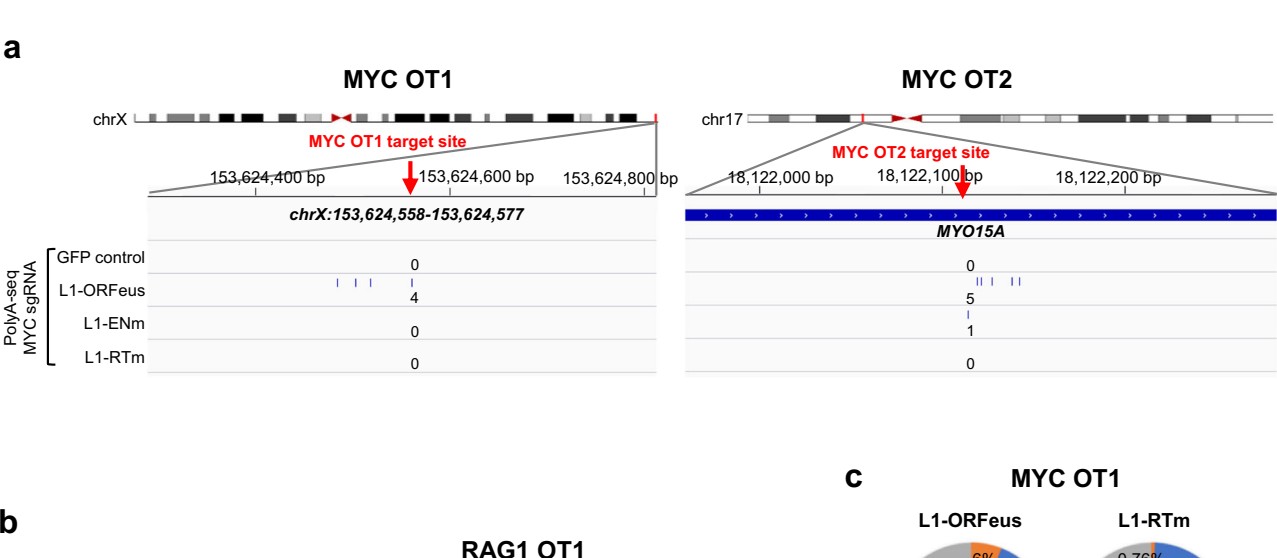

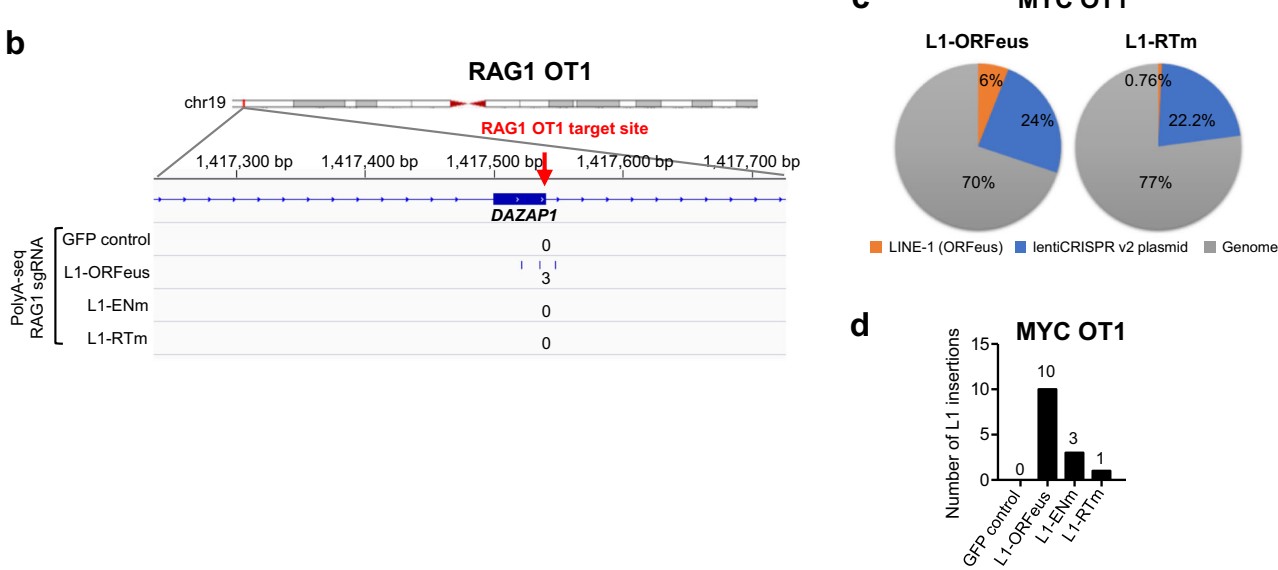

**Fig. 4 RT-dependent de novo L1-ORFeus insertions at CRISPR/Cas9 off-target sites. a** Detailed view of the distribution of L1-ORFeus insertions at two representative *MYC* CRISPR/Cas9 off-targets (OT), MYC OT1 (*ChrX:153,624,558–153,624,577*) (**a**, left) and MYC OT2 (*MYO15A*) (**a**, right), obtained by PolyA-seq in HEK293T cells targeted by *MYC* CRISPR/Cas9 expressing L1-ORFeus, L1-ENm, L1-RTm, or GFP control. **b** Detailed view of the distribution of L1-ORFeus insertions at the *RAG1* CRISPR off-target #1 (RAG1 OT1, *DAZAP1*) obtained by PolyA-seq in HEK293T cells targeted by *RAG1* CRISPR/Cas9 expressing L1-ORFeus, L1-ENm, L1-RTm, or GFP control. **c** Pie charts show the relative abundance of indicated three main types of insertions (i.e., L1-ORFeus, lentiCRISPR v2 plasmid, and genomic fragments) obtained by amplicon sequencing at the MYC OT1 locus in HEK293T cells targeted by *MYC* CRISPR/Cas9 that express L1-ORFeus or L1-RTm. **d** Numbers of L1-ORFeus insertions obtained by amplicon sequencing at the MYC OT1 locus in HEK293T cells targeted by *MYC* CRISPR/Cas9 that express L1-ORFeus, L1-ENm, L1-RTm, or GFP control.

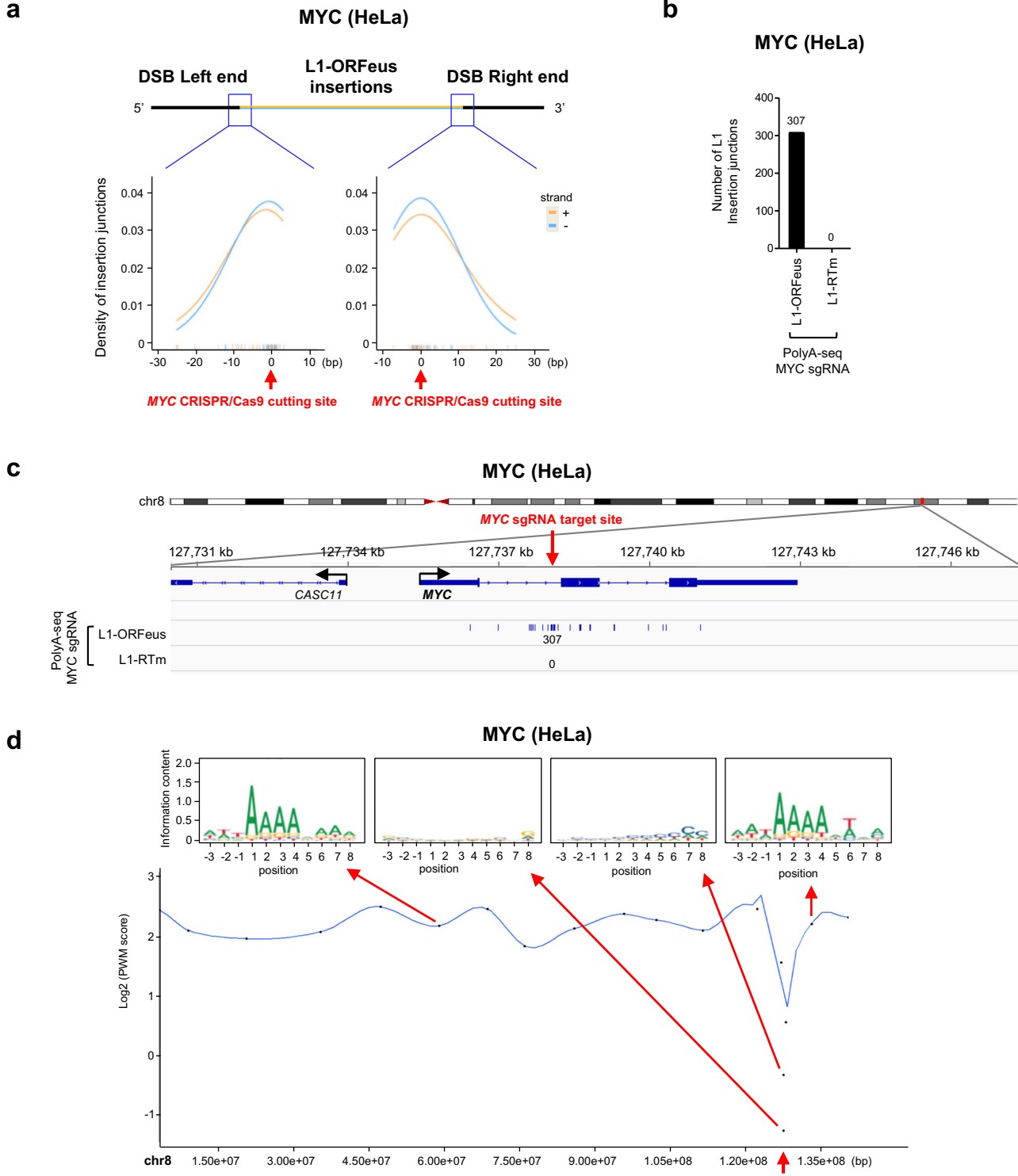

**Fig. 5 RT-dependent de novo L1-ORFeus insertions at CRISPR/Cas9 target site in HeLa cells. a** Detailed view of the distribution of L1-ORFeus insertions at the left or right end of the *MYC* locus obtained by amplicon sequencing at the *MYC* locus targeted by *MYC* CRISPR/Cas9 in HeLa cells expressing L1-ORFeus. **b** Numbers of L1-ORFeus insertions at the *MYC* locus obtained by PolyA-seq targeted by *MYC* CRISPR/Cas9 in HeLa cells that express L1-ORFeus or L1-RTm. **c** Detailed view of the distribution of L1-ORFeus insertions at the *MYC* locus obtained by PolyA-seq in HeLa cells targeted by *MYC* CRISPR/Cas9 expressing L1-ORFeus or L1-RTm. **d** Top: Sequence logo representing the consensus motif detected at L1-ORFeus pre-integration sites in example regions proximal or distant to the *MYC* CRISPR/Cas9 DSB. Each dot represents a chromosomal region encompassing 30 independent L1 insertions. Bottom: Spline interpolation curve of consensus motif position-weighted matrix (PWM) score in chromosome 8 obtained by PolyA-seq in HeLa cells targeted by *MYC* CRISPR/Cas9 that express L1-ORFeus. The mean interval size covered by the chromosomal region encompassing 30 independent L1-ORFeus insertions was: 235 bp (range 28–442 bp) within approx. ±5 kb of the *MYC* CRISPR/Cas9 target site, and 8,025,664 bp (range 863,222–16,390,320 bp) outside of the CRISPR/Cas9 target site.

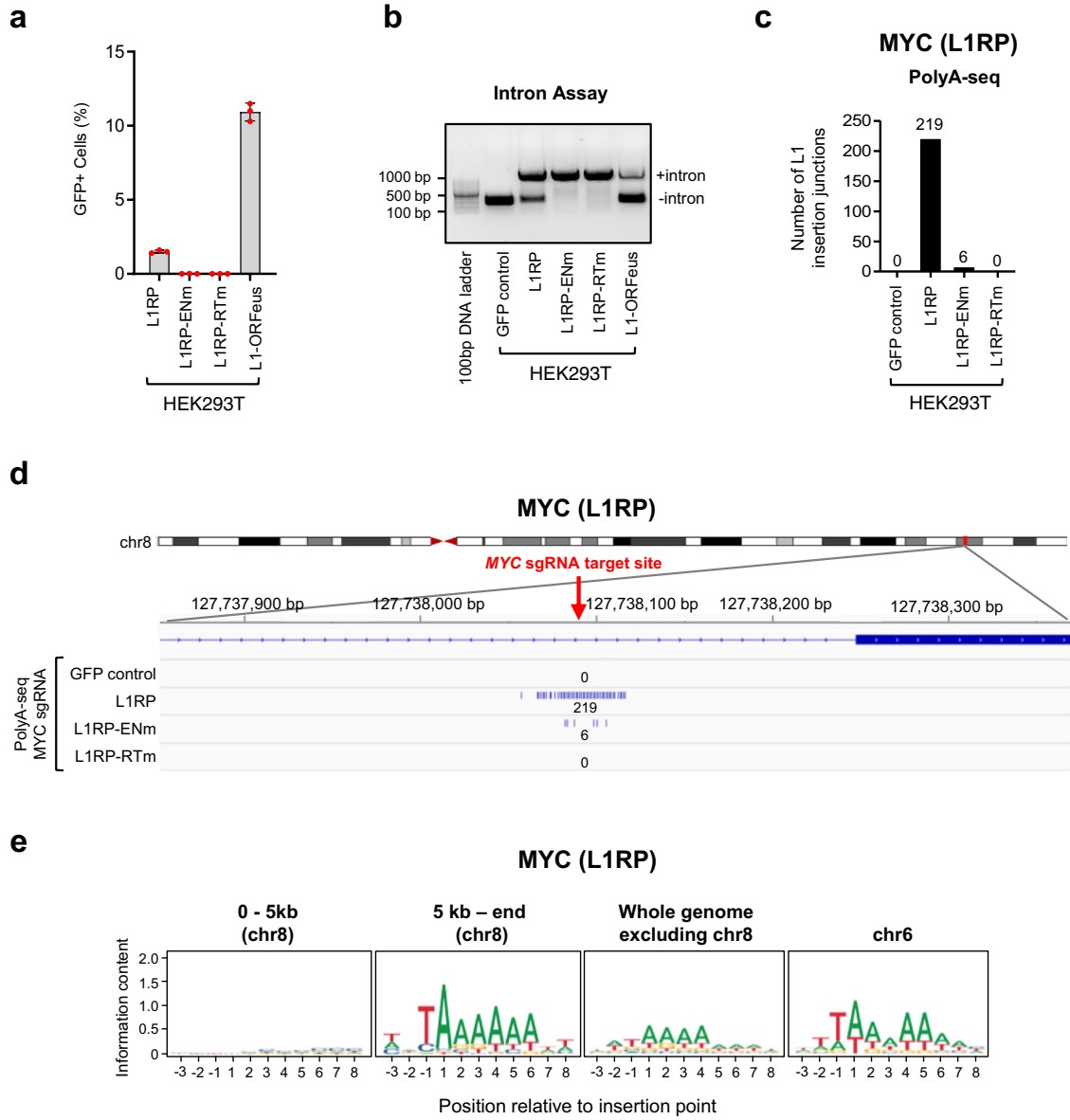

**Fig. 6 RT-dependent de novo L1RP insertions at CRISPR/Cas9 target site in HEK293T cells. a** The percentage of GFP-positive HEK293T cells was analyzed by FACS at day 11 after L1RP transfection and CRISPR/Cas9-mediated DSBs at the *MYC* locus as depicted in Supplementary Fig. 8b. Data are mean ± SD, *n* = 3 independent experiments. **b** PCR assay across the intron of the GFP reporter gene (Intron assay) was performed to probe the L1RP retrotransposition events by genomic DNA-PCR. HEK293T cells transfected with the same amount (1 μg) of L1-ORFeus reporter served as a control. **c** Numbers of L1RP insertions obtained by PolyA-seq at the *MYC* locus targeted by *MYC* CRISPR/Cas9 in HEK293T cells expressing L1RP, L1RP-ENm, L1RP-RTm, or GFP control. **d** Detailed view of the distribution of L1RP insertions obtained by PolyA-seq at the *MYC* locus targeted by *MYC* CRISPR/Cas9 in HEK293T cells expressing L1RP, L1RP-ENm, L1RP-RTm, or GFP control. Each line corresponds to an independent L1RP insertion; number of L1RP insertions is pooled from four independent PolyA-seq experiments. Red arrow shows the *MYC* CRISPR/Cas9 target site. **e** Sequence logos representing the consensus motif detected at L1RP pre-integration sites in the indicated regions proximal or distant to the *MYC* CRISPR/Cas9 DSB in chromosome 8. The canonical consensus motif in chromosome 6 served as a control. Source data are provided as a Source data file.

L1-ORFeus reporter (Supplementary Fig. 8c). The percentage of GFP+ cells was lower with the L1RP reporter compared to L1-ORFeus reporter (Fig. 6a) and the PCR assay across the intron of the GFP reporter showed that the removal of the intron in the L1RP reporter was less efficient than in the L1-ORFeus reporter, and not detected in the L1RP-ENm and L1RP-RTm mutants (Fig. 6b). Overall, these data are consistent with the lower retrotransposition efficiency of a wild-type L1Hs retrotransposon reporter construct compared to the L1-ORFeus reporter.

Next, we repeated key experiments in HEK293T cells with these L1RP reporters (Supplementary Fig. 8b). We detected de novo L1RP retrotransposition events at CRISPR/Cas9 target site by PolyA-seq, indicating that also a more physiologic L1 reporter integrates at CRIPSR/Cas9 genome editing site (Fig. 6c, d). Remarkably, no insertions were observed with the L1RP-RTm reporter and only few insertions with the L1RP-ENm reporter (Fig. 6c). The consensus motif of L1RP pre-integration site was lost within ±5 kb of the CRISPR/Cas9-induced DSB whereas it

was readily identified in insertions distant from the DSB (Fig. 6e). Overall, these data obtained with the L1RP reporter were remarkably similar to those obtained with the L1-ORFeus reporter.

**De novo L1-ORFeus insertions occur less frequently at editing sites targeted by prime editors.** Canonical CRIPSR/Cas9-mediated editing has shown some clinical success[56,61–63], yet novel CRSIPR-based genome editing tools have been developed to expand editing flexibility and enhance safety. Prime editors (PE) enable precise genome editing, including targeted insertions, deletions, all 12 possible base-to-base conversions and their combinations, using Cas9 nickase (H840A) fused to an engineered Moloney murine leukemia virus (M-MLV) reverse transcriptase (RT) that is programmed with a pegRNA to edit DNA sites[64]. PE2 nicks the target site to expose a 3′-hydroxyl group that could be used to prime the reverse transcription of edit-containing pegRNA directly into the target site, while PE3 makes a second nick on the non-edited strand to induce its replacement and increase editing efficiency. Importantly, PE3b was then developed as an improved strategy that nicks the non-edited strand only after edited strand flap resolution to minimize the presence of concurrent nicks and potential DSB in PE3[64] (Fig. 7a). PE do not require DSB intermediates for their editing activity, raising the possibility that L1 insertions might not occur frequently in their editing sites. Since HEK293T cells have been widely used to validate PE efficiency[64,65], we used this cell line to assay for de novo L1-ORFeus insertions at editing sites targeted by various prime editors: PE2, PE3, and PE3b, as well as nick only for PE3 served as a nickase-Cas9 generated single-strand break (SSB) control, or nick only for PE3b served as a negative control since this nicking sgRNA binds only to the edited sequence[64] (Fig. 7a).

First, we examined editing sites at the same position in the *MYC* locus targeted by canonical CRIPSR/Cas9 using two different PE editing approaches (+2-4AAAdel and +5GtoC, Supplementary Figs. 9a–c, 10a). Prime editing efficiency at the targeted sites varied with different PE systems ranging from 4 to 14% with PE2, 10 to 19% with PE3b, and 23 to 31% with PE3 (Supplementary Figs. 9d, 10c). Amplicon sequencing at editing sites targeted by prime editors showed the expected integrations of pegRNA, with the highest frequency in PE3, which are across the whole pegRNA scaffold with various insertion lengths and in the same orientation with reverse transcription along the RT template as described[64,66] (Supplementary Figs. 9e, h, 10e, 11g). Consistently, in PE3, but much less in PE2 or PE3b, we also found hundreds of integrations of the plasmid expressing the prime editing system (Supplementary Figs. 9e, g, 10e, 11e) and genomic fragments flanking the target sites (Supplementary Figs. 9e, i, 10e), although at lower levels than insertions observed in canonical CRISPR/Cas9.

Remarkably, de novo L1-ORFeus insons in the *MYC* locus were observed by PE3 (*MYC* + 2-4AAAdel) or PE3 (*MYC* + 5 GtoC) editing, though were at a much lower level compared to canonical CRISPR/Cas9 (Fig. 7b, c), while PE2 or PE3b editing has rare L1-ORFeus insertions consistent with their reduced plasmid and genomic fragments insertions (Supplementary Figs. 9e, 10e). De novo L1-ORFeus insertion events were decreased with the L1-ENm and virtually absent with the L1-RTm (Fig. 7b, c), consistent with RT-dependent and active L1 retrotransposition process, in contrast to the comparable insertions of pegRNA, plasmid or genomic fragments in cells expressing L1-ORFeus or L1-RTm (Fig. 7e, Supplementary Fig. 11a). Characteristics of retrotransposed sequences (i.e., the 3′ bias and intron splicing of the inserted L1) were similar to the

distribution of de novo L1-ORFeus insertions found at canonical CRISPR/Cas9-mediated DSBs (Fig. 7d, Supplementary Figs. 9f, 11c). Similar L1-ORFeus insertion frequency and characteristics were observed at one additional editing site (*FANCF* + 5GtoT)[64] (Supplementary Figs. 10b, d, f, 11b, d), as well as comparable pegRNA, plasmid, and genomic fragments insertions in cells expressing L1-ORFeus or L1-RTm (Supplementary Figs. 10f, 11b, f, h). Mechanistically, the outcomes of de novo L1-ORFeus insertions both at *MYC* and *FANCF* PE3-editing sites were either L1-ORFeus joined to the pegRNA-edited sequence or to a non-edited or editing site deleted sequence (Supplementary Fig. 12e–g). These L1-ORFeus insertions showed joining to the nicked sites via short microhomologies, mostly 1 to 6 bp with a decreasing trend over length (Supplementary Fig. 12h).

Moreover, PolyA-seq validated de novo L1-ORFeus insertions at PE3 editing sites in a RT-dependent manner, and again were at a lower level compared to canonical CRISPR/Cas9 (Fig. 7f, Supplementary Fig. 12i). Accordingly, de novo L1-ORFeus insertions were undetectable at PE3b editing sites by PolyA-seq (Fig. 7f), demonstrating that PE3b significantly improves the safety of prime editing, consistent with its reduced indels formation by minimizing the presence of DSBs[64]. Overall, we concluded that RT-dependent L1-ORFeus insertions can exploit DNA lesions associated with PE3 editing though were at a lower level compared to canonical CRISPR/Cas9, and are rare during PE2 or PE3b editing, shedding light on the safety concerns of research and possible clinical applications of these prime editors.

**De novo L1-ORFeus insertions are rare during base editors mediated DNA editing.** Two classes of base editors have been developed to install or correct these pathogenic point mutations[67]. In cytosine base editors (CBEs), the APOBEC1 enzyme is fused with a nickase Cas9 (D10A) for targeting via sgRNA and with UGI to increase the accuracy and efficacy of base editing, converting C•G base pair to T•A base pair. In adenine base editors (ABEs), a *E. coli*-derived tRNA adenine deaminase (TadA) is fused with nickase Cas9 to convert an A•T base pair to a G•C base pair. Briefly, in contrast to dCas9-fused BE2, BE3 uses nickase Cas9 to specifically nick the non-edited strand to increase the editing efficiency[68,69]. Fusing BE4 to Gam, a bacteriophage Mu protein that binds DSBs greatly reduces indel formation during base editing[70], while optimization of codon usage and nuclear location sequences (NLS) resulted in the most advanced and efficient AncBE4max or ABE8e[71,72]. Similar to prime editors, BE systems do not require the formation of a DSB intermediate for editing[67]. Thus, we also assayed for de novo L1-ORFeus insertions in multiple editing sites targeted by several CBEs (BE2, BE3, BE4-Gam, and AncBE4max) and one ABE (ABE8e) (Fig. 8a, Supplementary Fig. 13a, b).

In both *MYC* and *FANCF* loci, the base editing efficiency was high in all five BEs, with highest in AncBE4max and ABE8e, and comparable in cells expressing L1-ORFeus or L1-RTm (Supplementary Fig. 13d, e). Amplicon sequencing analysis revealed that de novo L1-ORFeus insertions were rare at the *MYC* site (Fig. 8b) and not significantly frequent at *FANCF* site (Fig. 8c, Supplementary Fig. 13l) edited with all five BE systems, which also validated by PolyA-seq (Fig. 8e, Supplementary Fig. 12j). We could find few plasmid and genomic fragments integrations among these BEs, though they were at much lower levels than in cells edited with canonical CRISPR/Cas9 or PE3 (Supplementary Fig. 13g, h). Notably, BE2 and BE4-Gam showed the lowest insertions (including L1-ORFeus, plasmid or genomic fragments) compared to the other BE systems (Fig. 8b, c, Supplementary Fig. 13g, h), consistent with their lowest rate of indels formation[68,70].

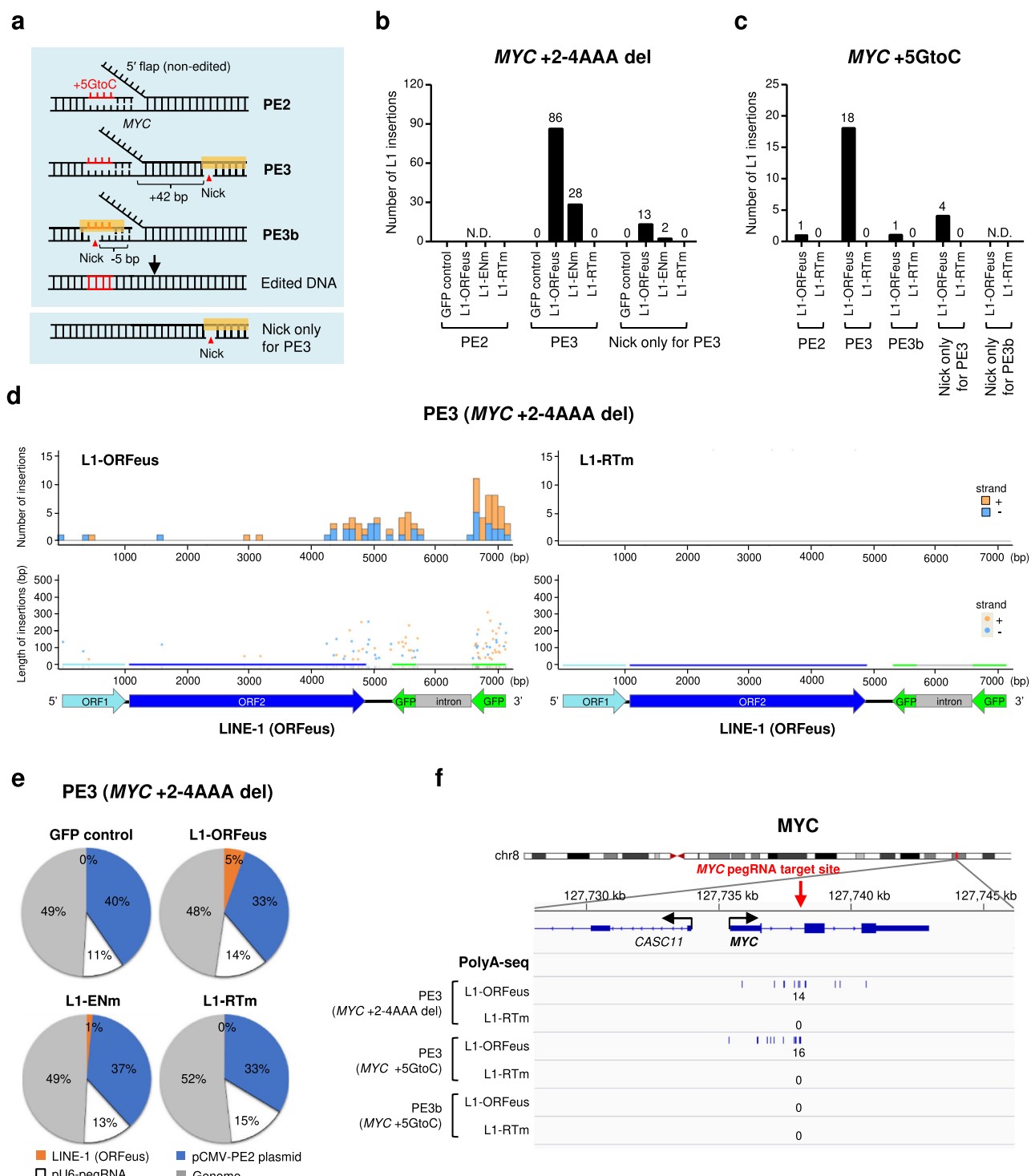

Moreover, we mapped de novo L1-ORFeus insertions in cells undergoing ABE8e editing at two disease-relevant loci (*BCL11A* enhancer and *HBG1/2* promoter) currently being evaluated for gene therapy of sickle cell disease and β-thalassemia[72–74] (Supplementary Fig. 13c, m, n). Not surprisingly, de novo L1-ORFeus insertions were almost undetectable both by amplicon sequencing (Fig. 8d) and PolyA-seq (Fig. 8f, Supplementary Fig. 13k, n) at all editing sites despite a high base editing efficiency (Supplementary Fig. 13f). The few insertions of plasmid and genomic fragments at *BCL11A* enhancer and *HBG1/2* promoter (−198 target or −175 target) editing sites were consistent with their low levels of indels formation as reported[72] (Supplementary

Fig. 13i). Overall, these data suggest that de novo L1-ORFeus insertions are rare with BE systems that avoid DSB formation, supporting their promising efficacy and safety for potential gene therapy even in cell types known to support retrotransposition of endogenous L1.

## Discussion
In this work, we describe LINE-1 retrotransposon insertions into CRISPR/Cas9-mediated DSBs as an outcome of canonical CRISPR/Cas9 genome editing as well as PE3 prime editing. Retrotransposed sequences are thus added to a growing list of insertions known to occur at DSBs generated by CRISPR/Cas9,

**Fig. 7 Safety evaluation of prime editors in HEK293T cells. a** Example diagram of the PE systems editing *MYC* + 5GtoC. While nCas9 (H840A) and pegRNA are required for all prime editing strategies, PE3 contains a nicking sgRNA at +42 bp from the pegRNA-mediated nick to increase the editing efficiency. PE3b contains a sgRNA (−5 bp from the pegRNA-mediated nick) with spacer that match the edited strand to minimize the presence of concurrent nicks. Nick only for PE3 introduces a single-strand break in the non-edited strand by a nicking sgRNA. **b** Numbers of L1-ORFeus insertions obtained by amplicon sequencing at the *MYC* locus targeted by PE2 (*MYC* + 2-4AAA del), PE3 (*MYC* + 2-4AAA del), or nick only for PE3 in HEK293T cells expressing L1-ORFeus, L1-ENm, L1-RTm, or GFP control. Number of L1-ORFeus insertions was pooled from three independent experiments with similar results. **c** Numbers of L1-ORFeus insertions obtained by amplicon sequencing at the *MYC* locus targeted by PE2 (*MYC* + 5GtoC), PE3 (*MYC* + 5GtoC), PE3b (*MYC* + 5GtoC), or nick only for PE3/PE3b in HEK293T cells expressing L1-ORFeus or L1-RTm. **d** Numbers and fragment lengths of L1-ORFeus insertions obtained by amplicon sequencing at the *MYC* locus targeted by PE3 (*MYC* + 2-4AAA del) in HEK293T cells expressing L1-ORFeus or L1-RTm. Example L1-ORFeus insertional sequences in cells expressing L1-ORFeus are shown in Supplementary Sequence 4. Example L1-ORFeus insertions bridging the intron of GFP in cells expressing L1-ORFeus are shown in Supplementary Sequence 5. **e** Pie charts show the relative abundance of four main types of insertions obtained by amplicon sequencing at the *MYC* locus targeted by PE3 (*MYC* + 2-4AAA del) in HEK293T cells expressing L1-ORFeus, L1-ENm, L1-RTm, or GFP control. **f** Detailed view of the distribution of L1-ORFeus insertions at the *MYC* locus obtained by PolyA-seq in HEK293T cells targeted by PE3 (*MYC* + 2-4AAA del), PE3 (*MYC* + 5GtoC), or PE3b (*MYC* + 5GtoC) that express L1-ORFeus or L1-RTm.

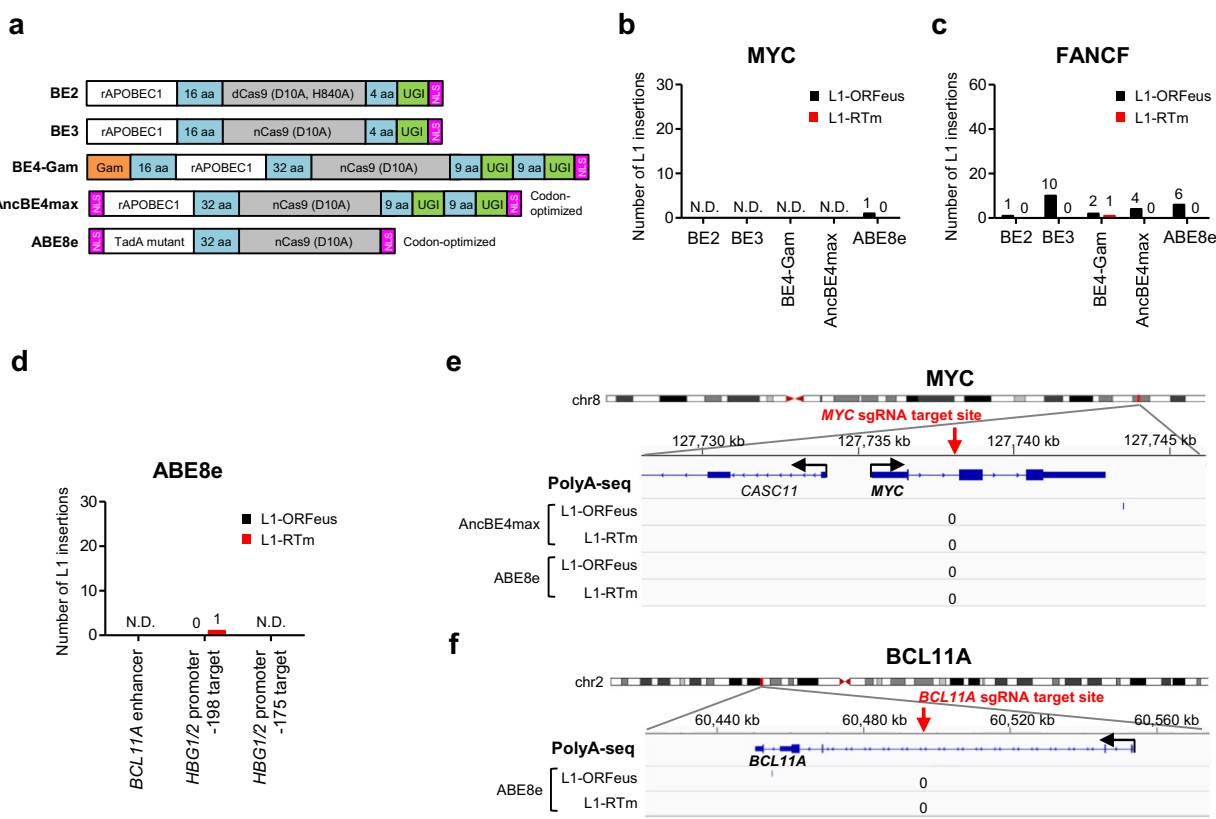

**Fig. 8 De novo L1-ORFeus insertions are rare in base editing. a** Diagram of the base editors. BE2, BE3, BE4-Gam, and AncBE4max are developed as cytosine base editors (CBEs), while ABE8e is an adenine base editor (ABE). **b**, **c** Numbers of L1-ORFeus insertions obtained by amplicon sequencing at the *MYC* (**b**) or *FANCF* (**c**) base editing site targeted by BE2, BE3, BE4-Gam, AncBE4max, or ABE8e in HEK293T cells expressing L1-ORFeus or L1-RTm. **d** Numbers of L1-ORFeus insertions obtained by amplicon sequencing at the *BCL11A* enhancer or *HBG1/2* promoter (−198 bp/−175 bp target) base editing site targeted by ABE8e in HEK293T cells expressing L1-ORFeus or L1-RTm. **e** Detailed view of the distribution of L1-ORFeus insertions at the *MYC* base editing site obtained by PolyA-seq in HEK293T cells targeted by AncBE4max or ABE8e that express L1-ORFeus or L1-RTm. **f** Detailed view of the distribution of L1-ORFeus insertions at the *BCL11A* enhancer base editing site obtained by PolyA-seq in HEK293T cells targeted by ABE8e that express L1-ORFeus or L1-RTm.

including various vectors such as plasmids used for HR-mediated editing[18,19], lentiviral[15], or adeno-associated virus (AAV)[17] vectors encoding CRISPR/Cas9, as well as DNA fragments from freely available DNA[20]. In contrast to these known events that rely on double-stranded DNA (dsDNA) sequences that are inserted during DSB repair, the mechanism of de novo L1 insertions is quite distinct, and appears to rely on the ability of RNA to template break repair in an RT-dependent manner at CRISPR/Cas9-mediated DSBs. Since large insertions are difficult to detect and can be easily missed by PCR and DNA sequencing that normally

used to evaluate the safety and outcomes of CRISPR/Cas9 editing[18,19], we developed PolyA-seq together with improved amplicon sequencing to fully appreciate these L1 insertions. Together, these approaches captured more than 2500 de novo L1 retrotransposition events at CRISPR/Cas9 editing sites, as well as 141,490 insertions of the lentiviral sequences that encodes for the CRISPR/Cas9 and 179,264 integrations of genomic DNA sequences that mostly originated from regions flanking the DSB.

Mechanistically, de novo L1 insertions at sites of genome editing are similar to canonical retrotransposition events in some

ways, but distinct in others. Like canonical retrotransposition, de novo L1 insertions at genome editing sites are dependent on RT activity but they are distinct in that they are largely independent of EN activity and show a random sequence motif pattern instead of the typical preference for TT/AAAA target sites (Figs. 3d, 5d, 6e, Supplementary Figs. 4h and 7h) with junctions that are either direct or mediated by short microhomologies junctions (Fig. 2d, Supplementary Fig. 6i). This suggests that the initiation of reverse transcription depends on a DNA break generated by CRISPR/Cas9 and that its resolution requires ORF2p RT activity like conventional retrotransposition. This EN-independent L1 activity has been previously described at other types of DNA breaks[26,43,45]. De novo L1 insertions were initiated by CRISPR/Cas9-induced DSB formation and extended for about 5 kilobases 5′ and 3′ of the DSB, consistent with extensive resection of the DSB[6–9] before L1 integration. In contrast, canonical retrotransposition events, which were EN-dependent and showed TT/AAAA insertion site preference predominated in more distant regions from the CRISPR/Cas9-mediated DSBs, including on the same chromosome or in the rest of the genome.

Of note, plasmids are known to integrate into DSBs initiated by CRISPR/Cas9 cutting[9,18,19]. Although we induced CRISPR/Cas9 DSBs 5 days after L1 plasmid transfection, still few copies of plasmid are present in cells at the time of CRISPR/Cas9 cutting. Consistently, the rare sequences found in GFP control samples aligned to the GFP portion of the vector and are likely plasmid integrations (Fig. 1e, Supplementary Fig. 2f). The few insertions mapping to the intron of the GFP cassette observed in L1-ORFeus and L1-ENm, but not in the GFP control and L1-RTm (Fig. 1e, Supplementary Fig. 2f) might be also plasmid insertions or due to an inefficient intron removal during the pre-mRNA splicing, that can occur in experiments conducted with a transfected plasmid reporter[75].

Unwanted outcomes of DSB repair after canonical CRISPR/Cas9 targeting are a potential threat even if occurring in a small fraction of the edited cell population. These unwanted outcomes can occur both at the on-target or off-target sites of CRISPR/Cas9 editing. Notably, PolyA-seq allows for genome-wide surveys of de novo L1 integrations, and not only identified L1 insertions at on-target but also at multiple off-targets generated by CRISPR/Cas9 (Fig. 4, Supplementary Fig. 5). Thus, detection of recurrent de novo L1 insertion sites could represent an additional method to detect off-target activities of various nuclease-based genome editing tools, in parallel to the existing GUIDE-seq[20], SITE-seq[76], LAM-HTGTS[10], PEM-seq[9], IDLV assay[16], CHANGE-seq[49], CIRCLE-seq[77], and DISCOVER-seq[78].

Increasing the safety of CRISPR/Cas9 editing is essential for gene therapy[79]. To this end, several CRISPR/Cas9-based genome editing tools that do not require the formation of DSB intermediates or donor DNA have been recently developed, including PE and BE systems. PE2 and PE3b systems as well as CBE and ABE systems rarely allowed de novo L1 insertions, similar to the low genomic or plasmid integrations at the editing site, consistent with their low propensity to permit indels[64,80]. We also compared canonical CRISPR/Cas9 with nickase-Cas9 using nick only for PE3 (Fig. 7a). In nick only for PE3, when SSBs are generated by nickase-Cas9 only rare L1-ORFeus, plasmid or genomic insertions were detected (Fig. 7b, c, Supplementary Figs. 9e and 10e, f). These results demonstrated that L1 insertions are much more frequent in DSBs than in SSBs generated by Cas9. This could also explain why L1-ORFeus insertions are lower in PE2 and BE systems that tend to generate SSBs rather than DSBs. In contrast, we found recurrent de novo L1-ORFeus insertions at the target sites using the PE3 editing system (Supplementary Fig. 12e–g) though were at a lower level compared to canonical CRISPR/Cas9. Interestingly, sequences with

de novo L1-ORFeus insertions in the PE3 system from amplicon sequencing showed an increased editing efficiency compared to the bulk sequences (Supplementary Fig. 12a–d). A similar increase of editing efficiency was also observed in sequences with inserted PE plasmid, pegRNA, or genomic fragments (Supplementary Fig. 12a–c).

While our experimental models are based on ectopically-expressed L1-ORFeus or L1RP reporters to show de novo L1 insertions, insertions generated by active endogenous L1 sequences are expected to be lower, and may vary according to the expression level and retrotransposition activity of the endogenous L1 in the cell type that is being edited. L1 retrotransposition is commonly suppressed in human somatic cells[34], but it appears to be reactivated not only in human cancers[81–83], but also in human embryonic stem cells (hESCs) and induced pluripotent stem cells (iPSCs)[84] that are frequently used for CRISPR/Cas9 editing[12,56,85,86]. Although HEK293T express lower L1 ORF1p than iPSCs (Supplementary Fig. 1g), suggestive of a low endogenous activity, we analyzed whether endogenous L1 insertions could be detected at CRISPR/Cas9-induced DSBs by aligning to L1Hs sequences. Currently, *Homo sapiens*-specific L1 (L1Hs) is the most active autonomous L1 element in the genome[34,35,87]. We detected L1Hs sequences with bias toward the 3′ of the L1Hs both in MYC and RAG1 editing sites (Supplementary Fig. 14a–c, Supplementary Sequence 12). However, it is difficult to conclude whether these L1Hs sequences represent retrotransposition events of active endogenous L1Hs retrotransposon occurring at CRISPR/Cas9-mediated DSBs or rather genomic insertions of non-active L1 sequences.

Overall, our results show that retrotransposition of L1 elements is a frequent outcome of canonical CRISPR/Cas9 editing. Although these findings will need to be further confirmed in more physiological or clinical settings, they should be carefully evaluated and searched for when editing is intended for a therapeutic approach given the potential pathogenetic effect of L1 insertions[34].

## Methods

**Plasmids, oligonucleotides, and cloning.** The human codon-optimized, synthetic L1-ORFeus was modified from the human codon-optimized, synthetic L1-ORFeus constructs published by the Boeke Lab[46]. The detailed information of pCEP4 GFP control, pCEP4 L1-ORFeus, pCEP4 L1-ORFeus-ENm (H230A) (L1-ENm), and pCEP4 L1-ORFeus-RTm (D702Y) (L1-RTm) can be found in Supplementary Sequence 1. Briefly, CMV promoter controls expression of the L1-ORFeus reporter cassette in the pCEP4-derived constructs. The L1-ORFeus mutant constructs contain the following modifications: the EN mutant construct contains the H230A residue in ORF2p (L1-ENm) and the RT mutant construct contains the D702Y residue in ORF2p (L1-RTm), which are disrupted residues critical for retrotransposition, as previously shown[38,88]. The GFP control vector was generated by cloning an intron-less GFP reporter in the same pCEP4 vector backbone. pCEP4 L1RP (L1RP) is modified from the Boeke Lab[46] and published previously[60] (Addgene plasmid #131392). pCEP4 L1RP uses native L1RP sequences[89] and the native L1 5′UTR promoter, sharing the same pCEP4 vector backbone and GFP-Artificial Intron cassette with pCEP4 L1-ORFeus vector; likewise, two mutants pCEP4 L1RP-ENm (H230A) and pCEP4 L1RP-RTm (D702Y) were generated (see Supplementary Fig. 8a and Supplementary Sequence 13). lentiCRISPR v2 (Addgene plasmid #52961) was a gift from Feng Zhang. Plasmids expressing sgRNAs were constructed by ligation of annealed oligonucleotides into BsmBI-digested Lenti-CRISPR v2. pCMV-PE2 (Addgene plasmid #132775) and pU6-pegRNA-GG-acceptor (Addgene plasmid #132777) were gifts from David Liu. Plasmids expressing pegRNAs or nicking sgRNAs (for PE3 and PE3b) were constructed by Golden Gate assembly using BsaI-digested pU6-pegRNA-GG-acceptor vector as reported previously[64]. pCMV-BE2 (Addgene plasmid #73020), pCMV-BE3 (Addgene plasmid #73021), pCMV-BE4-Gam (Addgene plasmid #100806), pCMV-AncBE4max (Addgene plasmid #112094), and pCMV-ABE8e (Addgene plasmid #138489) were gifts from David Liu. Oligonucleotides were purchased from Integrated DNA Technology (IDT), and sequences of sgRNAs and pegRNAs used in this work were listed in Supplementary Table 1. CRISPR off-targets (OT) were identified as previously reported[10,59], sequences were listed in Supplementary Table 2. All constructs were sequence-verified and purified by NucleoBond Plasmid DNA purification PC100/500 (Takara).

**Cell culture**. HEK293T, HeLa, and U2OS cells were maintained in DMEM supplemented with 10% fetal bovine serum (FBS), penicillin-streptomycin (100 units per ml), and L-glutamine (2 mM). Cells were cultured at 37 °C with 5% $CO_2$ and tested negative for mycoplasma. HEK293T and HeLa cells were authenticated as they were purchased from ATCC; U2OS cells were provided by Jeremy Stark, and validated by STR profiling[90].

**Probing LINE-1 retrotransposition events by flow cytometry and intron assay**. The experimental settings and timing were almost the same for canonical CRISPR/Cas9 editing, Prime editing, and Base editing with both L1-ORFeus and L1RP transfection. $8 \times 10^5$ HEK293T cells, $2 \times 10^5$ HeLa/U2OS cells were seeded on 6-well plates. The following day, for L1-ORFeus in HEK293T/HeLa cells, cells were transfected at ~60–70% confluency with 1.5 μL X-fect polymer (Takara; Cat. #631318) and 5 μg indicated pCEP4 L1-ORFeus plasmids, as well as pCEP4 GFP control vector according to the manufacturer's protocols. For L1-ORFeus in U2OS cells and L1RP in HEK293T cells, cells were transfected at ~60–70% confluency by Fugene HD (Promega; Cat. #E2311) with 1 μg indicated plasmids following the manufacturer's instructions. The media was changed 8–12 h after transfection. Two days post-transfection, medium was supplemented with puromycin (Sigma) at 1 μg/mL (for HEK293T/U2OS cells) or 2 μg/mL (for HeLa cells) to enrich the transfection positive cells (thus to enrich L1 retrotransposition events). The media was changed every 2 days. Cells were cultured for indicated days post-transfection, followed by flow cytometry and intron assay analysis on the bulk population of cells without GFP sorting. For flow cytometry, data acquisition was performed using a FACSVerse flow cytometer (BD Biosciences). For intron assay, genomic DNA was extracted using rapid lysis buffer containing 20 μg/ml Proteinase K and incubation at 56 °C overnight, followed by standard isopropanol extraction, wash in 70% ethanol, and resuspension in TE buffer. 200 ng genomic DNA, Taq DNA polymerase (Qiagen), as well as two primers were put in the PCR reactions (94 °C for 3 min, then 35 cycles of [94 °C for 30 s, 59 °C for 30 s, and 72 °C for 1 min 25 s], followed by a final 72 °C extension for 5 min). Following PCR, the products were separated on 1% agarose gel. The original PCR product (with intron) was 1192 bp, while LINE-1 retrotransposition generate intron-removed DNA which gives rise to a short PCR product (292 bp) (primer sequences are listed in Supplementary Table 3).

**RT-PCR**. RT-PCR was performed as previously described[91,92]. In brief, total RNA was extracted from HEK293T cells by Trizol (Fisher Scientific) and reversed-transcribed with iScript cDNA synthesis kit (Bio-Rad) following the manufacturer's instructions. Indicated gene induction was analyzed by PCR for 25–28 cycles at 94 °C for 30 s, 57 °C for 30 s, and 72 °C for 30 s. Primers are listed in Supplementary Table 3.

**Protein extraction and western blot analysis**. Cells were lysed in RIPA buffer with protease inhibitors (Roche), 1 mM phenylmethanesulfonylfluoride (PMSF), 10 mM NaF and 1 mM $Na_3VO_4$ by sonication. Extracts were centrifuged at 14,000 r.p.m ($18,407 \times g$) for 15 min. The supernatants were collected and assayed for protein concentration using the Bio-Rad protein assay method. 20 μg of proteins were loaded on 4–12% Criterion™ XT Bis-Tris Protein Gel (Biorad), transferred by Trans-Blot-Turbo (Biorad), and blocked with EveryBlot Blocking Buffer (Biorad). Primary antibodies were incubated with membranes overnight at 4 °C. Antibody against ORF1p (Sigma, Cat. # MABC1152; 1:10,000) was used and validated as previously published[60], anti-β-actin is commercially available (Sigma, Cat. #A5316; 1:10,000). Membranes were developed with ECL solution (GE Healthcare) and imaged by Image Lab. The relative expression of the ORF1p protein was measured by densitometry using Image J and normalized for the β-actin intensity of the corresponding lane.

**Statistical analysis**. We performed statistical analysis by using an unpaired Student's *t*-test (GraphPad Prism 9.0) for all studies unless otherwise indicated. We considered $P < 0.05$ to be statistically significant as previously described[93,94].

**Production and transduction with CRISPR/Cas9 lentiviruses**. The production and transduction with CRISPR/Cas9 lentiviruses were performed as previously described[59]. In brief, HEK293FT cells were maintained in 10% FBS-containing DMEM. To generate lentiviral particles, $5 \times 10^6$ HEK293FT cells were plated per 10 cm dish. The following day, cells were transfected by calcium phosphate transfection method with 7.2 μg of indicated lentiCRISPR v2 plasmid, 3.6 μg of VSVG, 3.6 μg of RSV-REV, and 3.6 μg of PMDLg/pPRE. The media was changed 8 h after transfection. The viral supernatant (10 mL/10 cm dish) was collected 36 h after transfection, passed through a 0.45-μm filter, pooled, and used either fresh or snap-frozen. For lentivirus infection, 2 mL of viral supernatant with polybrene (6 μg/ml) was added to one well of 6-well plates of cells with at ~60–70% confluency. The viral supernatant was exchanged for fresh medium 8 h later. After 48 h, cells were selected with 1–2 μg/mL (for HEK293T/U2OS cells) or 2–3 μg/mL (for HeLa cells) puromycin to enrich CRISPR/Cas9 lentiviruses infected cells.

**Generation of amplicon sequencing libraries**. Amplicon sequencing is a method of targeted next-generation sequencing. Phusion High Fidelity DNA polymerase (Thermo Fisher Scientific) was used to amplify selected target sites from 5 μg template genomic DNA within 5 PCR reactions as follows: 98 °C for 3 min, then 35 cycles of [98 °C for 30 s, 58–60 °C for 30 s, and 72 °C for 1 min], followed by a final 72 °C extension for 5 min. Multiple reactions were pooled and size fractionated for DNA fragments between 300 and 1000 base pairs (exclude the strong main band) on 1% agarose gel. DNA were purified using Gel extraction kit (Qiagen) following the manufacturer's protocol and sequenced bi-directionally 250 bp (PE250) in a Miseq sequencing platform at the Molecular Biology Core Facilities of the Dana-Farber Cancer Institute. Primers used for amplicon sequencing are listed in Supplementary Table 4. At least three independent samples were generated and analyzed for each experimental condition. The detailed information of Amplicon sequencing libraries is listed in Supplementary Data 1.

**Amplicon sequencing data analysis**. Sequences in library were processed as previously described[59]. Briefly, reads for each experimental condition were demultiplexed by designed forward and reverse barcodes (followed portion of primer totally plus barcode up to 8 bp was appended in case barcode length is <8 bp). To enhance specificity of target sequences, reads were further filtered by the presence of primer plus additional 10 downstream bases.

**Insertion sources analysis**. For focusing on insertion portion of PCR fragments obtained by amplicon sequencing, the barcode, primer, and target reference portion of the sequences were trimmed. Based on trimmed sequences, the exact duplicates treated as PCR repeats and sequences with length <30 bp were further eliminated. To analyze insertion sources, we next employed BOWTIE2 v2.1 to align the processed sequences to different references like human genome (GRCh38/hg38), exogenous LINE1 (ORFeus), and indicated plasmids, respectively. Because of high similarity among younger L1 members of L1PA subfamily including youngest active L1Hs[35], L1RP(L1PA1), and relatively younger inactive L1PA2, L1PA3,…, L1PA8, to identify endogenous L1 insertions we particularly mapped sequences against the youngest reference elements annotated as L1Hs in Repeat-Masker Repbase[95] (http://www.repeatmasker.org). As for LINE1 (ORFeus) alignment, we specially excluded the alignments where contain the common sequences with lentiCRISPR v2 plasmid and attributed those reads to lentiCRISPR v2 plasmid; excluded the alignments where contain the common sequences with endogenous L1Hs and attributed those reads to L1Hs reference. As for plasmids alignment, the indicated plasmid was chosen for corresponding experiment, i.e., lentiCRISPR v2 for canonical CRISPR/Cas9 editing analysis, pCMV-PE2 and pU6-pegRNA for PE editing analysis, pCMV-BE2 for BE2 editing analysis, pCMV-BE3 for BE3 editing analysis, pCMV-BE4-Gam for BE4-Gam editing analysis, pCMV-AncBE4max for AncBE4max editing analysis, pCMV-ABE8e for ABE8e editing analysis (see Supplementary Sequence 3). The aligned sequences were further cleaned by removing duplicates by using PICARD MarkDuplicates tool and those with low mapping quality (mapq < 2). Next, we used BEDTOOLS v2.29 bamtobed to convert paired-end sequence alignments in BAM format into BEDPE records, and further excluded redundant sequences having same reference alignment position (<4 bp difference for both reference start and end) to obtain the final inserted fragments and their alignments from each reference. Insertion dot and fragment maps were plotted by using R ggplot to depict the distribution of insertion sources in reference.

**L1-ORFeus insertion junction analysis**. To understand how L1-ORFeus mechanically joins to break point of CRISPR/Cas9-mediated DSBs to complete its insertion, we analyzed the position distribution and microhomology of insertion junction. First, the raw paired-end sequences that contain L1-ORFeus insertions identified by above insertion sources analysis obtained by amplicon sequencing were particularly selected and aligned to amplicon target reference sequence and LINE1 (ORFeus) by using BLAST v2.11, respectively. The forward sequences alignments records against target reference and LINE1 (ORFeus) were used to determine the left end of DSB and insertion junction, the overlapped base pairs between break end and insertion junction were calculated as microhomology, while the gap between them was considered as "other sources" of insertions (include genomic fragments or plasmid insertions). Likewise, the reverse alignments records were calculated for right end of DSB and insertion junction. Both forward and reverse sequences alignment against LINE1 (ORFeus) were analyzed to determine the L1 insertion fragments between two ends of break. Finally, a map of L1-ORFeus insertions at break point and insertion junction position distribution surrounding CRISPR/Cas9-mediated cutting site (three base pairs before PAM) were plotted by using R ggplot to illustrate the mechanisms of insertions at DSBs. BLASTN parameters '-reward' and '-penalty' were set as 1 and −2 for more conserved result. Only alignments record with alignment start in PCR target reference no more than 8 bp from 5′ end for forward and 3′ end for reverse sequence were considered as valid portion of PCR target reference. Furthermore, sequences with PCR target reference alignment length <18 bp or LINE1 (ORFeus) alignment length <30 bp were further eliminated as well. For PE experiments, the raw sequences from pegRNA nick strand were also aligned to pU6-pegRNA-GG-acceptor plasmid to identify pegRNA portion required to have edited base, minus alignment orientation

and at least 10 bp alignment length. Then by taking into account the LINE1 (ORFeus) alignment records, pegRNA portion's junction and microhomology with L1 were finally calculated.

**Generation of PolyA-seq libraries.** PolyA-seq was developed based on HTGTS[59] and a similar L1 retrotransposition capturing technique[43]. Briefly, 50 μg genomic DNA was digested overnight with HaeIII enzyme (recognizes and cuts at 5′-GG/CC-3′), followed by 3′A addition by Klenow polymerase (3′–5′ exo-; New England Biolabs) and adapter ligation (composed of an upper linker and a lower 3′-modified linker) by T4 DNA Ligase (Promega). To reduce sequencing background, ligation reactions were digested with Xba I to remove the exogenous LINE-1 sequence from plasmid (first blocking). In the first round of PCR, DNA was amplified using Biotin-LEAP, together with adapter-specific reverse primer (AP1) and Phusion polymerase (Thermo Fisher Scientific). 20 PCR cycles were performed in the following conditions: 98 °C for 20 s, 56 °C for 30 s, and 72 °C for 1 min. Multiple reactions were performed in generating large-scale libraries. Biotinylated PCR products were captured by Dynabeads MyOne StreptavidinC1 kit (Invitrogen) for 2–3 h at room temperature with rotation, followed by 2 h digestion with Xba I (second blocking). PCR products were then eluted in 95% formamide/10 mM EDTA and purified by PCR extraction kit (Qiagen). The second round emulsion-PCR (EM-PCR) was performed in an oil-surfactant mixture by nested forward primer (SV40-polyA-F) and nested adapter-specific reverse primer (AP2) using the following conditions: 20 cycles of 94 °C for 30 s, 57 °C for 30 s, and 72 °C for 1 min. The PCR products were then extracted three times with diethyl ether and DNA was re-purified by PCR extraction kit (Qiagen). The third round of PCR was performed with barcoded forward and reverse primer (10–12 cycles of 98 °C for 20 s, 60 °C for 30 s, and 72 °C for 1 min). PCR products were size fractionated for DNA fragments between 300 and 1000 bp on a 1% agarose gel and purified by Gel extraction kit (Qiagen).

Twelve to eighteen samples were pooled and sequenced through Illumina MiSeq (PE250) by the Molecular Biology Core Facilities of the Dana-Farber Cancer Institute. Oligonucleotides and primers used for PolyA-seq are listed in Supplementary Table 5. At least 2–6 independent libraries were generated and analyzed for each experimental condition. The detailed information of PolyA-seq libraries is listed in Supplementary Data 2.

**PolyA-seq data processing and alignment.** Sequences in library were similarly processed as previously described[59]. Briefly, reads for each experimental condition were demultiplexed by designed barcodes (followed portion of primer totally plus barcode up to 8 bp was appended in case barcode length is <8bp). Then reads were further filtered by the presence of primer (7 bp Poly(A) was imposed to primer directly) plus additional 8 bp Poly(A), and the barcode and primer portion of the remained sequences were masked for alignment analysis. Next, the processed sequences were aligned to human genome (GRCh38/hg38) using BLAT, and finally aligned sequences were further cleaned by removing PCR repeats (reads with same junction position in alignment to the reference genome and a start position in the read <3 bp apart), invalid alignments (including alignment scores <30, reads with multiple alignments having a score difference <4 and alignments having 10-nucleotide gaps) and ligation artifacts (for example, random ligation with HaeIII restriction sites). L1 insertion junction position was determined based on the genomic position of the 5′ end of the aligned read.

**Consensus motif analysis at L1 insertion sites.** The genomic DNA sequence flanking L1 insertion positions −3 (upstream) and +8 (downstream) was retrieved to analyze the consensus sequence motif at the L1 insertion sites obtained by PolyA-seq. R package 'Biostrings' and 'ggseqlogo' were performed to generate sequence logos. The corresponding position-weighted matrix (PWM) was calculated by R package 'seqLogo'. The motif score is defined as the sum of information content at each position.

**L1 3′ Poly(A) tract analysis.** To identify the 3′ Poly(A) tract additionally added to L1 intact 3′ end, we generated 200 bp Poly(A) reference sequence and BLASTN all reads with L1 insertion to such Poly(A) reference, restricting query start in read mapped to Poly(A) reference less than query start in read to Human genome to ensure that the identified Poly(A) was inserted and followed by genomic sequence at insertion junction. The default parameter 'word_size' of BLAST is 11, so Poly(A) size <11 was ignored accordingly. The above approach was applied to PolyA-seq.

**Editing efficiency analysis.** Nucleotide mutations were calculated from raw sequences as previously described[59,96]. Briefly, sequences with mean quality score <20 and length <50 were excluded. The remained sequences were aligned to target reference sequence using BLASTN. Only sequences with alignment length >80 were considered for calculation of editing efficiency (mutation rate). The real mutations were determined by passing a Neighborhood Quality Standard (NQS) filter criterion that requires a minimum Phred score of 30 for the mutation itself, and 20 for the five adjacent bases on either side. Finally, the percentage of reads that mutated at target base site represents editing efficiency.

**Data reporting.** No statistical methods were used to predetermine sample size.

**Reporting summary.** Further information on research design is available in the Nature Research Reporting Summary linked to this article.

## Data availability

The source data used to generate figures in this study are provided in the Supplementary Information/Source data file. Information for amplicon sequencing libraries can be found online in Supplementary Data 1: Information of Amplicon sequencing libraries; information for PolyA-seq libraries can be found online in Supplementary Data 2: Information of PolyA-seq libraries. All sequencing data have been deposited in the Gene Expression Omnibus database under accession number GSE178440. For GRO-seq analysis, publicly available HEK293T GRO-seq data were acquired from NCBI Gene Expression Omnibus (GEO) database (accession: GSM1249869). The downloaded hg18-based bigwig files were converted to hg38 by using CrossMap v0.5. For ChIP-seq and ATAC-seq analysis, publicly available HEK293T ChIP-seq and ATAC-seq data were acquired from GEO, and the accession numbers are Pol II: GSM1249891, H3K4me3: GSM1249885, H3K27ac: GSM1249889, ATAC-seq: GSM3271043. Raw reads are aligned against to Human genome (hg38) using BWA. Aligned BAM files are filtered by removing low quality and unpaired reads, and de-duplicated. Source data are provided with this paper.

## Code availability

Source code for genomic event analysis tools (GEAT) developed in our laboratory to perform the analysis is available at https://github.com/geatools/geat[97].

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

## Acknowledgements

We thank John V. Moran for critical reading the manuscript; William Pu and Maksymilian Prondzynski for providing iPSC (WTC-11) protein extract, Thorsten Schlaeger and Yang Tang for providing hESC protein extract. We thank Boxun Zhao, Daniel Ardeljan, and Jared P. Steranka for helpful discussion. This work was supported by 1R01-CA222598 to R.C.

## Author contributions

J.T. and R.C. conceived the project, designed experiments, and wrote the manuscript with the help of Q.W.; J.T. performed all of the experiments; Q.W. designed and performed the bioinformatics analysis and tools development; C.M.-D. and K.B. provided key reagents, contributed to design and interpretation of the experiments, and contribute to writing the manuscript; J.T., Q.W., and R.C. analyzed and interpreted the data, designed the figures; R.C. supervised the study.

## Competing interests

The authors declare no competing interests.
