## [Peer Review File · Nature Communications]

Reviewers' Comments:

Reviewer #1:

Remarks to the Author:

Manuscript: Tao et al. (NCOMMS-21-38175)

In the manuscript entitled 'Frequency and mechanisms of LINE-1 retrotransposon insertions at CRISPR/Cas9 sites' by Tao et al. (M NCOMMS-21-38175), the authors addressed the question if and to what extent L1 de novo insertions occur into DSBs after canonical CRISPR/Cas9-editing and as a consequence of prime editing applications. This is an important question because the identification of DNA modifications occurring at CRISPR/Cas9-target and off-target sites is critical to determine efficiency and safety of genome editing tools. The authors report two newly developed sequencing techniques termed PolyA-Seq and Non-PolyA-Seq which they used to identify mouse L1-ORFeus de novo insertions that occurred into canonical CRISPR/Cas9 editing sites in human HEK293 and HeLa cells, and insertions that occurred after introduction of nucleotide changes by cytidine or adenine base editors or by prime editing. Presented data suggest that L1 RT-dependent L1 insertions can exploit DNA lesions associated with PE3 editing although they occurred less frequently compared to L1 insertions that occurred into CRISPR/Cas9-caused DSBs, and are largely excluded during PE2 or PE3b editing.

The finding that L1 insertions at DSBs are not depending on the presence of an EN consensus motif is not new and has been reported earlier. It also would have been expected that L1 retrotransposition events could retrotranspose into DSBs cause by CRISPR/Cas 9 because it has been reported earlier that L1 and Alu retrotransposons can integrate into preexisting DSBs generated by other mechanisms. However, the presented data demonstrate that retrotransposon insertions can be a potential outcome of CRISPR/Cas9 genome editing (if functional L1s are expressed in the respective cell line), and add to the characterization of safety and efficiency of the different editing tools. This manuscript is highly relevant to evaluate the risks for safety and efficiency of genome editing technologies for gene- and/ or cell therapeutic approaches. Although the authors present an overwhelming amount of data, I would exclude this manuscript from being published in the journal Nature Communication in its current form. There are major points that have to be addressed before the manuscript can be considered for publication.

Major Criticism:

1. It is misleading when the L1-ORFeus reporter element is termed 'L1-WT' because this reporter element is a synthetic, codon-optimized mouse L1 element which the authors transfected into human cells. The mouse L1-ORFeus RNA sequence is completely different from the human L1Hs wildtype RNA and retrotransposition levels of L1-ORFeus is more than 200-fold increased relative to wildtype L1 elements from mouse. Therefore, the authors have to choose another designation for the L1-ORFeus reporter whose EN and RT was not inactivated by mutations in EN or RT domain. My suggestion would be to use the designation 'L1-ORFeus' consistently throughout the manuscript. The lettering of all figures in the manuscript has to be changed accordingly. It is questionable to what extent data obtained from this mouse L1-ORFeus element can be extrapolated to functional endogenous L1Hs elements in human cells. The question arises if it might have been more appropriate to use the human codon-optimized, synthetic L1-ORFeus element for these studies which also has been generated and published by the Boeke Lab several years ago.

2. page 5/lines 97/Suppl.Fig.1a: The authors have to clarify if the L1-ORFeus-Reporter constructs used were already reported and described in earlier publications or if they were generated by the authors and used here in this work for the first time. If the latter is true, expression and functionality of the L1 reporter elements has to be confirmed (e.g by Immunoblot analysis with anti ORF1p antibody) and GFP expression should be demonstrated by presenting images of green fluorescent cells in cell cultures transfected with the different constructs including negative control experiments. Demonstrate absence of GFP+-cells in L1-RTm transfected cells. The authors have to include information about the original designations of what the authors call L1-WT (although it is L1-ORFeus), L1RTm and L1-ENm reported in the publication by Han & Boeke 2004 /Nature in the Methods section. How do Boeke and Han designate these constructs that were used in the manuscript.

3. The current order of the presented figures does not correspond to the order the figures are

mentioned for the first time in the main text. For example, Figure 3 is mentioned in the text before Figure 2 is mentioned for the first time. The order of figures or text has to be modified so that both correspond to each other with regard to order of figures mentioned for the first time.

4. In order to demonstrate that the data obtained with the mouse L1-ORFeus reporter could be extrapolated to human L1Hs elements, the authors should add at least one experiment using a real wildtype L1Hs retrotransposon reporter construct, confirm retrotransposition of this marked element into CRISPR/Cas9 DSBs by PolyA-seq and/ or Non-PolyA-seq, and evaluate how insertion frequency changes when a human wildtype L1Hs reporter is used.

5. Two or three 2 or three complete L1-ORFeus de novo insertions (including genomic 5'-and 3' junctions) that occurred after PE2/PE3 editing into PE2/ PE3 editing sites , need to be isolated and fully sequenced and nucleotide sequences of the junctions have to be presented.

6. page 13/ lines 269-270: As mentioned before, authors have to present complete insertions including genomic 5' and 3' junctions and present the features of the insertions with their junctions on a nucleotide level showing fetures of the insertion that 'validate the L1-ORfeus insertions in PE3 editing sites in an 'Rt-dependent manner'. Are TSDs present or absent?

7. Comparability of the number of L1-ORFeus insertions obtained after the cells were submitted to the various CRISPR/Cas9-, base editing-, and prime editor methods is only possible, if the efficiencies of the different editing methods are comparable. While the authors have provided editing efficiency data for prime editing and base editing, they did not present editing efficiency data for canonical CRISPR/Cas9 editing. The authors have to provide data confirming comparable efficiencies of the different editing methods in order to eb able to relatively quantify retrotransposition events observed after the application of the different editing methods.

8. page 6/lines 123-125: While the rare L1 insertions reported here could indeed be in theory a consequence of trans-complementation form endogenous L1HS encoded L1 proteins but it is unclear how these GFP expressing de novo insertions could result from residual L1-ORFeus reporter plasmids, because these plasmids harbour only GFP cassettes that are interrupted by an intron and , therefore cannot lead to GFP expressing cells. The authors have to clarify how such a capture could result in GFP expressing cells. Also, the authors could could easily answer this question if they would isolate an sequence complete L1-ORFeus insertions (including genomic 5'- and 3' junctions) . These sequences would elucidate how GFP expressing cells could arise.

9. Suppl. Fig 3d: The fact that L1-RTm-derived DNA inserts into DSBs at the high rate of 20% relative to L1-ORFeus is confusing and raises questions. This finding requires clarification how these L1-Rtm insertions are generated in the absence of L1-reporter encoded functional L1-RT by analyzing the structure of these insertions.

10. page 8/ lines 170-173: In order to validate that the PolyA-seq and the Non-PolyA-seq approach works well and indeed detects L1 de novo insertions/retrotransposition events that were launched from the respective L1 reporter construct, the authors should isolate 2 -3 complete de novo insertions including their genomic 5'- and 3' junctions. The nucleotide sequences of the junctions have to be presented. Of special interest would be de novo insertions that are derived from the L1-ORFeus RT-mutant. By analyzing the nucleotide sequences of the genomic junctions of these de novo insertions in greater detail, they will find out, if these de novo insertions occurred through trans-mobilization by the protein machinery of endogenous L1s or by recombination with L1 reporter plasmids (as suggested by the authors!!). Also, the sentence 'Using PolyA-seq.....in the entire genome...' is misleading because authors show in Suppl. Fig. 4c to f that there are several thousand L1-ENm and L1-RTm insertions. This has to be clarified because in its current version the text is contradicting the data preented in Suppl. Fig. 4c-f.

11. Authors have to provide a rationale for presenting GRO-seq-, PolII and ATAC-seq data in these figures or otherwise remove these data form the figures.

12. page 9/ lines 190-192: The authors need to clarify if they are implying here that these CRISPR/Cas9-generated breaks are genomic off-target pre-existing breaks caused by CRISPR/cCas9 , or how do they explain these preexisting breaks?

13. Page 10-11/ Paragraph ' L1 insertions occur at editing sites targeted by prime editors', Figure 6: To emphasize the difference between PE approach and 'canonical CRISPR/Cas9-mediated editing with regard to the number of L1-ORfeus insertions, it is essential to show that stochastics of experimental settings used in both approaches are identical. For example , it has to be ensured that comparable amounts of transfected L1-reporter plamids and transfected cells were used, and that comparable editing efficiencies were accomplished with the different editing approaches to be compared, and that retrotransposition events were measured over comparable periods of time.

14. page 11/ lines 235-238 and Fig. 8d: It is not comprehensible , why the authors tested prime

editing efficiency in the presence of L1-ORFeus and L1-RTm plasmid. Also, authors did not provide any data showing expression of both L1 reporter constructs. Evidence for expression in this experimental setting has to be provided.

15. page 17/ lines 370-373: This statement is only valid, if the editing efficiencies of canonical CRISPR/Cas9, PE2 and PE3b, and CRE and ABE are comparable. While the authors have provided editing efficiency data for PE2, PE3b, CBE and ABE approaches used in this work, there were NO DATA presented on editing efficiencies of the presented canonical CRISPR/Cas9 approaches.

16. page 18/ lines 391-393: To convincingly demonstrate that the authors identified endogenous L1Hs retrotransposition events in the CRISPR/Cas9 cleavage sites, the authors have to i) provide evidence that intact endogenous L1Hs elements are expressed in these cells (for example by immunoblot analysis for endogenous L1-ORF1p expression) and ii) provide nucleotide sequences of a few complete L1Hs de novo insertions in the CRISPR/Cas9 -caused DSBs including hallmarks of L1 retrotransposition events (5'truncation or full length insertion but intact 3' end including poly A tail at 3' end but absence of TSDs). The authors have to assess the possibility that the observed insertions are a consequence of genomic recombination events by analyzing complete insertions including genomic 5' and 3' junctions.

17. page 18/ lines 385-387: Presented reference #71 (Wissing et al. 2012) is not appropriate because it does NOT report reactivation of retrotransposition of endogenous, genomic L1 elements in hESCs and hiPSCs. The publication by Wissing et al 2012 shows only expression of endogenous L1 mRNAs and L1-ORF1p in hESCs and hiPSCs, and it reports L1 retrotransposition from an engineered, plasmid-encoded L1 reporter construct in pluripotent stem cells. In contrast, Klawitter et al. 2016/Nature Communication 7:10286 (DOI:10.1038/ncomms10286) reports for the first time reactivation of retrotransposition of genomic, endogenous LINE-1, Alu and SVA retrotransposons in both hESCs and hiPSCs. Therefore, the reference Wissing et al.2012 has to be replaced here by Klawitter et al.2016.

General problem: The manuscript suffers from frequent imprecise and sloppy wording describing presented results. The authors are recommended to involve a native speaker proofreading the manuscript before submission of a revised version. There are two exemplary sentences that desperately need rewording and are located on page 12 (line 256-259 :'Characteristics of.....Supplementary Fig 10c)) and page 16 (lines 335-339: Together, these...flanking the DSB.)

Minor comments

- Suppl. Fig. 1d: Authors have to indicate in a schematic figure where PCR primers bind on the cDNA generated from the L1-ORFeus mRNA to understand if the generated PCR products have the expected length. Which size do obtained PCR products have? This schematic will also help the authors to realize that there is only one L1 mRNA which is bicistronic and codes for both ORF1 and ORF2. Accordingly, they have to correct their wording throughout the entire manuscript and correct sentences like 'RT PCR was performed to detect expression of ORF1 and ORF2 mRNA at day 2.'(Page 32, Suppl. Fig legend/ lines 803-804).
- Page 3, line 53: Correct 'microhomology via end joining (EJ)' to 'microhomology-mediated end joining (MMEJ)'
- Page 3, line 55: Add here Morrish et al.2002/ Nature Genetics 31, and more current literature such as Morales et al. 2015/ PLOS Genetics 11(3) e1005016 and Ono et al. 2015/ Scientific Reports 5: 12281 to the references.
- Page 4, line 69: Replace '...at the target sequence (typically; 5'-TT/AAA-3')...' by '...at the consensus target sequence (5'-TT/AAA-3')...'
- Page 4, line 71: Correct '...between thymines at the...L1 RNA.' to '..thymidines at the cleavage site and the polyA stretch at the 3' end of L1 mRNA.'
- Page 4, line 74: Correct '...incomplete, lacking...end.' To '...incomplete because the majority of L1 insertions is 5'-truncated.'
- Page 5, line 94: Correct '...induced GFP+ cells...' to '..GFP-expressing cells...'

- page 6/lines 105-107/Fig.3b: The authors mention Myc and Rag1 as those genes that were targeted, but present only Myc in Fig. 3b. The authors have to either change the wording of this sentence and refer to Fig. 3b or change Figure 3b by including Rag1.
- page 6/lines 119-120: The manuscript requires proofreading by a native speaker. One example requiring proofreading is the sentence presented from line 119-120. Please replace sentence by '

The number of L1 de novo insertions at the CRISPR/Cas9-mediated DSBs that were derived from the L1-EN mutant construct, were reduced relative to the L1-WT reporter, and extremely low in the case of L1-RTm.'

- Page 7, lines 128-129; page 13, lines 266-268; page 16/ line 346: It is important to note either in the results section or in the discussion section that the 1-12 nt microhomologies reported for both DSB-left-end and DSB-right-end (Fig. 2d) and on 'Nick on the edited strand' and 'Nick -on the non-edited strand' (Suppl. Fig 11 h) are consistent with the microhomologies reported for the 5'-junctions of human L1 insertions that occurred via TPRT (Zingler et al.2009/ Genome Research 15: 780-789). This suggests that a cellular nonhomologous DNA end-joining (NHEJ) pathway may be involved in these L1 insertions in DSBs. Zingler et al.2009 has to be referenced here or in the discussion section. The authors also have to explain what they mean by direct junctions
- page 7/ lines 130-131: The wording is unclear. Does the author mean '...the extension of the L1-insertions from L1-ORFeus or L1-ENm reporter insertions showed a strong bias towards the 3'end of the respective L1 EGFP reporter cassette.' or '...distribution of L1 insertions derived from L1-ORFeus or L1-ENm reporters showed a strong insertional bias into the 3'end of L1-GFP reporter...'. The meanings of both sentences are completely different. Please use unambiguous and clear verbalization – use native speaker to avoid misconceptions!
- page 7/ lines 135 -136: L1 retrotransposition events are mediated by the entire L1-encoded protein machinery and not by ORF2p alone. The wording has to be corrected accordingly to : '...reminiscent of canonical L1 insertions mediated by the L1 protein machinery...'
- page 8/ lines 160-161: This sentence is not verbalized unambiguously and could lead to misconceptions: It is unclear, if the authors want to evaluate features of de novo L1 reporter insertions at CRISPR/Cas9-initiated DSBs versus L1 reporter de novo insertions or preexisting L1s insertions genome-wide. The authors have to distinguish between L1 reporter de novo insertion and preexisting L1 insertions! If the authors do not discriminate between these two kinds of L1 insertions, the text becomes confusing to the reader!
- Page 9, line 175: The publication by Feng et al. 1996/ Cell 87: 905-916 has to be referenced at least in addition to #37 and #43 because function and cleavage specificity of L1-EN has been described for the first time in this milestone publication.
- page 26/ lines 677 ff: Correct Figure legend of Fig. 1e. To my understanding, the X-axis indicates the nucleotide position of the full length L1-GFP reporter cassette. Therefore, the position numbers under the respective bars indicate the position of the nucleotide that represents the 5'-end of the de novo insertion. If this interpretation is correct, the authors are asked to describe it that way in their figure legend.
- page 27/ lines 696-698/Fig-legend of Fig. 2c: The figure legend is only rudimentary and not sufficient because the figure is not self-explanatory. The authors have to provide a more thorough and detailed description of this figure. Why is there + and - curves presented for the 'DSB-right end' but not for the 'DSB-left-end'?
- page 28/ line 718/Fig-legend of Fig. 3c: Indicate how many nucleotides are covered by the chromosomal region encompassing 30 independent L1-ORFeus insertions.
- page 32/ line 794: The authors are asked to discipline their sloppy language in many places throughout the manuscript and replace '...at the indicated days after L1-WT transfection .' by '...at the indicated days after transfection of the L1-ORFeus reporter plasmid.'
- page 32/ line 804: The authors are asked to discipline their sloppy language in many places throughout the manuscript and replace 'HPRT as a loading control.' by 'HPRT expression served as loading control.'

Reviewer #2:

Remarks to the Author:

In the manuscript entitled "Frequency and mechanisms of LINE-1 retrotransposon insertions at CRISPR/Cas9 sites", Tao and colleagues develop novel methods to characterise insertions of LINE1 retrotransposons in human cells. Using these new methods, authors conclude that LINE-1 insertions can occur at DNA breaks induced by CRISPR/Cas9 in cultured human cells.

This could be a study of paramount importance, as uncover a potential severe limitation of CRISPR/Cas9 systems. These potential limitations will hamper the future applications of CRISPR/Cas9 systems, which are widely used now to edit genomes in basic and clinical research. In fact, research aimed to understand the limitation/s of novel therapeutic tools is important and

can prevent major disasters if these tools are ever used in humans without their thorough characterisation. Thus, I value both the data included in the study and its impact. CRISPR/Cas9 is perhaps the most important biotech development in recent years, and there is hope for treatment and curing several human disorders using this novel technology. As a result, any study aiming to prove limitations of such a revolutionary tool should be of the maximum quality possible. And is here where I have a major issue with the publication of this study. On one hand, the manuscript omits key methodological details that limit its review; additionally, the conditions used by authors are far (or very far) from a physiological context. I strongly feel that authors have overinterpreted their data, and as a result, their conclusions are bombastic. In fact, authors focus on techniques that rarely will be used in humans (canonical Cas9) and more importantly, use transformed cell lines which are far from physiological models to study retrotransposition. The study is not novel, as it is well established that retrotransposon cDNAs can be captured at DNA breaks.

In sum, the manuscript is preliminary, lacks key methodological details/information, is based on the transfection of 5-times more L1 than other studies and fail to validate the non-polyA method to sequence L1 integrations (see below). Thus, I would not recommend publication of this study. Below I include several major and minor points for the consideration of authors.

MAJOR POINTS

1. Description of L1 retrotransposition materials. Presumably, authors used human LINE-1 constructs; I am basing my conclusion on the position of EN and RT missense mutations. However, methods indicate that these plasmids were previously described and direct to reference 38. Unfortunately, ref 38 is for the codon optimization of a mouse L1 (L1spa), AND THERE IS NO DESCRIPTION OF HUMAN CODON OPTIMISED PLASMIDS IN THIS STUDY.

Thus, evaluating the quality of data and conclusions is very complicated. In fact, codon optimization is known to overproduce L1 encoded proteins, and some reports even suggest can alter important aspect of L1 biology such as translation of ORF2 (Alisch et al., 2006). Additionally, what is Empty Vector (EV)? It can't be pCEP4 (Invitrogen) as stated in methods. In all PCRs shown in the manuscript, EV transfected cells amplify a 292bp band indicative of retrotransposition, but why?? This is very confusing.

Which human L1 was used as a basis for the codon optimization? Does the construct used contain the 5' UTR of the L1? Where were the retrotransposition indicator cassettes cloned? etc
A proper description of materials is not only needed to be able to review this study, but also for colleagues aiming to replicate these data.

2. Retrotransposition assays: beside not knowing which plasmids were used, authors transfect 5-times the amount of L1 plasmids normally used in 2×10^5 cells (see Kopera et al., for a recent review). Why? Using a codon optimised L1, although clearly increase expression and retrotransposition, is far from physiological; if on top authors transfected 5-times the amount of DNA normally used, I am not sure whether authors are chasing an artefact created by heavily overexpressing L1 in transformed cells.

Thus, authors need to titrate the amount of plasmid transfected, ideally using a natural L1, control for toxicity, and then revisit the whole series of experiments included in this study (from Figure 1).

3. Choice of cells for retrotransposition assays. HEK293T cells are known to over replicate SV40-containing plasmids, as pCEP4. When considering that authors used 5-times more DNA than usual, and a codon optimized L1 (human??), the expression level of L1 encoded proteins in assays is far from physiological. Consistently, FigS1e shows that transfected HEK293Ts express more L1-ORF1p than hESCs, which are a positive control of L1 expression.

Thus, authors should use a primary cell type and physiological L1 expression levels when analysing whether L1 could (or not) use CRISPR-induced DSBs for integration.

4. Retrotransposition/CRISPR experiments. According to the scheme shown in Fig 1b, I doubt authors could control for retroviral integration, as at the time of infection cells were already resistant to puromycin. One more time, there are key details missing (MOI used, did authors use polybrene, etc) that make interpreting these data very problematic/impossible. Because cells were already puromycin resistant, how could authors control for cleaving efficiency of Cas9? How do they compare cells transfected with WT-L1 (human??) with ENm-L1 if there is no way to ensure that the same number of DSBs are generated? There is no description of how experiments shown in Figure 1f and S2c, which measure efficiency of cleaving by Cas9, were

made. Again, lack of details prohibits to properly evaluating these data.

5. DATA INTERPRETATION: Frankly, it is already weird that a mutation known to reduce L1 retrotransposition by >50-fold (RT mutant) could generate L1 insertions, but how can cells transfected with EMPTY VECTOR (EV) be able to generate L1 insertions? See Fig 1d. This is simply impossible. What are authors exactly sequencing?

6. Similarly, the presence of introns in sequenced L1 integrations is inconsistent with bona fide L1 Reverse Transcription mediated processes. In fact, considering the serious over transfection of L1 plasmids and their replication in HEK293T (points2&3), and the inability to control for Cas9 cleaving (point 4), it is very likely that a large proportion of "L1 integrations" might simply reflect PCR artefacts, which are common when applying NGS to L1 sequences.

7. While poly-A-seq data is consistent with L1 biology, which is not surprising when considering that authors used the same method (or a very similar approach) recently employed to characterize >80K insertions in a variety of physiologically relevant human cells (Flasch et al., 2019), the same is not true for the non-polyA datasets. In fact, there is no validation of the non-polyA method (i.e., not a single L1 insertion has been characterized at both ends). This is not trivial, as Flasch et al reported that when using the poly-A-seq method, as much as 40% of all sequencing CCSs (Pac Bio) correspond to plasmid artefacts.

In fact, when looking at the position of primers in the non-polyA method (FigS4), EGFP expression is expected from insertions captured using this method, Then, why authors do not detect EGFP-expressing cells in retrotransposition assays using L1-ENm but can detect as much as 30% the number of insertions detected with WT-L1. In other words, at day 11, nearly 10% of cells transfected with WT-L1 express EGFP, and using the non-polyA method, authors captured 127K insertions; however, when using the ENm-L1, EGFP expression of EGFP is presumably 0% but authors can sequence nearly 39K insertions. This strongly suggest that the majority of insertions captured using the non-polyA method might correspond to artifacts.

Authors need to validate many de novo L1 integrations using the non-polyA method, at both ends.

8. While the comparisons with PE&CBE are nice, they don't represent an apple vs apple comparison with canonical CRISPR-Cas9. Thus, to truly demonstrate that L1 preferentially target DSBs, authors should use a nickase Cas9 in the context of LentiCRISPR, as this would be a side-by-side comparison with Cas9.

MINOR POINTS

a) INTRODUCTION: Is over negative and only focus on limitations of CRISPR/Cas9. To be fair and collegial, authors should, at the very least, comment on the expectations that CRISPR/Cas9 offer in Biomedicine.

b) INTRODUCTION: Tone down statements such as: "All retrotransposition in humans depends on a reverse transcriptase (RT) encoded by the long-interspersed element-1 (LINE-1, or L1)²⁶." While likely true, authors should also acknowledge that: 1) There are HERV with coding potential for pol; 2) perhaps more relevant, some DNA polymerases are known to have intrinsic RT activity; and 3) Telomerase is just another RT source in human cells.

c) INTRODUCTION: There are many many references missing. Below I include a few examples, but admittedly, the Introduction needs substantial revision.

- Cite Beck et al., Cell-2010 when referring to number of active L1s in humans. By the way, the accepted nomenclature for these is "Retrotransposition Competent L1s"

- Add additional known activities of L1-ORF1p and add references!

- Add Symer et al., Embo 2002 for TPRT of human L1s.

- Add references when referring to the consensus sequence of L1-EN: Jurka-1997 and Flasch et al., 2019 among others.

- etc

d) Add sequence of primers used to explore L1 retrotransposition by regular PCR

e) Figures and panels are all over the place, jumping from Fig 1 to 3 etc etc. Why not using a

chronological referral to Figures as most studies do?

Manuscript: Tao et al. (NCOMMS-21-38175)

Title: Frequency and mechanisms of LINE-1 retrotransposon insertions at CRISPR/Cas9 sites

Point-by-point rebuttal letter

Reviewer #1:

In the manuscript entitled 'Frequency and mechanisms of LINE-1 retrotransposon insertions at CRISPR/Cas9 sites' by Tao et al. (M NCOMMS-21-38175), the authors addressed the question if and to what extent L1 de novo insertions occur into DSBs after canonical CRISPR/Cas9-editing and as a consequence of prime editing applications. This is an important question because the identification of DNA modifications occurring at CRISPR/Cas9-target and off-target sites is critical to determine efficiency and safety of genome editing tools. The authors report two newly developed sequencing techniques termed PolyA-seq and Non-PolyA-seq which they used to identify mouse L1-ORFeus de novo insertions that occurred into canonical CRISPR/Cas9 editing sites in human HEK293 and HeLa cells, and insertions that occurred after introduction of nucleotide changes by cytidine or adenine base editors or by prime editing. Presented data suggest that L1 RT-dependent L1 insertions can exploit DNA lesions associated with PE3 editing although they occurred less frequently compared to L1 insertions that occurred into CRISPR/Cas9-caused DSBs, and are largely excluded during PE2 or PE3b editing.

The finding that L1 insertions at DSBs are not depending on the presence of an EN consensus motif is not new and has been reported earlier. It also would have been expected that L1 retrotransposition events could retrotranspose into DSBs cause by CRISPR/Cas 9 because it has been reported earlier that L1 and Alu retrotransposons can integrate into preexisting DSBs generated by other mechanisms. However, the presented data demonstrate that retrotransposon insertions can be a potential outcome of CRISPR/Cas9 genome editing (if functional L1s are expressed in the respective cell line), and add to the characterization of safety and efficiency of the different editing tools. This manuscript is highly relevant to evaluate the risks for safety and efficiency of genome editing technologies for gene- and/ or cell therapeutic approaches. Although the authors present an overwhelming amount of data, I would exclude this manuscript from being published in the journal Nature Communication in its current form. There are major points that have to be addressed before the manuscript can be considered for publication.

We would like to thank the reviewer for his/her valuable comments and the recognition that our study is "highly relevant to evaluate the risks for safety and efficiency of genome editing technologies" and that there is "an overwhelming amount of data". As suggested, we have incorporated the answers to this Reviewer's concerns in a revised version of the manuscript.

Major Criticism:

1. It is misleading when the L1-ORFeus reporter element is termed 'L1-WT' because this

reporter element is a synthetic, codon-optimized mouse L1 element which the authors transfected into human cells. The mouse L1-ORFeus RNA sequence is completely different from the human L1Hs wildtype RNA and retrotransposition levels of L1-ORFeus is more than 200-fold increased relative to wildtype L1 elements from mouse. Therefore, the authors have to choose another designation for the L1-ORFeus reporter whose EN and RT was not inactivated by mutations in EN or RT domain. My suggestion would be to use the designation 'L1-ORFeus' consistently throughout the manuscript. The lettering of all figures in the manuscript has to be changed accordingly. It is questionable to what extent data obtained from this mouse L1-ORFeus element can be extrapolated to functional endogenous L1Hs elements in human cells. The question arises if it might have been more appropriate to use the human codon-optimized, synthetic L1-ORFeus element for these studies which also has been generated and published by the Boeke Lab several years ago.

We regret using the wrong citation here. Our L1 plasmid is indeed a human codon-optimized, synthetic L1-ORFeus modified from the human codon-optimized, synthetic L1-ORFeus constructs published by the Boeke Lab (An et al., 2011). The exact plasmid sequences can be found in **Supplementary Sequence 1** for the Reviewer's and readers' use. We will also deposit the plasmids in Addgene to make it available to the scientific community upon publication of the manuscript. In the revised manuscript, we corrected the citation and now clearly specified human codon-optimized, synthetic L1-ORFeus to avoid any misunderstanding.

We also agree the label "L1-WT" to mean L1-ORFeus reporter is misleading, so according to this Reviewer's suggestion we changed the designation of the "L1-WT" reporter to "L1-ORFeus" throughout the manuscript and figures.

2. page 5/lines 97/Suppl.Fig.1a: The authors have to clarify if the L1-ORFeus-Reporter constructs used were already reported and described in earlier publications or if they were generated by the authors and used here in this work for the first time. If the latter is true, expression and functionality of the L1 reporter elements has to be confirmed (e.g by Immunoblot analysis with anti ORF1p antibody) and GFP expression should be demonstrated by presenting images of green fluorescent cells in cell cultures transfected with the different constructs including negative control experiments. Demonstrate absence of GFP⁺-cells in L1-RT^m transfected cells. The authors have to include information about the original designations of what the authors call L1-WT (although it is L1-ORFeus), L1RT^m and L1-EN^m reported in the publication by Han & Boeke 2004 /Nature in the Methods section. How do Boeke and Han designate these constructs that were used in the manuscript.

As indicated above, our pCEP4 L1-ORFeus expression constructs do contain the human codon-optimized, synthetic L1-ORFeus sequence published by the Boeke Lab (An et al., 2011). Our pCEP4 vectors also contain a puromycin resistance gene, instead of the hygromycin resistance gene in the original pCEP4 vector from Invitrogen. Our L1-ORFeus mutant constructs contain the following modifications: the EN mutant construct contains the H230A residue in ORF2p (L1-EN^m) and the RT mutant construct contains the D702Y residue in ORF2p (L1-RT^m), which are disrupted residues critical for retrotransposition, as previously shown (Adney et al., 2019;

Moran et al., 1996), and confirmed here. Meanwhile, we also included a GFP control vector (called Empty Vector in the original version) which expresses an intron-less GFP reporter in the same pCEP4 vector backbone (**new Supplementary Fig. 1a**). We have changed the designation of the “Empty vector (EV)” to “GFP control vector (GFP control)” in the revised manuscript to avoid any misunderstanding. All constructs were sequence-verified.

As requested by the Reviewer we now provide evidence that validates the L1-ORFeus constructs we have used in this study. For example, we have confirmed that all L1-ORFeus express comparable ORF1p in HEK293T cells using immunoblotting analysis (**Supplementary Fig. 1g**). We also showed our gating strategy for our flow cytometry analysis to detect GFP-expressing cells induced with the L1-ORFeus constructs transfected in HEK293T cells, including a negative control (Untransfected) and a positive GFP expression vector control (GFP control, see **new Supplementary Fig. 1c**). In addition, we performed a PCR assay across the intron of the GFP reporter gene (intron assay), such that detection of PCR products lacking the intron indicate GFP cDNA integrated into the genome (**Supplementary Fig. 1e**). Lastly, we also conducted RT-PCR to show comparable L1-ORFeus RNA level (**Supplementary Fig. 1f**).

Importantly, our data are consistent with previous reports showing that D702Y RT mutation fully disrupts retrotransposition, and the H230A EN mutation causes a drastic decrease in retrotransposition, based on our flow cytometry analysis and the PCR intron assay (**Supplementary Fig. 1d,e**). Thus, we have validated the L1-ORFeus constructs we have used in this study.

Overall, our approach is to collect and sequence the bulk population of cells without enriching for GFP-expressing cells. The reason to do so was to have data of all insertion types (plasmid, genomic, PE insertions) also in the RT mutant that has almost no GFP-expressing cells because of the defective RT activity. By sequencing the bulk population, we had comparable data in all 4 conditions (GFP control, L1-ORFeus, L1-ENmut, L1-RTmut) of all types of insertions.

3. The current order of the presented figures does not correspond to the order the figures are mentioned for the first time in the main text. For example, Figure 3 is mentioned in the text before Figure 2 is mentioned for the first time. The order of figures or text has to be modified so that both correspond to each other with regard to order of figures mentioned for the first time.

We agree with this point, and have revised the manuscript accordingly.

4. In order to demonstrate that the data obtained with the mouse L1-ORFeus reporter could be extrapolated to human L1Hs elements, the authors should add at least one experiment using a real wildtype L1Hs retrotransposon reporter construct, confirm retrotransposition of this marked element into CRISPR/Cas9 DSBs by PolyA-seq and/ or Non-PolyA-seq, and evaluate how insertion frequency changes when a human wildtype L1Hs reporter is used.

As described above, we are using the human, not mouse, L1-ORFeus sequence published by the

by the Boeke Lab (An et al., 2011) in our expression constructs and we apologize again for generating confusion on this point.

The reason we use codon-optimized human L1-ORFeus reporter is to clearly distinguish L1-ORFeus insertions from insertions of the genomic L1 sequences. Genomic insertions are by far the most common type of insertions observed in CRISPR/Cas9 sites (>70% of insertions, **Fig. 1c**). Because globally L1 sequences comprise ~17% of the human genome (Lander et al., 2001), many insertions and rearrangements at CRISPR/Cas9 sites contain L1 sequences without necessarily representing *de novo* retrotransposition events. Use of a codon-optimized human L1-ORFeus reporter allows us to clearly distinguish *de novo* insertions from insertions of inactive endogenous L1 elements by the sequence (**Supplementary Sequence 2**).

Nevertheless, as suggested by the Reviewer, we also sought to assay the insertions at CRISPR/Cas9 DNA breaks from a “real wildtype L1Hs retrotransposon reporter construct”. For this, we used a construct containing a full-length native L1Hs sequence competent for retrotransposition (L1RP). The L1RP reporter was modified from the Boeke Lab (An et al., 2011) and published by us more recently (Addgene plasmid #131392) (Ardeljan et al., 2020). The L1RP sequence was further mutated to generate the EN mut (H230A) (L1RP-ENm) vector or the RT mut (D702Y) (L1RP-RTm) vector. Specifically, the L1RP reporter contains the native L1 5'UTR promoter and the 3' GFP retrotransposition reporter cassette with an artificial intron (**new Supplementary Fig. 8a**). The full sequence of the L1RP reporter vector is shown in **new Supplementary Sequence 13**.

First, we validated these new reporters by several assays. Immunoblot analysis showed that ORF1p expression levels is consistent across L1RP, L1RP-ENm and L1RP-RTm reporters and slightly lower than observed with the L1-ORFeus (**new Supplementary Fig. 8c**). The percentage of GFP-expressing cells is lower with the L1RP reporter compared to L1-ORFeus reporter (**new Fig. 6a**). Consistently, the PCR assay across the intron of the GFP reporter gene showed that the removal of the intron in the L1RP reporter was less efficient than in the L1-ORFeus reporter, and not observed in the L1RP-ENm and L1RP-RTm mutants (**new Fig. 6b**).

Based on these validations, we repeated key experiments in HEK293T cells with these L1RP reporters (**new Supplementary Fig. 8b**). Consistent with the L1-ORFeus results, we detected *de novo* L1RP retrotransposition events at CRISPR/Cas9 target site by PolyA-seq, indicating that also a “real wildtype L1Hs reporter” can retrotranspose at CRISPR/Cas9 genome editing sites. Remarkably, no insertions were observed with the L1RP-RTm reporter and only few insertions with the L1RP-ENm reporter (**new Fig. 6c,d**). The consensus motif of L1RP pre-integration site was lost within ± 5 kb of the CRISPR/Cas9-induced DSB whereas it was readily identified in insertions distant from the DSB (**new Fig. 6e**). Overall, data obtained with the L1RP reporter were completely concordant with those obtained with the L1-ORFeus.

5. Two or three complete L1-ORFeus *de novo* insertions (including genomic 5'-and 3' junctions) that occurred after PE2/PE3 editing into PE2/ PE3 editing sites, need to be isolated and fully sequenced and nucleotide sequences of the junctions have to be presented.

We agree with this point and are now providing the sequences of six L1-ORFeus *de novo* insertions obtained by our amplicon sequencing approach (data from **Supplementary Fig. 12e**) - including genomic 5'-and 3' junctions - that occurred after PE3 editing into MYC locus in **new Supplementary Sequence 4**.

Specifically, we also provide 2 example sequences bridging the intron of GFP from PE3 (MYC +2-4AAA del) editing in **new Supplementary Sequence 5**. These intron-removed *de novo* L1-ORFeus insertions indicate they are real active insertions from cDNA generated by L1 ORF2p, rather than plasmid integrations.

Furthermore, we systematically analyzed the L1-ORFeus sequences bridging the intron of GFP and showed some example sequences from CRISPR/Cas9 editing in *MYC*, *RAG1* and *CCR5* loci in HEK293T cells, as well as in the *MYC* locus in HeLa cells obtained by amplicon sequencing approach (data from **Fig. 2b, Supplementary Fig. 2i, 3g, 6g**). As shown in **new Supplementary Sequence 6-9**, these L1-ORFeus *de novo* insertions in which the intron has been removed support the interpretation as insertions generated by L1 ORF2p activity.

6. page 13/ lines 269-270: As mentioned before, authors have to present complete insertions including genomic 5' and 3' junctions and present the features of the insertions with their junctions on a nucleotide level showing features of the insertion that 'validate the L1-ORFeus insertions in PE3 editing sites in an 'Rt-dependent manner'. Are TSDs present or absent?

In the revised manuscript we re-analyzed precise junctions (to the nucleotide) of 146 complete L1-ORFeus *de novo* insertions that occurred after PE3 editing into PE3 editing sites obtained by amplicon sequencing approach (data from **Supplementary Fig. 12e-g**). We specifically investigated TSD sequences and showed that TSD sequences are absent (examples are shown in **new Supplementary Sequence 4,5**).

Meanwhile, we similarly re-analyzed junctions of L1-ORFeus insertions bridging the intron of GFP from CRISPR/Cas9 editing in *MYC*, *RAG1* and *CCR5* loci in HEK293T cells, as well as in the *MYC* locus in HeLa cells obtained by amplicon sequencing approach (data from **Fig. 2b, Supplementary Fig. 2i, 3g, 6g**). Here, we show that TSD sequences are absent (examples are shown in **new Supplementary Sequence 6-9**).

7. Comparability of the number of L1-ORFeus insertions obtained after the cells were submitted to the various CRISPR/Cas9-, base editing-, and prime editor methods is only possible, if the efficiencies of the different editing methods are comparable. While the authors have provided editing efficiency data for prime editing and base editing, they did not present editing efficiency data for canonical CRISPR/Cas9 editing. The authors have to provide data confirming comparable efficiencies of the different editing methods in order to be able to relatively quantify retrotransposition events observed after the application of the different editing methods.

While we agree in principle with this point raised by the Reviewer, we think that the

mechanisms of DSBs formation by canonical CRISPR/Cas9 are very different from mechanisms of DNA editing of the base editors or prime editors, limiting the possibility for an accurate comparison. Nonetheless, we want to highlight that we have provided the editing efficiency data for canonical CRISPR/Cas9 in our study, based on the generation of indels, which is the most common method to estimate editing efficiency (Rao et al., 2021). By this approach, we demonstrate that the editing efficiency of canonical CRISPR/Cas9 is ~30% at the *MYC* locus in cells transfected with the L1-ORFeus reporters (**Supplementary Fig. 1h**), compared to ~25-30% efficiency for the PE3 system at the *MYC* locus in cells transfected with the L1-ORFeus reporters (**Supplementary Fig. 9d, Supplementary Fig. 10c**) and ~40% for the most efficient base editors (AncBE4max and ABE8e) at the *MYC* locus in cells transfected with all L1-ORFeus reporters (**Supplementary Fig. 13d**). Of note, we used the same targeting sgRNA sequence in the *MYC* locus for the canonical CRISPR/Cas9, the base editors and the prime editors to make results as comparable as possible (**Supplementary Table 1**).

8. page 6/lines 123-125: While the rare L1 insertions reported here could indeed be in theory a consequence of trans-complementation from endogenous L1HS encoded L1 proteins but it is unclear how these GFP expressing de novo insertions could result from residual L1-ORFeus reporter plasmids, because these plasmids harbour only GFP cassettes that are interrupted by an intron and , therefore cannot lead to GFP expressing cells. The authors have to clarify how such a capture could result in GFP expressing cells. Also, the authors could easily answer this question if they would isolate a sequence complete L1-ORFeus insertions (including genomic 5'- and 3' junctions). These sequences would elucidate how GFP expressing cells could arise.

As described in response to Major point #2, to detect insertions at CRISPR/Cas9 editing sites we collected and sequenced the bulk population of cells and not only GFP-expressing cells. We clarify that this is our approach in the revised manuscript. This strategy captures all L1-ORFeus insertions including those that do not result in GFP-expressing cells, like short insertions of portion of the GFP-cassette as described above in response to Major Point #5 and #6.

The rare insertion sequences obtained in cells transfected with the L1-RTm likely do not reconstitute GFP, but rather reflect incorporation of the expression plasmid still present in the cells at the time of CRISPR/Cas9-induced DSBs even 5 days after transfection. Alternatively, we cannot exclude that some of these rare events could originate by trans-complementation from endogenous active L1s. Of note, rare plasmid insertions were observed also in cells transfected with the GFP control plasmid (**Fig. 1d,e**) (**two** insertions at the *MYC* locus and **four** insertions at the *RAG1* locus targeted by canonical CRISPR/Cas9).

As an example, we provide the only sequence identified in L1-RTm transfected cells with CRISPR/Cas9 editing of the *CCR5* gene. This sequence, including genomic 5'- and 3' junctions, is shown in **new Supplementary Sequence 10**. This sequence mapped to the intron of the GFP cassette, suggesting a plasmid insertion rather than a product of retrotransposition.

9. Suppl. Fig 3d: The fact that L1-RTm-derived DNA inserts into DSBs at the high rate of 20% relative to L1-ORFeus is confusing and raises questions. This finding requires clarification how

these L1-RTm insertions are generated in the absence of L1-reporter encoded functional L1-RT by analyzing the structure of these insertions.

We agree with this point raised by the Reviewer that insertions in samples expressing the L1-RTm reporter are unexpected. To further remove any possible sources of background noise, we manually curated the 3 insertion sequences observed in Supplementary Fig 3d. Among them, one read was correct (shown in the **new Supplementary Sequence 10**), while the other two reads contained the sequence of the reporter vector abnormally attached external to the primers used for the PCR amplification and was likely a rare non-specific PCR artifact. To clean out these rare non-specific reads, we further improved our filtering computing method and re-analyzed all the data based on PCR amplicon sequencing in the entire manuscript. Overall, changes were minimal with just few reads being filtered out in some experimental conditions after this further removal of PCR artifacts.

After this re-analysis, the percentage of L1-ORFeus insertions is now updated as 10% in L1-ORFeus and 0.56% in L1-RTm (i.e. only one insertion) as shown in the revised **Supplementary Fig. 3d**. The single insertion still observed in L1-RTm is now shown in the **new Supplementary Sequence 10**. This sequence mapped to the intron of the GFP cassette, suggesting a plasmid insertion rather than a product of retrotransposition.

10. page 8/ lines 170-173: In order to validate that the PolyA-seq and the Non-PolyA-seq approach works well and indeed detects L1 de novo insertions/retrotransposition events that were launched from the respective L1 reporter construct, the authors should isolate 2 -3 complete de novo insertions including their genomic 5'- and 3' junctions. The nucleotide sequences of the junctions have to be presented. Of special interest would be de novo insertions that are derived from the L1-ORFeus RT-mutant. By analyzing the nucleotide sequences of the genomic junctions of these de novo insertions in greater detail, they will find out, if these de novo insertions occurred through trans-mobilization by the protein machinery of endogenous L1s or by recombination with L1 reporter plasmids (as suggested by the authors!!).

For PolyA-seq, it is designed based on similar approaches previously validated and published (Flasch et al., 2019; Sultana et al., 2019). In particular, the primer design, library generation and data analysis of PolyA-seq approach is largely identical to a previously validated technique (Flasch et al., 2019). All these high throughput sequencing techniques are designed to capture only the 3' junctions but not the 5' junctions (**Supplementary Fig. 4a,b**). The analysis of 3' junctions of the L1-ORFeus insertions captured by our PolyA-seq showed the expected consensus motif 5'-TT/AAAAA-3' (**Supplementary Fig. 4d**), and the analysis of the 3' poly(A) tracts (range 15–75bp) (**Fig. 3a, Supplementary Fig. 4e**) were consistent with previous reports (Flasch et al., 2019; Sultana et al., 2019). As requested by this Reviewer, we are now providing 8 examples of typical sequences obtained by PolyA-seq from *MYC* CRISPR editing in HEK293T cells transfected with L1-ORFeus (**new Supplementary Sequence 11**).

In contrast, the non-PolyA-seq is a novel sequencing approach developed by us to further

validate the data obtained by the published PolyA-seq approach. The data obtained with the non-PolyA-seq were remarkably comparable, but not superior, to those obtained with PolyA-seq, giving us confidence in their validity. Nevertheless, since PolyA-seq and non-PolyA-seq data were so comparable and somehow redundant, we streamlined the revision by presenting only PolyA-seq data in detail. Therefore, all the non-PolyA-seq data and figures were removed from the revised manuscript.

Also, the sentence 'Using PolyA-seq.....in the entire genome...' is misleading because authors show in Suppl. Fig. 4c to f that there are several thousand L1-ENm and L1-RTm insertions. This has to be clarified because in its current version the text is contradicting the data presented in Suppl. Fig. 4c-f.

We agreed that the sentence 'Using PolyA-seq.....in the entire genome...' is misleading. The text has been revised accordingly.

11. Authors have to provide a rationale for presenting GRO-seq-, PolII and ATAC-seq data in these figures or otherwise remove these data from the figures.

GRO-seq-, PolII and ATAC-seq data are useful to show the readers that in the HEK293T cell line that we selected the MYC gene is indeed active and in an open chromatin conformation, while the RAG1 gene is not transcribed and in a close chromatin conformation. Specifically, GRO-seq indicates the transcriptional activity by measuring nascent RNA; ChIP-seq of pol II and active histone modifications (H3K4me3 and H3K27ac) indicate transcriptional activity; ATAC-seq indicates chromatin accessibility. Thus, our data show that L1-ORFeus insertions are independent of gene expression or chromatin accessibility, consistent with previously work (Flasch et al., 2019; Sultana et al., 2019).

12. page 9/ lines 190-192: The authors need to clarify if they are implying here that these CRISPR/Cas9-generated breaks are genomic off-target pre-existing breaks caused by CRISPR/cCas9 , or how do they explain these preexisting breaks?

We agree that this sentence was confusing. The sentence has been revised as follows: "indicating that L1 retrotransposition in the region flanking the CRISPR/Cas9-induced DSB does not rely on the EN activity of ORF2p".

13. Page 10-11/ Paragraph ' L1 insertions occur at editing sites targeted by prime editors', Figure 6: To emphasize the difference between PE approach and 'canonical CRISPR/Cas9-mediated editing with regard to the number of L1-ORfeus insertions, it is essential to show that stochastics of experimental settings used in both approaches are identical. For example, it has to be ensured that comparable amounts of transfected L1-reporter plasmids and transfected cells were used, and that comparable editing efficiencies were accomplished with the different editing approaches to be compared, and that retrotransposition events were measured over comparable periods of time.

We agree with this Reviewer's point. The experimental settings and timing were the same for canonical CRISPR/Cas9 editing and PE editing (**Fig. 1b** for canonical CRISPR/Cas9, **Supplementary Fig. 9a** for PE). We used the same amount of transfected plasmid for canonical CRISPR/Cas9 editing and PE editing as well as the same number of HEK293T cells (see **Material and Method section**). We agree that assessing editing efficiency is an important point, also addressed in major point #7. We demonstrated that the editing efficiency of canonical CRISPR/Cas9 is ~30% in the *MYC* locus (**Supplementary Fig. 1h**), compared to ~25-30% efficiency for the PE3 system in the *MYC* locus (**Supplementary Fig. 9d, Supplementary Fig. 10c**) and ~40% for the most efficient base editors (AncBE4max and ABE8e) in the *MYC* locus (**Supplementary Fig. 13d**). Of note, we used the same targeting sgRNA sequence in the *MYC* locus for the canonical CRISPR/Cas9, the base editors and the prime editors to make results as comparable as possible (**Supplementary Table 1**).

14. page 11/ lines 235-238 and Fig. 8d: It is not comprehensible, why the authors tested prime editing efficiency in the presence of L1-ORFeus and L1-RTm plasmid. Also, authors did not provide any data showing expression of both L1 reporter constructs. Evidence for expression in this experimental setting has to be provided.

We tested prime editing efficiency in the presence of L1-ORFeus and L1-RTm plasmid because we wanted to be sure that the experimental conditions were comparable. As raised by this Reviewer in major point #7 and #13, comparing editing efficiency in the different conditions is important to support the data. In response to major point #2 we provided explanations on how we validated the expression of L1-ORFeus constructs. Indeed, we showed that L1-ORFeus and L1-RTm express comparable levels of ORF1p using immunoblotting analysis (**Supplementary Fig. 1g**). In addition, we detected similar levels of ORFeus RNA in cells transfected with L1-ORFeus versus L1-RTm (**Supplementary Fig. 1f**). Finally, we showed that comparable amounts of L1 reporter constructs were transfected into cells by PCR of the intron-containing band (that originated from plasmid sequence), as shown in **Supplementary Fig. 9b,c** for PE3 (MYC +2-4AAA del), **Supplementary Fig. 10a** for PE3 (MYC +5GtoC), **Supplementary Fig. 10b** for PE3 (FANCF +5GtoT), **Supplementary Fig. 13a,b,c** for Base editors.

15. page 17/ lines 370-373: This statement is only valid, if the editing efficiencies of canonical CRISPR/Cas9, PE2 and PE3b, and CRE and ABE are comparable. While the authors have provided editing efficiency data for PE2, PE3b, CBE and ABE approaches used in this work, there were NO DATA presented on editing efficiencies of the presented canonical CRISPR/Cas9 approaches.

We agree with this point that has already been addressed above in major point #7 and #13.

16. page 18/ lines 391-393: To convincingly demonstrate that the authors identified endogenous L1Hs retrotransposition events in the CRISPR/Cas9 cleavage sites, the authors have to i) provide evidence that intact endogenous L1Hs elements are expressed in these cells (for example by immunoblot analysis for endogenous L1-ORF1p expression) and ii) provide nucleotide sequences of a few complete L1Hs de novo insertions in the CRISPR/Cas9 -caused

DSBs including hallmarks of L1 retrotransposition events (5'truncation or full length insertion but intact 3' end including poly A tail at 3' end but absence of TSDs). The authors have to assess the possibility that the observed insertions are a consequence of genomic recombination events by analyzing complete insertions including genomic 5' and 3' junctions.

These are very important points, and we agree with this Reviewer to further characterize these rare events.

1) We provide evidence that intact endogenous L1Hs elements are expressed in HEK293T cells by immunoblot analysis for endogenous L1-ORF1p expression in **Supplementary Fig. 1g** and **new Supplementary Fig. 8c**. HEK293T cells transfected with the GFP control (that does not contain sequences encoding for exogenous ORF1p) show a distinct band, that is however weaker than cells transfected with L1-ORF1p constructs, thereby demonstrating that HEK293T express endogenous ORF1p consistent with previous report (Philippe et al., 2016).
2) We now provide two examples of L1Hs nucleotide sequences including the 5' and 3' junctions. These sequences show 5'truncation, intact 3' end with short poly A tail and absence of TSDs, as well as a young, active trinucleotide (ACA) in L1 3'UTR (**new Supplementary Sequence 12**).

Despite these suggestive features, we cannot definitely determine whether these insertions represent *de novo* L1Hs insertions or genomic insertions of non-active L1 sequences. Therefore, we decided to avoid terms such as “*de novo*” “L1 insertions” or “endogenous L1Hs retrotransposition events” to describe these sequences. Accordingly, we revised the text as follows: “We detected L1Hs sequences with bias toward the 3' of the L1Hs both in MYC and RAG1 editing sites (**Supplementary Fig. 14a-c, Supplementary Sequence 12**). However, it is difficult to conclude whether these L1Hs sequences represent retrotransposition events of active endogenous L1Hs retrotransposon occurring at CRISPR/Cas9-mediated DSBs or rather genomic insertions of non-active L1 sequences”.

17. page 18/ lines 385-387: Presented reference #71 (Wissing et al. 2012) is not appropriate because it does NOT report reactivation of retrotransposition of endogenous, genomic L1 elements in hESCs and hiPSCs. The publication by Wissing et al 2012 shows only expression of endogenous L1 mRNAs and L1-ORF1p in hESCs and hiPSCs, and it reports L1 retrotransposition from an engineered, plasmid-encoded L1 reporter construct in pluripotent stem cells. In contrast, Klawitter et al. 2016/Nature Communication 7:10286 (DOI:10.1038/ncomms10286) reports for the first time reactivation of retrotransposition of genomic, endogenous LINE-1, Alu and SVA retrotransposons in both hESCs and hiPSCs. Therefore, the reference Wissing et al.2012 has to be replaced here by Klawitter et al.2016.

Thank you for this appropriate suggestion. The reference has been changed accordingly.

General problem: The manuscript suffers from frequent imprecise and sloppy wording describing presented results. The authors are recommended to involve a native speaker proofreading the manuscript before submission of a revised version. There are two exemplary sentences that desperately need rewording and are located on page 12 (line 256-

259 :‘Characteristics of.....Supplementary Fig 10c)) and page 16 (lines 335-339: Together, these...flanking the DSB.)

We polished the manuscript with extensive proofreading. Specifically, the two indicated sentences have been rephrased as follows: “Characteristics of retrotransposed sequences (i.e., the 3’ bias and intron splicing of the inserted L1) were similar to the distribution of *de novo* L1-ORFeus insertions found at canonical CRISPR/Cas9-mediated DSBs” and “Together, these approaches captured >2,500 *de novo* L1-ORFeus retrotransposition events at CRISPR/Cas9 editing sites, as well as insertions of the lentiviral sequences that encodes for the CRISPR/Cas9 and integrations of genomic DNA sequences that largely originated from regions flanking the DSB.”

Minor comments

- Suppl. Fig. 1d: Authors have to indicate in a schematic figure where PCR primers bind on the cDNA generated from the L1-ORFeus mRNA to understand if the generated PCR products have the expected length. Which size do obtained PCR products have? This schematic will also help the authors to realize that there is only one L1 mRNA which is bicistronic and codes for both ORF1 and ORF2. Accordingly, they have to correct their wording throughout the entire manuscript and correct sentences like ‘RT PCR was performed to detect expression of ORF1 and ORF2 mRNA at day 2.’(Page 32, Suppl. Fig legend/ lines 803-804).

We provided the schematic primer position, primer sequences, PCR product length, as well as annealing temperature in revised **Supplementary Table 3**. We also carefully modified these sentences to remove any inappropriate usages.

- Page 3, line 53: Correct ‘microhomology via end joining (EJ)’ to ‘microhomology-mediated end joining (MMEJ)’

The manuscript has been revised accordingly.

- Page 3, line 55: Add here Morrish et al.2002/ Nature Genetics 31, and more current literature such as Morales et al. 2015/ PLOS Genetics 11(3) e1005016 and Ono et al. 2015/ Scientific Reports 5: 12281 to the references.

The suggested references have been added in the revised version.

- Page 4, line 69: Replace ‘...at the target sequence (typically; 5’-TT/AAA-3’)...’ by ‘...at the consensus target sequence (5’-TT/AAA-3’)...’

This sentence has been changed as suggested.

- Page 4, line 71: Correct ‘...between thymines at the....L1 RNA.’ to ‘.thymidines at the cleavage site and the polyA stretch at the 3’ end of L1 mRNA.’

This sentence has been changed as suggested.

- Page 4, line 74: Correct ‘...incomplete, lacking...end.’ To ‘...incomplete because the majority of L1 insertions is 5’-truncated.’

This sentence has been changed as suggested.

- Page 5, line 94: Correct ‘...induced GFP+ cells...’ to ‘..GFP-expressing cells...’

This sentence has been changed as suggested.

- page 6/lines 105-107/Fig.3b: The authors mention Myc and Rag1 as those genes that were targeted, but present only Myc in Fig. 3b. The authors have to either change the wording of this sentence and refer to Fig. 3b or change Figure 3b by including Rag1.

We targeted both *MYC* and *RAG1*. Data for *MYC* targeting are presented in **Fig. 3c**, the data of *RAG1* can be found in **Supplementary Fig. 4g**. In addition, all the details of canonical CRISPR/Cas9 editing in *RAG1* locus, including L1 retrotransposition activity (for both GFP assay and intron assay), CRISPR/cas9 editing efficiency, *de novo* L1-ORFeus insertions, plasmid and genomic integrations, detailed L1-ORFeus junction analysis, can be found in **Supplementary Fig. 2a-h**.

- page 6/lines 119-120: The manuscript requires proofreading by a native speaker. One example requiring proofreading is the sentence presented from line 119-120. Please replace sentence by ‘ The number of L1 de novo insertions at the CRISPR/Cas9-mediated DSBs that were derived from the L1-EN mutant construct, were reduced relative to the L1-WT reporter, and extremely low in the case of L1-RTm.’

This sentence has been replaced according to the Reviewer’s suggestion.

- Page 7, lines 128-129; page 13, lines 266-268; page 16/ line 346: It is important to note either in the results section or in the discussion section that the 1-12 nt microhomologies reported for both DSB-left-end and DSB-right-end (Fig. 2d) and on ‘Nick on the edited strand’ and ‘Nick –on the non-edited strand’ (Suppl. Fig 11 h) are consistent with the microhomologies reported for the 5’-junctions of human L1 insertions that occurred via TPRT (Zingler et al.2009/ Genome Research 15: 780-789). This suggests that a cellular nonhomologous DNA end-joining (NHEJ) pathway may be involved in these L1 insertions in DSBs. Zingler et al.2009 has to be referenced here or in the discussion section. The authors also have to explain what they mean by direct junctions

This is a good point. The text has been revised as follows: “L1 insertions showed joining to the DSB either direct (no microhomology) or via short microhomologies, mostly 1bp to 6bp with a decreasing trend over length (**Fig. 2d**), consistent with the microhomologies reported for the

5'-junctions of human L1 insertions that occurred via TPRT (Zingler et al., 2005)".

- page 7/ lines 130-131: The wording is unclear. Does the author mean '...the extension of the L1-insertions from L1-ORFeus or L1-ENm reporter insertions showed a strong bias towards the 3' end of the respective L1 EGFP reporter cassette.' or '...distribution of L1 insertions derived from L1-ORFeus or L1-ENm reporters showed a strong insertional bias into the 3' end of L1-GFP reporter...'. The meanings of both sentences are completely different. Please use unambiguous and clear verbalization – use native speaker to avoid misconceptions!

The sentence has been rephrased as follows: "the distribution of *de novo* L1 insertions derived from L1-ORFeus or L1-ENm reporters showed a strong enrichment for sequences mapping to the 3' end of L1-GFP reporter and a sharp decrease in coverage spanning the intron".

- page 7/ lines 135 -136: L1 retrotransposition events are mediated by the entire L1-encoded protein machinery and not by ORF2p alone. The wording has to be corrected accordingly to : '...reminiscent of canonical L1 insertions mediated by the L1 protein machinery...'

The sentence has been corrected accordingly.

- page 8/ lines 160-161: This sentence is not verbalized unambiguously and could lead to misconceptions: It is unclear, if the authors want to evaluate features of *de novo* L1 reporter insertions at CRISPR/Cas9-initiated DSBs versus L1 reporter *de novo* insertions or preexisting L1 insertions genome-wide. The authors have to distinguish between L1 reporter *de novo* insertion and preexisting L1 insertions! If the authors do not discriminate between these two kinds of L1 insertions, the text becomes confusing to the reader!

We agree that this sentence was confusing. In the revised manuscript it has been changed as follows: "We next sought to evaluate features of *de novo* L1-ORFeus reporter insertions at CRISPR/Cas9-initiated DSBs versus *de novo* L1-ORFeus reporter insertions genome-wide".

- Page 9, line 175: The publication by Feng et al. 1996/ Cell 87: 905-916 has to be referenced at least in addition to #37 and #43 because function and cleavage specificity of L1-EN has been described for the first time in this milestone publication.

The reference has been added.

- page 26/ lines 677 ff: Correct Figure legend of Fig. 1e. To my understanding, the X-axis indicates the nucleotide position of the full length L1-GFP reporter cassette. Therefore, the position numbers under the respective bars indicate the position of the nucleotide that represents the 5'-end of the *de novo* insertion. If this interpretation is correct, the authors are asked to describe it that way in their figure legend.

The Reviewer understood correctly that the X-axis indicates the nucleotide position of the full length L1-GFP reporter cassette. However, the bars represent the projection of the dots to the

nucleotide position of the full length L1-GFP reporter cassette. Dots and bars do not represent the 5'-end of the de novo insertion but the center position of the insertion fragment oriented either 5' to 3' (+) in orange or 3' to 5' (-) in blue. The figure legend has been edited for clarity as follows: "Numbers and fragment lengths of L1-ORFeus insertions obtained by amplicon sequencing at the *MYC* locus targeted by *MYC* CRISPR/Cas9 in HEK293T cells expressing L1-ORFeus, L1-ENm, L1-RTm or GFP control. The X-axis indicates the nucleotide position of the full length L1 (ORFeus) reporter cassette. Top: each histogram represents the number of L1 insertions mapping to a 100bp interval of the L1 (ORFeus) sequence represented on the X-axis. In orange are insertions oriented from 5' to 3' end of the L1 (ORFeus) sequence (+), in blue are insertions oriented from 3' to 5' (-); Bottom: Length of the L1 (ORFeus) insertions mapping to the L1 (ORFeus) sequence represented on the X-axis. Each dot indicates the center of the insertion fragment oriented either 5' to 3' (+) in orange or 3' to 5' (-) in blue. Each bar represents the projection of the dot to indicate the nucleotide position in the full length L1 (ORFeus) reporter cassette."

- page 27/ lines 696-698/Fig-legend of Fig. 2c: The figure legend is only rudimentary and not sufficient because the figure is not self-explanatory. The authors have to provide a more thorough and detailed description of this figure. Why is there + and – curves presented for the 'DSB-right end' but not for the 'DSB-left-end'?

The figure legend has been edited for clarity. The + and – curves are presented also for the 'DSB-left-end' but they are highly overlapping. We have now edited the figure to show that both curves are present.

- page 28/ line 718/Fig-legend of Fig. 3c: Indicate how many nucleotides are covered by the chromosomal region encompassing 30 independent L1-ORFeus insertions.

We provided this information as requested by the Reviewer and the figure legend of Fig. 3c (now **Fig. 3d** in revised manuscript) has been edited as follows: "The mean interval size covered by the chromosomal region encompassing 30 independent L1-ORFeus insertions was: 1,353.5 bp (range 198-8440 bp) within approx. ± 5 kb of the *MYC* CRISPR/Cas9 target site, and 2,824,123 bp (range 801,848-6,764,412 bp) outside of the CRISPR/Cas9 target site". Likewise, the figure legends for *MYC* CRISPR/Cas9 editing in HeLa cells (**Fig. 5d**) and *RAG1* CRISPR/Cas9 editing in HEK293T cells (**Supplementary Fig. 4h**) were edited to provide the same information.

- page 32/ line 794: The authors are asked to discipline their sloppy language in many places throughout the manuscript and replace '...at the indicated days after L1-WT transfection.' by '...at the indicated days after transfection of the L1-ORFeus reporter plasmid.'

We edited this sentence and revised all the manuscript to improve the language.

- page 32/ line 804: The authors are asked to discipline their sloppy language in many places throughout the manuscript and replace 'HPRT as a loading control.' by 'HPRT expression

served as loading control.'

As mentioned above, we edited this sentence and revised all the manuscript to improve the language.

Finally, we would like to point out that we reproduced key experiments in one more cell line, U2OS cells that is commonly used in the L1 field (Benitez-Guijarro et al., 2018; MacLennan et al., 2017). These new data obtained with CRISPR/Cas9 editing the *MYC* locus in U2OS cells are now included in the revised manuscript in the **new Supplementary Fig. 7**.

Reviewer #2:

In the manuscript entitled "Frequency and mechanisms of LINE-1 retrotransposon insertions at CRISPR/Cas9 sites", Tao and colleagues develop novel methods to characterize insertions of LINE1 retrotransposons in human cells. Using these new methods, authors conclude that LINE-1 insertions can occur at DNA breaks induced by CRISPR/Cas9 in cultured human cells.

This could be a study of paramount importance, as uncover a potential severe limitation of CRISPR/Cas9 systems. These potential limitations will hamper the future applications of CRISPR/Cas9 systems, which are widely used now to edit genomes in basic and clinical research. In fact, research aimed to understand the limitation/s of novel therapeutic tools is important and can prevent major disasters if these tools are ever used in humans without their thereof characterisation. Thus, I value both the data included in the study and its impact.

We thank this Reviewer for these comments recognizing the novelty and potential impact of these findings.

CRISPR/Cas9 is perhaps the most important biotech development in recent years, and there is hope for treatment and curing several human disorders using this novel technology. As a result, any study aiming to prove limitations of such a revolutionary tool should be of the maximum quality possible. And is here where I have a major issue with the publication of this study. On one hand, the manuscript omits key methodological details that limit its review; additionally, the conditions used by authors are far (or very far) from a physiological context. I strongly feel that authors have overinterpreted their data, and as a result, their conclusions are bombastic. In fact, authors focus on techniques that rarely will be used in humans (canonical Cas9) and more importantly, use transformed cell lines which are far from physiological models to study retrotransposition. The study is not novel, as it is well established that retrotransposons cDNAs can be captured at DNA breaks. In sum, the manuscript is preliminary, lacks key methodological details/information, is based on the transfection of 5-times more L1 than other studies and fail to validate the non-polyA method to sequence L1 integrations (see below).

We understand these general concerns raised by the Reviewer. In the revised manuscript we are now answering these concerns.

- Regarding the methodological details, in the revised version of the manuscript we are now providing all the requested details that could have been missed in the Method section as well as improving descriptions of experimental design.
- For the question of the “physiological context” we agree with this Reviewer that the L1-ORFeus system is artificial due to the overexpression of a codon-optimized vector that could be far from a physiological context. This experimental approach, however, allowed us to collect insertions at a scale in multiple experimental settings (different cell lines, multiple editing systems and multiple editing sites) that would be impossible depending on endogenous L1 retrotransposition. Similar “artificial” experimental systems (i.e., using exogenous L1 elements) have been used in landmark papers that provided important insights on the landscape of L1 retrotransposons in the human genome (Flasch et al., 2019; Sultana et al., 2019). Therefore, while we acknowledge the limitations of these models, this experimental system provides us with an unprecedented, large number of *de novo* L1-ORFeus insertions occurring at targeted genomic sites by canonical CRISPR/Cas9 editing or PE and BE editing systems.
- To further expand and validate the findings, we reproduced key findings in one more cell line (U2OS) transfected with low amount (1 µg) of L1-ORFeus plasmid. Details are described in the response to Major point #2.
- To reproduce findings in a context of more “physiological” L1 retrotransposition events, we performed new experiments with a new L1RP reporter vector transfected in low amounts (1 µg) in HEK293T cells. Details of these new experiments are provided below in the response to Major point #1.
- Regarding the comment that “canonical CRISPR/Cas9 will be rarely used in humans”, we respectfully disagree because there are already several ongoing clinical trials that are based on canonical CRISPR/Cas9, including potential treatments for Beta-Thalassemia (Frangoul et al., 2021), transthyretin amyloidosis (Gillmore et al., 2021)(NCT04601051) or for retinal degeneration (see for example NCT03872479) as well as other currently in clinical trial. For the clinical importance of these editing systems, we believe that our insights about the potential side effects is relevant for the scientific community. We hope that our findings will suggest to the scientific community that the safety of the Cas9-based editing systems needs to be carefully evaluated for potential retrotransposition events.

MAJOR POINTS

1. Description of L1 retrotransposition materials. Presumably, authors used human LINE-1 constructs; I am basing my conclusion on the position of EN and RT missense mutations. However, methods indicate that these plasmids were previously described and direct to reference 38. Unfortunately, ref 38 is for the codon optimization of a mouse L1 (L1spa), AND THERE IS NO DESCRIPTION OF HUMAN CODON OPTIMISED PLASMIDS IN THIS STUDY.

We apologize for our inadvertent use of the wrong citation. Our L1 plasmid is indeed a human

codon-optimized, synthetic L1-ORFeus modified from the human codon-optimized, synthetic L1-ORFeus constructs published by the Boeke Lab (An et al., 2011). Our pCEP4 vectors also contain a puromycin resistance gene, instead of the hygromycin resistance gene in the original pCEP4 vector from Invitrogen. The structure and exact plasmid sequences can be found in **Supplementary Sequence 1** for the Reviewer's and readers' use. We will also deposit the plasmids in Addgene to make it available to the scientific community upon publication of the manuscript. In the revised manuscript, we corrected the citation and we have changed the designation of our "L1-WT" plasmid to "L1-ORFeus" to emphasize we are using the human codon-optimized, synthetic L1-ORFeus sequence. Additionally, our L1-ORFeus mutant constructs contain the following the modifications: the EN mutant construct contains the H230A residue in ORF2p and the RT mutant construct contains the D702Y residue in ORF2p, which are disrupted residues critical for retrotransposition, as previously shown (Adney et al., 2019; Moran et al., 1996), that we have confirmed here.

In answer to Reviewer 1 (Major point #2), we are providing data that validate the L1-ORFeus constructs we have used in this study. We have confirmed that all L1-ORFeus express comparable ORF1p in HEK293T cells using immunoblotting analysis (**Supplementary Fig. 1g**). We also showed our gating strategy for our flow cytometry analysis to detect GFP+ cells induced with the L1-ORFeus constructs transfected in HEK293T cells, including a negative control (Untransfected) and a positive GFP expression vector control (GFP control, see **new Supplementary Fig. 1c**). We performed a PCR assay across the intron of the GFP reporter gene (intron assay), such that detection of PCR products lacking the intron indicate GFP cDNA integrated into the genome (**Supplementary Fig. 1e**). Lastly, we also conducted RT-PCR to show that L1-ORFeus RNA levels are comparable across the different vectors (**Supplementary Fig. 1f**). Also, our data are consistent with previous reports showing that D702Y RT mutation fully disrupts retrotransposition, and the H230A EN mutation causes a drastic decrease in retrotransposition, based on our flow cytometry analysis and the PCR intron assay (**Supplementary Fig. 1d,e**). We think that overall these data provide a sufficient validation of the L1-ORFeus constructs we have used in this study.

Thus, evaluating the quality of data and conclusions is very complicated. In fact, codon optimization is known to overproduce L1 encoded proteins, and some reports even suggest can alter important aspect of L1 biology such as translation of ORF2 (Alisch et al., 2006 John Moran).

We agree that the codon optimization overproduces L1 encoded proteins and we agree that this is one general limitation of our experimental system as addressed above in the general comment section. In the revised version of the manuscript, we comment on this limitation. In fact, when we measured the abundance of the L1 encoded protein (ORF1p) expressed in HEK293T cells from our plasmids expressing the codon-optimized L1-ORFeus sequence, we found that ORF1p was 4-6 times higher than the endogenous ORF1p protein in HEK293T cells (see the GFP control lane) but comparable to the endogenous ORF1p in iPSC (**Supplementary Fig. 1g**). In the revised manuscript, we generated a new figure showing the relative expression of the LINE-1 ORF1p protein detected using immunoblotting analysis, which we measured by

densitometry using Image J (**Supplementary Fig. 1g**). These results show that, while L1-ORFeus is overexpressed, comparable levels of endogenous ORF1p are seen in certain cell types such as iPSC.

Furthermore, the use of a codon-optimized human L1-ORFeus reporter provides important advantages for the analysis of data as it allows to clearly distinguish L1-ORFeus insertions from insertions of the genomic L1 sequences. Genomic insertions are by far the most common type of insertions observed in CRISPR/Cas9 sites (>70% of insertions, **Fig. 1c**). Because globally L1 sequences comprise ~17% of the human genome (Lander et al., 2001), many of these genomic insertions in CRISPR/Cas9 sites contain L1 sequences without necessarily representing *de novo* retrotransposition events. In contrast, the use of a codon-optimized human L1-ORFeus reporter allowed us to precisely study *de novo* insertions that were distinguished by their sequence from the endogenous L1 elements (**Supplementary Sequence 2**).

Nevertheless, as suggested by this Reviewer, we also sought to assay the insertions at CRISPR/Cas9 DNA breaks from a more “physiological” L1 reporter. For this, we used a construct containing a full-length endogenous L1 sequence competent of retrotransposition (L1RP). The L1RP reporter was modified from the Boeke Lab (An et al., 2011) and published by us more recently (Addgene plasmid #131392) (Ardeljan et al., 2020). The L1RP sequence was mutated to generate the EN mut (H230A) (L1RP-ENm) vector or the RT mut (D702Y) (L1RP-RTm) vector. The L1RP reporter contains the native L1 5’UTR promoter and the 3’ GFP reporter cassette with an artificial intron (**new Supplementary Fig. 8a**). The structure and full sequence of the L1RP reporter vector are shown in **new Supplementary Sequence 13**.

First, we validated these new reporters by several assays. Immunoblot analysis showed an expression of the ORF1p protein consistent across L1RP, L1RP-ENm and L1RP-RTm reporters and slightly lower than observed with the L1-ORFeus (**new Supplementary Fig. 8c**). The percentage of GFP-expressing cells was lower with the L1RP reporter compared to L1-ORFeus reporter (**new Fig. 6a**). Consistently, the PCR assay across the intron of the GFP reporter gene showed that the removal of the intron in the L1RP reporter was less efficient than in the L1-ORFeus reporter, and not observed in the L1RP-ENm and L1RP-RTm mutants (**new Fig. 6b**).

Based on these validations, we repeated key experiments in HEK293T cells with these L1RP reporters transfected at lower amounts (see below response to Major Point #2) (**new Supplementary Fig. 8b**). Consistent with the L1-ORFeus results, we detected *de novo* L1RP retrotransposition events at CRISPR/Cas9 target site by PolyA-seq, indicating that also a more “physiological” L1RP reporter can retrotranspose at CRISPR/Cas9 genome editing sites. Remarkably, no insertions were observed with the L1RP-RTm reporter and only few insertions with the L1RP-ENm reporter (**new Fig. 6c,d**). The consensus motif of L1RP pre-integration site was lost within ± 5 kb of the CRISPR/Cas9-induced DSB whereas it was readily identified in insertions distant from the DSB (**new Fig. 6e**). Overall, the data obtained with the L1RP reporter were remarkably similar to those obtained with the L1-ORFeus.

Additionally, what is Empty Vector (EV)? It can't be pCEP4 (Invitrogen) as stated in methods. In

all PCRs shown in the manuscript, EV transfected cells amplify a 292bp band indicative of retrotransposition, but why?? This is very confusing.

We apologize for not more clearly describing the Empty Vector (EV) plasmid. We included the structure and entire sequence of this vector in **Supplementary Sequence 1**. The vector backbone is pCEP4, in which an intron-less EGFP reporter was cloned, is described in details in **Supplementary Sequence 1** of the manuscript, as well as in the **new Supplementary Fig. 1a**. The presence of the intron-less EGFP reporter in Empty vector (EV) explains the PCR-amplified 292bp band that we used as a positive control of intron-removed band as explained in the revised figure legend of **Supplementary Fig. 1e**. For clarity, we have changed the designation of the “Empty Vector (EV)” to “GFP control vector (GFP control)” to avoid any additional misunderstanding.

Which human L1 was used as a basis for the codon optimization? Does the construct used contain the 5'UTR of the L1? Where were the retrotransposition indicator cassettes cloned? etc A proper description of materials is not only needed to be able to review this study, but also for colleagues aiming to replicate these data.

The human codon-optimized, synthetic L1-ORFeus sequence we used in our plasmids is the sequence published by the Boeke Lab (An et al., 2011). The complete nucleotide sequences and structure of all the L1-ORFeus reporters are provided in **Supplementary Sequence 1**. All the reporters will be deposited in Addgene to make them available to the scientific community upon publication of the manuscript. Regarding the question of the 5'UTR, our L1-ORFeus sequence do not contain the 5'UTR of L1 as shown in the **new Supplementary Fig. 1a**. In contrast, the L1RP reporter described in Major point #1 contains the 5'UTR (**new Supplementary Fig. 8a**).

2. Retrotransposition assays: beside not knowing which plasmids were used, authors transfect 5-times the amount of L1 plasmids normally used in 2×10^5 cells (see Koperka et al., for a recent review). Why? Using a codon optimised L1, although clearly increase expression and retrotransposition, is far from physiological; if on top authors transfected 5-times the amount of DNA normally used, I am not sure whether authors are chasing an artefact created by heavily overexpressing L1 in transformed cells. Thus, authors need to titrate the amount of plasmid transfected, ideally using a natural L1, control for toxicity, and then revisit the whole series of experiments included in this study (from Figure 1).

As described in the Method section and noted by the Reviewer, we used 5 μg of DNA in 2×10^5 HeLa cells as starting material for our transfections. There are two publications listed as Koperka et al: *Methods Mol Biol.* 2016; 1400: 139–156 (Koperka et al., 2016b) and *Methods Mol Biol.* 2016; 1400: 339–355 (Koperka et al., 2016a). Because we are using 6-well plates, we referred to *Methods Mol Biol.* 2016; 1400: 139–156 (Koperka et al., 2016b). In this publication the suggested protocol is 1 μg of DNA for 2×10^4 cells (see image below). The suggested number of cells is 10 times less than what we use. Therefore, if corrected for the number of cells provided by this reference, we are actually transfecting 0.5 μg of DNA in 2×10^4 cells, which is 50% less than Koperka et al.

3.1 LINE-1 retrotransposition assay in HeLa-JVM cells (*in cis*)

1. Day 1 - Plate cells: Seed 2×10^4 HeLa-JVM cells in each well of a 6-well tissue culture plate in HeLa-JVM DMEM growth media. Cells are grown in a humidified incubator at 37°C with 7% CO₂ (see Note 1).
2. Day 2 – Transfect cells: Cells typically are transfected 14 to 16 hours post-plating, day zero (d0) (Figure 1B), using the FuGENE® 6 transfection reagent following the manufacturer's instructions. Every retrotransposition assay should include the following transfection conditions: 1) a vector-only (pCEP4) or mock transfection; 2) a wild type LINE-1 retrotransposition plasmid (*e.g.*, pJM101/L1.3, pJJ101/L1.3, or pLRE3-*mEGFP1*), which serves as a positive control; and 3) a mutant LINE-1 plasmid (*e.g.*, pJM105/L1.3 has a mutation in the RT domain of L1 (51)), which serves as a negative control. To assay for retrotransposition, prepare a transfection mix in a 1.5 mL microcentrifuge tube containing 1 µg pCEP4 or an LINE-1 expression plasmid (Figure 1A) and 3 to 4 µL of FuGENE® 6 in 100 µL of Opti-Mem® I. Incubate the solution at room temperature for 20 minutes. Add the transfection mix to the growth

Nevertheless, we have performed new experiments by transfecting lower amount of reporter vectors according to the Reviewer's suggestion.

- 1) We included one more cell line, U2OS cells that is commonly used in the L1 field (Benitez-Guijarro et al., 2018; MacLennan et al., 2017), and titrated the amount of transfected L1-ORFeus plasmids to 1 µg in 2×10^5 U2OS cells (see revised **Materials and Methods section**). Key findings were reproduced in U2OS cells with CRISPR/Cas9 editing the *MYC* locus and expressing L1-ORFeus or L1-RTm (**new Supplementary Fig. 7**).
- 2) As answered in Major point #1, the L1RP reporters were transfected at 1 µg in 8×10^5 HEK293T cells (see revised **Materials and Methods section**) and confirmed retrotransposition of L1RP elements into CRISPR/Cas9 DSBs by PolyA-seq (**new Fig. 6c,d**).
3. Choice of cells for retrotransposition assays. HEK293T cells are known to over replicate SV40-containing plasmids, as pCEP4. When considering that authors used 5-times more DNA than usual, and a codon optimized L1 (human??), the expression level of L1 encoded proteins in assays is far from physiological. Consistently, FigS1e shows that transfected HEK293Ts express more L1-ORF1p than hESCs, which are a positive control of L1 expression.

We agree with these points raised by the Reviewer, and we think we provided answers to most of these concerns by the data generated in two cell lines different from HEK293T cells (HeLa and U2OS cells) as well as with new data obtained in HEK293T cells with L1RP vectors.

Regarding the expression levels achieved with L1-ORFeus in HEK293T cells, we agree that the human codon-optimized L1-ORFeus is expressed to levels that are higher than the H1 hESCs but not higher than iPS cells (**Supplementary Fig. 1g**). However, the expression of ORF1p is

variable in different hESC clones. For example, ORF1p differs in abundance between the H9 HECs and H13 hESCs as shown in the figure 1j,k below (Wissing et al., 2012). Therefore, it is likely that the H1 hESCs we used have low expression of ORF1p. In contrast, iPSCs expressed relatively abundant ORF1p similar to HEK293T cells expressing L1-ORFeus (**Supplementary Fig. 1g**). Importantly, iPSCs are likely to be used more often than hESCs in clinical applications with canonical CRISPR/Cas9 (De Masi et al., 2020). Therefore, while L1-ORFeus is overexpression system that could overestimate the findings, the use of this reporter has made it feasible to collect many events, revealing insertions that in clinical applications would be rare.

Figure 1 from (Wissing et al., 2012) . (J) Western blot analysis of RNP with an L1 ORF1p antibody. (K) Western blot analysis using a polyclonal antibody against ORF1p in ribonucleo-proteins isolated from iAND-4 and MSUH001 iPSCs. b-Actin served as a loading control.

Thus, authors should use a primary cell type and physiological L1 expression levels when analysing whether L1 could (or not) use CRISPR-induced DSBs for integration.

We agree with this point raised by the Reviewer, and we tried to reproduce data in a primary cell type by using iPSC cells. Despite trying several conditions of electroporation, transfection efficiency in iPS cells was extremely low resulting in a very low number of GFP-expressing cells (see Figure below), consistent with previous report of the low efficiency of transfection and retrotransposition in iPS cells (Wissing et al., 2012) (see Figure 4E below). We found this precluded insertion sequencing.

Figure for Reviewer's use. Florescence Imaging of iPSC either untransfected or transfected with pCEP4 GFP control vector showing a very low number of GFP-expressing cells.

Figure 4E from (Wissing et al., 2012). Flow cytometry plots of iPS-F iPSCs and HDF-F cells either Untransfected or transfected with pLRE3-EF1-mEGFP reporter (harvested 4 days post-transfection) or the pLRE3-EF1-mEGFP(Δ intron) control plasmid (harvested 2 days post-transfection at the peak of EGFP expression).

Regarding physiological L1 expression levels, we used the more “physiological” L1RP reporter as described in the response to Major point #1 and #2 and confirmed retrotransposition of L1RP into CRISPR/Cas9 DSBs by PolyA-seq (**new Fig. 6c,d**).

4. Retrotransposition/CRISPR experiments. According to the scheme shown in Fig 1b, I doubt authors could control for retroviral integration, as at the time of infection cells were already resistant to puro. One more time, there are key details missing (MOI used, did authors used polybrene, etc) that make interpreting these data very problematic/impossible. Because cells were already puro resistant, how could authors control for cleaving efficiency of Cas9? How do they compare cells transfected with WT-L1 (human??) with ENm-L1 if there is

no way to ensure that the same number of DSBs are generated? There is no description of how experiments shown in Figure 1f and S2c, which measure efficiency of cleaving by Cas9, were made. Again, lack of details prohibits to properly evaluating these data.

We now provide all the technical details of the lentivirus production, transduction and puromycin selection in the revised version of the manuscript (see revised **Materials and Methods section**). We agree with the Reviewer that it is essential that the editing efficiencies are comparable across all experimental conditions (all cells transfected with the L1-ORFeus reporters) for the correct interpretation of the data. To this end, we want to highlight that we have measured the editing efficiency for canonical CRISPR/Cas9 based on the generation of indels, which is the most common method to estimate editing efficiency (Rao et al., 2021). We show that the editing efficiency of canonical CRISPR/Cas9 is comparable, ~30%, at the *MYC* locus in HEK293T cells transfected with all L1-ORFeus reporters (**Supplementary Fig. 1h**). Likewise, the editing efficiency of canonical CRISPR/Cas9 is comparable at the *RAG1* locus in cells transfected with for all the L1-ORFeus reporters (**Supplementary Fig. 2c**). Therefore, the editing efficiency is overall comparable at targeted sites by canonical CRISPR/Cas9 in cells transfected with different L1-ORFeus plasmids.

5. DATA INTERPRETATION: Frankly, it is already weird that a mutation known to reduce L1 retrotransposition by >50-fold (RT mutant) could generate L1 insertions, but how can cells transfected with EMPTY VECTOR (EV) be able to generate L1 insertions? See Fig 1d. This is simply impossible. What are authors exactly sequencing?

We thank the reviewer for this point and we would like to provide additional clarification. For Empty Vector (EV) (now called GFP control in the revised manuscript) there were only 2 insertions for canonical CRISPR/Cas9 cutting the *MYC* locus and 4 insertions for the *RAG1* locus (**Fig. 1d**). However, as shown in **Fig. 1e** for the *MYC* locus and in **Supplementary Fig. 2f** for the *RAG1* locus, these 6 sequences aligned to the GFP portion of the GFP control that indeed contains an intron-less EGFP sequence. Plasmids are known to integrate into DSBs caused by CRISPR/Cas9 cutting (Liu et al., 2021; Norris et al., 2020; Skryabin et al., 2020). Thus, we interpret these sequences as integration of the GFP control plasmid into the CRISPR/Cas9 initiated DSBs. Although we induced CRISPR/Cas9 DSBs 5 days after L1 plasmid transfection, still few copies of plasmid are present in cells at the time of CRISPR/Cas9 cutting. We think that it is important to show these rare sequences because they provide a quantification of rare plasmid insertions (for GFP control and L1-ORFeus plasmids); this contrasts strikingly with the hundreds of *de novo* L1-ORFeus insertions found in cells transfected with L1-ORFeus plasmid. In the revised manuscript, we discussed these rare empty vector insertions and provided this interpretation.

In the case of the rare insertions seen with the L1-RTm reporter, we also think that they are likely insertions of the L1-RTm plasmid, trans-complementation from endogenous active L1s being exceedingly rare. In the revised manuscript we address these alternative possibilities. For example, in the case of CCR5, the single insertion still observed in L1-RTm is now shown in

the **new Supplementary Sequence 10**. This sequence mapped to the intron of the GFP cassette, suggesting a plasmid insertion rather than a product of retrotransposition.

6. Similarly, the presence of introns in sequenced L1 integrations is inconsistent with bona fide L1 Reverse Transcription mediated processes. In fact, considering the serious over transfection of L1 plasmids and their replication in HEK293T (points 2&3), and the inability to control for Cas9 cleaving (point 4), it is very likely that a large proportion of “L1 integrations” might simply reflect PCR artefacts, which are common when applying NGS to L1 sequences.

We thank the reviewer for this point and we would like to provide clarification. We are aware that artefacts might arise from PCR and NGS approaches. However, we want to highlight that our NGS sequencing approach have extremely low background in all experiments. For example, we did not detect any sequence mapping to the intron of the GFP reporter cassette in the L1-RTm samples (**Fig. 1e, Supplementary Fig. 2f**). Moreover, insertions mapping to the intron of the GFP reporter cassette are consistently absent in cells edited with PE editors or BE editors even in cells that were transfected with the L1-ORFeus plasmid (**Fig. 7d, Supplementary Fig. 11c,d**). Because all samples from all experiments were transfected, amplified, and sequenced using the same primers with the same approach, we think it is unlikely that the insertions mapping to the intron of the GFP reporter cassette found in L1-ORFeus edited with canonical CRISPR/Cas9 are PCR artefacts.

In contrast, we think that these few rare and unexpected insertions mapping to the intron of the GFP cassette, that are much lower than the flanking GFP exons, depend on RT activity as they are seen only in L1-ORFeus and L1-ENm samples. While we do not have a definitive explanation, they might be due to an inefficient intron removal during the pre-mRNA splicing, that can occur in experiments conducted with a transfected plasmid reporter (Nasim and Eperon, 2006). The single sequence mapping to the intron of the GFP cassette observed in the L1-RTm sample in cells with RAG1 DSBs (**Supplementary Fig. 2f**) could be a rare plasmid insertion. We have commented on this point on the revised version of the manuscript.

7. While poly-A-seq data is consistent with L1 biology, which is not surprising when considering that authors used the same method (or a very similar approach) recently employed to characterize >80K insertions in a variety of physiologically relevant human cells (Flasch et al., 2019), the same is not true for the non-polyA datasets. In fact, there is no validation of the non-polyA method (i.e., not a single L1 insertion has been characterized at both ends). This is not trivial, as Flasch et al reported that when using the poly-A-seq method, as much as 40% of all sequencing CCSs (Pac Bio) correspond to plasmid artefacts.

In fact, when looking at the position of primers in the non-polyA method (FigS4), EGFP expression is expected from insertions captured using this method, Then, why authors do not detect EGFP-expressing cells in retrotransposition assays using L1-ENm but can detect as much as 30% the number of insertions detected with WT-L1. In other words, at day 11, nearly 10% of cells transfected with WT-L1 express EGFP, and using the non-polyA method, authors captured 127K insertions; however, when using the ENm-L1, EGFP expression of EGFP is presumably

0% but authors can sequence nearly 39K insertions. This strongly suggest that the majority of insertions captured using the non-polyA method might correspond to artifacts.

Authors need to validate many de novo L1 integrations using the non-polyA method, at both ends.

For PolyA-seq, it is designed based on similar approaches previously validated and published (Flasch et al., 2019; Sultana et al., 2019). In particular, the primer design, library generation and data analysis of PolyA-seq approach is largely identical to a previously validated technique (Flasch et al., 2019). All these high throughput sequencing techniques are designed to capture the 3' junctions of L1 (**Supplementary Fig. 4a,b**). The analysis of 3' junctions of the L1-ORFeus insertions captured by our PolyA-seq showed the expected consensus motif 5'-TT/AAAAA-3' (**Supplementary Fig. 4d**), and the analysis of the 3' poly(A) tracts (range 15–75bp) (**Fig. 3a, Supplementary Fig. 4e**) were consistent with previous reports(Flasch et al., 2019; Sultana et al., 2019).

In contrast, the non-PolyA-seq is a novel sequencing approach developed by us to further validate the data obtained by the published PolyA-seq approach. The data obtained with the non-PolyA-seq were remarkably comparable, but not superior, to those obtained with PolyA-seq, giving us confidence in their validity. Nevertheless, since PolyA-seq and non-PolyA-seq data were so comparable and somehow redundant, we streamlined the revision by presenting only PolyA-seq data in detail. Therefore, all the non-PolyA-seq data and figures were removed from the revised manuscript.

8. While the comparisons with PE&CBE are nice, they don't represent an apple vs apple comparison with canonical CRISPR-Cas9. Thus, to truly demonstrate that L1 preferentially target DSBs, authors should use a nickase Cas9 in the context of LentiCRISPR, as this would be a side-by-side comparison with Cas9.

We agree that comparison between canonical Cas9 and nickase-Cas9 is an important comparison. We included experiments with nickase-Cas9 targeted to the MYC and FANCF loci using the “nick only for PE3 system” as illustrated in **Fig. 7a**. The “nick only for PE3 system” expresses only the nickase-Cas9 and the sgRNA and is therefore a direct comparison to the canonical CRISPR-Cas9 system. The results showed that L1-ORFeus, plasmid or genomic insertions were rarely generated at the targeted sites by this nickase-Cas9 using the “nick only for PE3” strategy (**Fig. 7b,c; Supplementary Fig. 9e; Supplementary Fig. 10e,f**). As expected, these results demonstrated that L1-ORFeus insertions are much more frequent at the sites targeted by canonical CRISPR/Cas9 versus at the sites targeted by the nickase-Cas9 system. Insertions are also rare at the targeted sites using PE and BE systems, which also use nickase-Cas9 and tend to generate DNA nicks. Based on these findings, we suggest that L1 insertion occur preferentially at DSBs. We have modified the revised manuscript to describe these findings.

MINOR POINTS

a) INTRODUCTION: Is over negative and only focus on limitations of CRISPR/Cas9. To be fair and collegial, authors should, at the very least, comment on the expectations that CRISPR/Cas9 offer in Biomedicine.

We agree with this point. We revised the introduction to highlight the high promise that the CRISPR/Cas9 approach will change biomedicine.

b) INTRODUCTION: Tone down statements such as: “All retrotransposition in humans depends on a reverse transcriptase (RT) encoded by the long-interspersed element-1 (LINE-1, or L1)²⁶.”

While likely true, authors should also acknowledge that: 1) There are HERV with coding potential for pol; 2) perhaps more relevant, some DNA polymerases are known to have intrinsic RT activity; and 3) Telomerase is just another RT source in human cells.

We agree with these points and revised the manuscript accordingly.

c) INTRODUCTION: There are many many references missing. Below I include a few examples, but admittedly, the Introduction needs substantial revision.

- Cite Beck et al., Cell-2010 when referring to number of active L1s in humans. By the way, the accepted nomenclature for these is “Retrotransposition Competent L1s”
- Add additional known activities of L1-ORF1p and add references!
- Add Symer et al., Embo 2002 for TPRT of human L1s.
- Add references when referring to the consensus sequence of L1-EN: Jurka-1997 and Flasch et al., 2019 among others.
- etc

We thank the Reviewer for this comment and agree that some key references were missing and we revised the manuscript to include all those indicated by the Reviewer, as well as other important references.

d) Add sequence of primers used to explore L1 retrotransposition by regular PCR

In the revised manuscript we include all the primers used in this work in **Supplementary Table 4**.

e) Figures and panels are all over the place, jumping from Fig 1 to 3 etc etc. Why not using a chronological referral to Figures as most studies do?

We agree with this point, and the revised manuscript accordingly.

Cited references

Adney, E. M., Ochmann, M. T., Sil, S., Truong, D. M., Mita, P., Wang, X., Kahler, D. J., Fenyö, D., Holt,

L. J., and Boeke, J. D. (2019). Comprehensive Scanning Mutagenesis of Human Retrotransposon LINE-1 Identifies Motifs Essential for Function. *Genetics* 213, 1401-1414.

An, W., Dai, L., Niewiadomska, A. M., Yetil, A., O'Donnell, K. A., Han, J. S., and Boeke, J. D. (2011). Characterization of a synthetic human LINE-1 retrotransposon ORFeus-Hs. *Mob DNA* 2, 2.

Ardeljan, D., Steranka, J. P., Liu, C., Li, Z., Taylor, M. S., Payer, L. M., Gorbounov, M., Sarnecki, J. S., Deshpande, V., Hruban, R. H., *et al.* (2020). Cell fitness screens reveal a conflict between LINE-1 retrotransposition and DNA replication. *Nature structural & molecular biology* 27, 168-178.

Benitez-Guijarro, M., Lopez-Ruiz, C., Tarnauskaitė, Ž., Murina, O., Mian Mohammad, M., Williams, T. C., Fluteau, A., Sanchez, L., Vilar-Astasio, R., Garcia-Canadas, M., *et al.* (2018). RNase H2, mutated in Aicardi-Goutières syndrome, promotes LINE-1 retrotransposition. *Embo j* 37.

De Masi, C., Spitalieri, P., Murdocca, M., Novelli, G., and Sangiuolo, F. (2020). Application of CRISPR/Cas9 to human-induced pluripotent stem cells: from gene editing to drug discovery. *Human Genomics* 14, 25.

Flasch, D. A., Macia, A., Sanchez, L., Ljungman, M., Heras, S. R., Garcia-Perez, J. L., Wilson, T. E., and Moran, J. V. (2019). Genome-wide de novo L1 Retrotransposition Connects Endonuclease Activity with Replication. *Cell* 177, 837-851 e828.

Frangoul, H., Altshuler, D., Cappellini, M. D., Chen, Y. S., Domm, J., Eustace, B. K., Foell, J., de la Fuente, J., Grupp, S., Handgretinger, R., *et al.* (2021). CRISPR-Cas9 Gene Editing for Sickle Cell Disease and beta-Thalassemia. *The New England journal of medicine* 384, 252-260.

Gillmore, J. D., Gane, E., Taubel, J., Kao, J., Fontana, M., Maitland, M. L., Seitzer, J., O'Connell, D., Walsh, K. R., Wood, K., *et al.* (2021). CRISPR-Cas9 In Vivo Gene Editing for Transthyretin Amyloidosis. *The New England journal of medicine* 385, 493-502.

Kopera, H. C., Flasch, D. A., Nakamura, M., Miyoshi, T., Doucet, A. J., and Moran, J. V. (2016a). LEAP: L1 Element Amplification Protocol. *Methods Mol Biol* 1400, 339-355.

Kopera, H. C., Larson, P. A., Moldovan, J. B., Richardson, S. R., Liu, Y., and Moran, J. V. (2016b). LINE-1 Cultured Cell Retrotransposition Assay. *Methods Mol Biol* 1400, 139-156.

Lander, E. S., Linton, L. M., Birren, B., Nusbaum, C., Zody, M. C., Baldwin, J., Devon, K., Dewar, K., Doyle, M., FitzHugh, W., *et al.* (2001). Initial sequencing and analysis of the human genome. *Nature* 409, 860-921.

Liu, M., Zhang, W., Xin, C., Yin, J., Shang, Y., Ai, C., Li, J., Meng, F. L., and Hu, J. (2021). Global detection of DNA repair outcomes induced by CRISPR-Cas9. *Nucleic Acids Res.*

MacLennan, M., García-Cañadas, M., Reichmann, J., Khazina, E., Wagner, G., Playfoot, C. J., Salvador-Palomeque, C., Mann, A. R., Peressini, P., Sanchez, L., *et al.* (2017). Mobilization of LINE-1 retrotransposons is restricted by Tex19.1 in mouse embryonic stem cells. *eLife* 6.

Moran, J. V., Holmes, S. E., Naas, T. P., DeBerardinis, R. J., Boeke, J. D., and Kazazian, H. H., Jr. (1996). High frequency retrotransposition in cultured mammalian cells. *Cell* 87, 917-927.

Nasim, M. T., and Eperon, I. C. (2006). A double-reporter splicing assay for determining splicing efficiency in mammalian cells. *Nature protocols* 1, 1022-1028.

Norris, A. L., Lee, S. S., Greenlees, K. J., Tadesse, D. A., Miller, M. F., and Lombardi, H. A. (2020). Template plasmid integration in germline genome-edited cattle. *Nature Biotechnology* 38, 163-164.

Philippe, C., Vargas-Landin, D. B., Doucet, A. J., van Essen, D., Vera-Otarola, J., Kuciak, M., Corbin, A., Nigumann, P., and Cristofari, G. (2016). Activation of individual L1 retrotransposon instances is restricted to cell-type dependent permissive loci. *Elife* 5.

Rao, S., Yao, Y., Soares de Brito, J., Yao, Q., Shen, A. H., Watkinson, R. E., Kennedy, A. L., Coyne, S., Ren, C., Zeng, J., *et al.* (2021). Dissecting ELANE neutropenia pathogenicity by human HSC gene editing. *Cell stem cell* 28, 833-845 e835.

Skryabin, B. V., Kummerfeld, D.-M., Gubar, L., Seeger, B., Kaiser, H., Stegemann, A., Roth, J., Meuth, S. G., Pavenstädt, H., Sherwood, J., *et al.* (2020). Pervasive head-to-tail insertions of DNA templates mask desired CRISPR-Cas9-mediated genome editing events. *Science Advances* 6, eaax2941.

Sultana, T., van Essen, D., Siol, O., Bailly-Bechet, M., Philippe, C., Zine El Aabidine, A., Pioger, L., Nigumann, P., Saccani, S., Andrau, J. C., *et al.* (2019). The Landscape of L1 Retrotransposons in the Human Genome Is Shaped by Pre-insertion Sequence Biases and Post-insertion Selection. *Molecular cell* 74, 555-570 e557.

Wissing, S., Munoz-Lopez, M., Macia, A., Yang, Z., Montano, M., Collins, W., Garcia-Perez, J. L., Moran, J. V., and Greene, W. C. (2012). Reprogramming somatic cells into iPS cells activates LINE-1 retroelement mobility. *Human molecular genetics* 21, 208-218.

Zingler, N., Willhoeft, U., Brose, H. P., Schoder, V., Jahns, T., Hanschmann, K. M., Morrish, T. A., Lower, J., and Schumann, G. G. (2005). Analysis of 5' junctions of human LINE-1 and Alu retrotransposons suggests an alternative model for 5'-end attachment requiring microhomology-mediated end-joining. *Genome research* 15, 780-789.

Reviewers' Comments:

Reviewer #1:

Remarks to the Author:

Manuscript: Tao et al. (NCOMMS-21-38175)

Title: Frequency and mechanisms of LINE-1 retrotransposon insertions at CRISPR/Cas9 Sites

The authors have addressed every major criticism and all minor comments thoroughly and satisfactorily. However, there are two points in the context of the authors response to major criticism 1 and 2 that still need to be addressed before the manuscript can be accepted for publication:

Ad authors response to Major Criticism 1:

Introduction/Page 4, line 80: Here the authors suddenly and unexpectedly present the expression 'L1-ORFeus' at the end of the introduction without any explanation what L1-ORFeus is and how it is defined. It is not comprehensible to the reader, why the authors suddenly switch to the expression 'L1-ORFeus' although, in the preceding section, the biology of endogenous L1 elements was described. The authors suddenly switch to the phrase 'L1-ORFeus' although L1-ORFeus is very different from endogenous wildtype L1s. Therefore, the authors have to make an effort to explain what L1-ORFeus is in a couple of sentences in the last paragraph.

In their rebuttal letter (page 4, 2nd paragraph), the authors provide a nice explanation for using the codon-optimized human L1-ORFeus reporter. ('The reason we use.... (Supplementary Sequence 2)). I would recommend to include a modified/shorter version of this explanation in the introduction to make the reader understand at the beginning, why L1-ORFeus was used instead of a wildtype L1 elements as reporter element.

Ad authors response to Major Criticism 2:

Suppl. Figs 1e, 2b, 3b, 7b, 8a and b, 9c, 10 a and b, 13 a-c, Fig 6b: How do the authors know that the PCR products identified in the presented agarose gels have a size of 292 bp and 1196 bp if they do not run the PCR products against any size marker/DNA ladder on these gels? Therefore, the authors are asked to replace the presented images of 'Intron-Assay' agarose gels by images of intron assays that INCLUDE a size marker/DNA ladder lane of the kind the authors presented in Suppl. Fig. 1i. Agarose gels including a sizemarker /DNA ladder lane allow the readers to decide themselves if the presented PCR products have indeed the length of 292 bp and 1196 bp as suggested by the authors.

Minor criticism:

Introduction/Page 4 (first paragraph): For the sake of completeness the authors should also include information about the third L1-encoded ORF, termed ORF0, (Denli et al.2015) which coded for a 72 aa-polypeptide, in the Introduction section.

- New Suppl. Fig. 1a: The authors have to indicate in the figure itself and in the figure legend, which promoter controls expression of the L1 reporter cassettes in the pCEP4-derived constructs—the authors only state in their rebuttal letter that it is NOT the 5'UTR.

- New Suppl. Fig. 1g: The authors are asked to provide information about the used iPSC line: What is the parental celltype of the iPSC line? Where is the iPSC line characterized and/ or published?

Manuscript: Tao et al. (NCOMMS-21-38175A-Z)

Title: Frequency and mechanisms of LINE-1 retrotransposon insertions at CRISPR/Cas9 sites

Point-by-point rebuttal letter

Reviewer #1:

The authors have addressed every major criticism and all minor comments thoroughly and satisfactorily.

We would like to thank the reviewer for his/her recognition that our revision “have addressed every major criticism and all minor comments thoroughly and satisfactorily”.

However, there are two points in the context of the authors response to major criticism 1 and 2 that still need to be addressed before the manuscript can be accepted for publication.

We have incorporated the answers to this Reviewer’s concerns in a newly revised version of the manuscript as described below.

Ad authors response to Major Criticism 1:

Introduction/Page 4, line 80: Here the authors suddenly and unexpectedly present the expression ‘L1-ORFeus’ at the end of the introduction without any explanation what L1-ORFeus is and how it is defined. It is not comprehensible to the reader, why the authors suddenly switch to the expression ‘L1-ORFeus’ although, in the preceding section, the biology of endogenous L1 elements was described. The authors suddenly switch to the phrase ‘L1-ORFeus’ although L1-ORFeus is very different from endogenous wildtype L1s. Therefore, the authors have to make an effort to explain what L1-ORFeus is in a couple of sentences in the last paragraph.

In their rebuttal letter (page 4, 2nd paragraph), the authors provide a nice explanation for using the codon-optimized human L1-ORFeus reporter. (“The reason we use.... (Supplementary Sequence 2)). I would recommend to include a modified/shorter version of this explanation in the introduction to make the reader understand at the beginning, why L1-ORFeus was used instead of a wildtype L1 elements as reporter element.

We agree with this point, and have revised the introduction to include a description of the L1-ORFeus reporter.

Ad authors response to Major Criticism 2:

Suppl. Figs 1e, 2b, 3b, 7b, 8a and b, 9c,10 a and b, 13 a-c, Fig 6b: How do the authors know that the PCR products identified in the presented agarose gels have a size of 292 bp and 1196 bp if

they do not run the PCR products against any size marker/DNA ladder on these gels? Therefore, the authors are asked to replace the presented images of 'Intron-Assay' agarose gels by images of intron assays that INCLUDE a size marker/DNA ladder lane of the kind the authors presented in Suppl. Fig. 1i. Agarose gels including a sizemarker /DNA ladder lane allow the readers to decide themselves if the presented PCR products have indeed the length of 292 bp and 1196 bp as suggested by the authors.

We agree with the Reviewer's comment, and we replaced all the figures that show agarose gels with figures that include the DNA ladder. Moreover, uncropped figures of the same gels are now also provided in the Source Data file.

Minor criticism:

Introduction/Page 4 (first paragraph): For the sake of completeness the authors should also include information about the third L1-encoded ORF, termed ORF0, (Denli et al.2015) which coded for a 72 aa-polypeptide, in the Introduction section.

We thank the Reviewer for this suggestion and a brief description of the ORF0 has been added together with ORF1p and ORF2p in the newly revised version of introduction.

New Suppl. Fig. 1a: The authors have to indicate in the figure itself and in the figure legend, which promoter controls expression of the L1 reporter cassettes in the pCEP4-derived constructs—the authors only state in their rebuttal letter that it is NOT the 5'UTR.

The CMV promoter, which is the promoter controlling the expression of the L1 reporter cassette, has been added to the figure as well as the figure legend of a revised New Suppl. Fig. 1a accordingly. It has been also added to the Methods section.

New Suppl. Fig. 1g: The authors are asked to provide information about the used iPSC line: What is the parental cell type of the iPSC line? Where is the iPSC line characterized and/or published?

The iPSC line is WTC-11, provided by our collaborator Dr. William T. Pu at Boston Children's Hospital/Harvard Stem Cell Institute (HSCI). This contribution is now specified in the Acknowledgement section.

Dr. Pu lab purchased the WTC-11 line from Coriell Institute (https://www.coriell.org/0/Sections/Search/Sample_Detail.aspx?Ref=GM25256). Briefly, this iPSC line (WTC-11) is an iPSCs derived from a healthy donor, submitted by Dr. Bruce R. Conklin (Gladstone Institute of Cardiovascular Disease, UCSF).

Now we have updated the information in revised New Suppl. Fig. 1g accordingly.